



# Chemical and isotopic composition of secondary organic aerosol generated by $\alpha$-pinene ozonolysis

Carl Meusinger[1], Ulrike Dusek[2,3], Stephanie M. King[1,4], Rupert Holzinger[2], Thomas Rosenørn[1,5], Peter Sperlich[6,7], Maxime Julien[8], Gerald S. Remaud[8], Merete Bilde[1,9], Thomas Röckmann[2], and Matthew S. Johnson[1]

[1]Department of Chemistry, University of Copenhagen, DK 2100, Copenhagen Ø, Denmark
[2]Institute for Marine and Atmospheric research Utrecht (IMAU), Utrecht University, 3584 CC, Utrecht, The Netherlands
[3]Centre for Isotope Research, Energy and Sustainability Research Institute Groningen, 9747 AG Groningen, The Netherlands
[4]now at: Haldor Topsøe A/S, DK 2800, Kgs. Lyngby, Denmark
[5]Infuser ApS, DK 2200, Copenhagen N, Denmark
[6]Max-Planck Institute for Biogeochemistry, 07745 Jena, Germany
[7]now at: National Institute of Water and Atmospheric Research (NIWA), Wellington 6021, New Zealand
[8]CEISAM, UMR CNRS6230, BP 92208, Nantes 44322 cedex 3, France
[9]now at: Aarhus University, Department of Chemistry, 8000 Aarhus C, Denmark

*Correspondence to:* C.M. (c.meusinger@gmail.com)

**Abstract.**

Secondary organic aerosol (SOA) plays a central role in air pollution and climate. However, an exact description of the sources and mechanisms leading to SOA is elusive despite decades of research. Stable isotope analysis may help to constrain atmospheric SOA budgets but the isotope effects associated with the underlying processes have to be determined in order to do so. In this paper, SOA formation from ozonolysis of $\alpha$-pinene - an important precursor and perhaps the best-known model system used in laboratory studies - was investigated using stable carbon isotope analysis, position-specific isotope analysis (PSIA), and high-resolution chemical analysis based on a thermal-desorption proton-transfer-reaction mass-spectrometer (PTR-MS).

SOA was formed in a constant-flow chamber under dark, dry and low-$NO_x$ conditions, with OH scavengers in the absence of seed particles. Product SOA was collected on doubly-stacked quartz filters (front and back filters). During analysis, the filters were heated stepwise over the range of 100-400 °C to desorb organic compounds that were (i) detected using PTR-MS for chemical analysis and to determine the O:C ratio, and (ii) converted to $CO_2$ for $^{13}C$ analysis. In addition, the total carbon isotopic composition of selected samples was measured. For the first time PSIA has been performed for $\alpha$-pinene.

More than 400 ions in the mass range from 39-800 Da were detected and quantified using the PTR-MS. The largest mass fraction desorbed from the filters at 150 °C. The measured O:C ratio of front filter material increased from 0.18 to 0.25 as the desorption temperature was raised from 100 to 250 °C. The rising trend is consistent with the fact that functionalization decreases the volatility of chemical species. At temperatures above 250 °C the O:C ratio of thermally desorbed material, presumably from oligomeric precursors, was constant. The observation of a number of components across the full range of desorption temperatures suggests that they are generated by thermal decomposition of oligomers.


SOA on front filters was enriched in $^{13}$C by 0.2-2.9 ‰ relative to the initial $\alpha$-pinene, at all desorption temperatures. The total carbon isotopic composition was similar to the enrichment of the major fraction desorbing at 150 °C. Gas-phase compounds desorbing from the back filters showed much lower concentrations but were depleted in $^{13}$C by 0.7 ‰ compared to the initial $\alpha$-pinene and by 1.9 ‰ compared to the corresponding front filter. PSIA showed that the isotope enrichment at individual carbon positions in $\alpha$-pinene ranged from -6.9 to +10.5 ‰ relative to the bulk composition. However, there was not a clear mechanistic connection between those values and the observed isotopic enrichment of bulk SOA. Instead, fragmentation reactions favouring the loss of small, isotopically light products to the gas phase are consistent with the observations. In monoterpene ozonolysis, functionalization is known to follow fragmentation, however it is the fragmentation step that seems to govern the isotope budget. The isotope effect associated with oligomerization is small. The suggested isotope effects are important for the interpretation of isotopic compositions of ambient aerosol.

## 1 Introduction

Secondary Organic Aerosol (SOA) is formed in the atmosphere by oxidation of Volatile Organic Compounds (VOC's). SOA contributes significantly to the aerosol burden of the atmosphere and has impacts on climate, health and visibility (Stocker et al., 2013; Hänninen et al., 2004; Dockery et al., 1993; Andreae and Crutzen, 1997). The formation and processing of SOA in the atmosphere encompasses a complex array of chemical and physical processes, like condensation, evaporation, water uptake, and reactions on the particle surface, rendering it difficult to determine the sources of SOA and predict the effect of SOA on climate and health.

Stable carbon isotopes are an established method of investigating the processing and sources of aerosols in the atmosphere. The possibility to distinguish individual soruces, including marine aerosol (Turekian et al., 2003; Ceburnis et al., 2011) and biomass burning (Kirillova et al., 2013), by isotope measurements has been particularly useful when it comes to source apportionment of ambient aerosol (e.g., Sakugawa and Kaplan, 1995; Narukawa et al., 2008; Turekian et al., 2003; Widory et al., 2004; Ho et al., 2006; Huang et al., 2006; Fisseha et al., 2009; Kirillova et al., 2013; Ceburnis et al., 2011; Fu et al., 2012; Miyazaki et al., 2012; O'Dowd et al., 2014; Masalaite et al., 2015).

An incomplete understanding of how atmospheric processes influence isotopic abundances has however limited the use of isotope information in constraining SOA contributions to the aerosol burden. This is reflected in the literature where both enrichment and depletion in $^{13}$C relative to common sources were found in ambient aerosol: Irei et al. (2015) found that atmospheric processing of aerosol (reactions with OH radicals or ozone) lead to depletion in $^{13}$C in the low-volatile fraction of the aerosol compared to common emission signatures. Fu et al. (2012) on the other hand reported isotopic enrichment in ambient SOA that is distinguishable from biomass burning and primary emissions, possibly hinting at the oxidation of biogenic VOC. Pavuluri et al. (2011) attributed the enrichment they observed in $C_2$-$C_4$ acids in aerosol from Chennai, India, to photochemical ageing during long-range transport.

Different chemical and physical processes fractionate isotopes, leading to isotopic enrichment (or depletion) in SOA. More specifically, carbon isotopic fractionation changes the $^{13}$C/$^{12}$C ratio of a sample (Coplen, 2011). Briefly, there are two types





of isotopic fractionation: equilibrium and kinetic fractionation. Equilibrium fractionation describes the change in the isotopic composition due to unequal partitioning of isotopic analogues of a compound (isotopologues) between two phases. Typically the condensed phase is isotopically enriched relative to the gas phase. Equilibrium fractionation is important for light molecules like water because the extra neutron in isotopes like $^2$H, $^{17}$O (or in other molecules, $^{13}$C) significantly adds to the molecular

mass of the compound. Monoterpenes and their oxidation products have large molecular masses relative to the added 1 amu and the isotopic equilibrium fractionation due to their gas/particle partitioning becomes negligible (Gensch et al., 2011; Irei et al., 2011).

Kinetic fractionation results from differences in reaction rate constants of isotopologues of the reagents. Let $^{12}k$ denote the reaction rate constant for the reaction involving a compound containing only carbon-12 and let $^{13}k$ denote the reaction rate

constant for the reaction involving a single-substituted $^{13}$C isotopologue, then the expression $\varepsilon = {}^{13}k/{}^{12}k - 1$ is called isotopic fractionation. It is common to distinguish "normal" and "inverse" kinetic isotope effects. During normal kinetic fractionation the extra neutron slows chemical reactions (Johnson et al., 2002) and $\varepsilon$ becomes negative (this definition follows the definition of the $\delta$-value in Sect. 2.5.2 but different definitions exist in the literature). Over the course of the reaction the reactant is successively enriched in heavy $^{13}$C and the oxidation product is depleted. Conversely, during inverse kinetic fractionation

$\varepsilon > 1$, depleting the reactant and enriching the product. Once the reaction has come to completion with no reactants left, and assuming no branching to other product channels, the isotopic composition of the product will be identical to that of the reactant at start (for both normal and inverse kinetic fractionation) to satisfy mass balance.

Reactions involved in atmospheric VOC oxidation fall into three categories: functionalization, fragmentation and oligomerization (Kroll and Seinfeld, 2008; Rudich et al., 2007; Chacon-Madrid and Donahue, 2011). The volatility (Donahue et al.,

2006; Jimenez et al., 2009) and oxygen-to-carbon ratio, O:C (Donahue et al., 2011; Kroll et al., 2011), of involved species allow characterising these processes. Chemical analysis based on proton-transfer-reaction mass spectrometry has proven very useful in ambient and laboratory studies investigating aerosol and gas-phase compounds and their properties (e.g., Holzinger et al., 2010a, b; Shilling et al., 2008; Presto and Donahue, 2006). Advantages of the technique include the soft ionisation, high sensitivity, wide range of detectable compounds, and the possibility of quantifying them. Using a proton-transfer-reaction

time-of-flight mass spectrometer (PTR-ToF-MS, shortened to PTR-MS for the remainder of the article) additionally allows chemical characterisation and identification of compounds and estimation of the O:C ratio. The characteristics of the three types of atmospheric VOC reactions and the current understanding on how they contribute to isotopic fractionation (Kirillova et al., 2013, 2014) are summarised as follows:

- Functionalization describes the addition of oxygenated functional groups to the parent compound. A typical example

is the first reaction step in VOC oxidation. Here, products are less volatile and have higher O:C values than the parent compound. Functionalization is typically accompanied by normal kinetic fractionation leading to oxidation products depleted in $^{13}$C (Rudolph and Czuba, 2000). Compound specific laboratory studies have reported kinetic fractionations in ozonolysis of the SOA precursors isoprene and $\beta$-pinene of $\varepsilon_{\text{isoprene+O}_3} = -8.4‰$ (Iannone et al., 2010) and $\varepsilon_{\beta\text{-pinene+O}_3} = -3.6‰$ (Gensch et al., 2011). Kinetic fractionation in the oxidations of many (mostly anthropogenic)





VOC's by OH and ozone have been reported (e.g., Rudolph et al., 2000), but no such data is available for many abundant (biogenic) VOC's, including ozonolysis of $\alpha$-pinene.

– Fragmentation describes cleavage of carbon-carbon bonds possibly followed by addition of oxygen to the fragments. Fragmentation reactions in the condensed phase can result in molecules that are small, like $CO_2$ and $CH_2O$, have higher volatilities than the parent compound and might that escape to the gas phase (Kroll et al., 2009). The O:C ratio of the products is often higher than for the parent VOC and is typically highest for compounds remaining in the aerosol phase. During fragmentation normal kinetic fractionation can enrich the aerosol in [13]C: when some of the depleted reaction products are lost to the gas phase the remaining aerosol phase will be enriched. The effect of fragmentation in the gas phase on isotopic composition is not clear. In one of the few laboratory studies of this effect, Aggarwal and Kawamura (2008) report enrichment of oxalic acid in [13]C during photochemical ageing. During ozonolysis of monoterpenes both functionalization and fragmentation occur simultaneously (Chacon-Madrid and Donahue, 2011). This increases the O:C ratio but the overall effect on the isotopic balance is not well established.

– Oligomerization (sometimes also referred to as accretion) describes the building of larger organic structures from monomers, often in the aerosol phase (Kalberer et al., 2004, 2006; Hallquist et al., 2009). The O:C ratio remains constant during oligomerization while the vapour pressure of the products drops significantly. The formation of complex organic mixtures in the aerosol phase (Cappa et al., 2008) is expected to show similar characteristics in vapour pressure and O:C ratio. The influence of oligomerization on the isotopic composition of the aerosol is not clear. On the one hand, a kinetic isotope effect could be associated with oligomerization. On the other hand, if semivolatile material becomes 'locked' in an oligomer, this will shift the partitioning between gas and particle phase giving rise to fractionation.

Position-specific isotope analysis (PSIA) of the initial reactant offers another way of understanding bulk isotopic data. A microscopic view of the isotopic composition is made possible by mapping the intra-molecular isotope distribution. PSIA by [13]C isotope ratio monitoring by NMR (irm-[13]C NMR) has proven valuable in interpreting a number of (bio)chemical (Bayle et al., 2014a; Botosoa et al., 2009a; Gilbert et al., 2011, 2012) and physical-chemical processes such as distillation and sorption (Botosoa et al., 2008, 2009b; Höhener et al., 2012), leading to a deeper understanding of the underlying phenomena causing isotope fractionation in nature. The position-specific isotope composition could yield unexpected (inverse) isotopic fractionation in atmospheric aerosol. If for example a specific C-atom in $\alpha$-pinene is enriched but preferentially expelled solely due to its position during fragmentation reactions the remaining fragment would be isotopically depleted. Such a process, transferring a pre-existing site-specific enrichment within a molecule to fragments, is distinct from the normal kinetic fractionation that preferentially enriches the remaining aerosol and expels isotopically lighter fragments.

The goal of this study was to determine whether kinetic fractionation or position-specific mechanistic transfer governs isotopic fractionation in ageing of fresh SOA.



## 2 Material and methods

### 2.1 Chemical compounds used

Chamber experiments were performed using the following chemicals: (+)-$\alpha$-pinene (Aldrich, $> 99\,\%$, batch #80796DJV), 1-Butanol (Sigma-Aldrich, $> 99.4\,\%$) and cyclohexane (Labscan, $99.5\,\%$). PSIA was performed on several samples of $\alpha$-pinene from a range of manufacturers, see Table 1. The batch of $\alpha$-pinene used in the chamber experiments could not be analysed using PSIA, because the manufacturer does not supply it anymore.

### 2.2 Chamber design and characteristics

A new aerosol smog chamber was built in Copenhagen based on a steady-state design (King et al., 2009; Shilling et al., 2008; Kleindienst et al., 1999). It consists of a $4.5\,\mathrm{m^3}$ teflon bag mounted inside a temperature controlled insulated room of walk-in size (Viessmann). The design and operation mode of the chamber resemble that of a flow reactor, except for a much smaller surface to volume ratio and with turbulent mixing ensured by the orientations of the inlet tubings. Residence times are of the order of hours compared to the typical time scale of minutes in flow reactors.

Inlet and outlet flanges feed through the insulating walls on opposite sides and provide numerous ports for injection and sampling. The chamber was operated in a constant-flow mode where dry air and reactants were flushed into the chamber constantly using mass flow controllers. A LabView computer program read a differential pressure transmitter (model 5266, Gems Sensors) and triggered an exhaust valve with a high volume pump (VT4.16, Becker) in line in order to restore nominal pressure. The chamber was always operated at slight overpressure (relative to ambient), and the exit valve closed and opened at 2.5 and 3.5 Pa respectively. This range in pressure difference defined a cycle (exit valve open-close-open) that usually took some minutes. Besides the change in flow patterns due to the pressure regulation, there were no other means of mixing the contents of the chamber. All connections that supply the chamber were made of stainless steel, except for the differential pressure sensor line (made of Teflon).

On the inlet side, in house air was dried (SP 14, VarioDry) and cleaned of particulate matter and oil mist (F64L, Norgren) before being fed into the chamber and any other parts of the setup. In the VOC injection system, a syringe pump (NE-300, New Era Pump Systems Inc.) continuously injected a mixture of $\alpha$-pinene and 1-butanol or a mixture of $\alpha$-pinene and cyclohexane into a warmed glass bulb. 1-butanol and cyclohexane were used as OH scavengers. The mixing ratio between $\alpha$-pinene and the OH scavenger was 1:600 (v/v). A flow of air carried the evaporate from the glass bulb into the chamber. The speed of the syringe pump and the total flow defined the VOC concentration inside the bag. The VOC injection system gave rise to minor fluctuations in the VOC concentrations due to bubbles forming in the syringe. A small flow of clean, dry air (0.1 L/min) directed over a Hg lamp emitting UV light (model 600, Jelight company Inc.) generated ozone which was fed into the chamber separately.

The total inlet flow consisted of 20 L/min dry air, 2 L/min $\alpha$-pinene + OH scavenger in air and 0.1 L/min ozone-rich air. Under these conditions, the (nominal) residence time was $\tau_{\mathrm{nominal}} = 3.4\,\mathrm{h}$, which is long compared to the open-close cycle





of the pressure regulation system. $\tau_{\mathrm{nominal}}$ was confirmed during simple dilution experiments. The flows (and therefore the residence time inside the chamber) were kept constant during all experiments described here.

On the outlet side, the generated aerosol was sampled after an ozone scrubber on doubly stacked quartz-fibre filters (4.7 cm diameter, QMA 1851, Whatman) for offline chemical and isotope analysis at $10\,\mathrm{L/min}$. The ozone scrubber had a denuder design and used potassium iodine (Williams and Grosjean, 1990). It protected instruments from high ozone levels but also precluded further reaction of collected samples with ozone on the filters. Collection times were around 1-2 days in order to provide sufficient amounts of carbon on the filters for isotope analysis, see Table 2.

Several instruments were used for characterization of the aerosol and gas phase composition inside the bag: a scanning mobility particle sizer, SMPS (TSI 3081 DMA and 3772 CPC, 0.0508 cm impactor) was used to measure particle size distributions (10-500 nm diameter if not stated otherwise) and a cloud condensation nuclei counter (CCNC, Droplet Measurement Technologies) gave information on the CCN properties of generated SOA (King et al., 2012). Temperature and relative humidity were measured continuously inside the bag (Hygroflex HF532, Rotronic) and read by the same software that controled the pressure inside the bag. $NO_x$ $(= NO + NO_2)$ levels were monitored using a chemiluminescence $NO_x$ analyzer (42$i$, Thermo). The same line fed an UV photometric $O_3$ analyzer (49$i$, Thermo) to monitor ozone levels.

Prior to experiments, the chamber was cleaned at $40\,^{\circ}\mathrm{C}$ by flushing with $20\,\mathrm{L/min}$ of dry, clean air and high ozone concentrations (ca. 200 ppb) until the total particle concentration was below $1\,\mathrm{cm}^{-3}$ for at least 12 h. Once the VOC injection was started, the setup was running for several days without interruption.

## 2.3 $\alpha$-pinene ozonolysis

All aerosol was generated from the dark ozonolysis of $\alpha$-pinene under low $NO_x$ conditions ($< 2\,\mathrm{ppb}$) without any seed particles present. Two types of experiments were performed distinguished by the OH-scavenger used: Experiment A: 1-butanol and experiment B: cyclohexane. The amount of $\alpha$-pinene injected using the syringe pump resulted in a steady state concentration of ca. 60 ppb inside the bag (without oxidants). Ozone concentrations during the experiment were always above 150 ppb, i.e. ozone was always in excess. The temperature was stable at $22\,^{\circ}\mathrm{C}$ and RH $< 1\,\%$. Without seed particles present, aerosol formed via new particle formation. Table 2 gives an overview of the conditions under which experiments A and B were performed.

## 2.4 Filter handling protocol

Glass vials with plastic stoppers were used to store the quartz filters for off-line analysis before and after the experiments. The glass vials themselves were cleaned in a ceramic oven at $600\,^{\circ}\mathrm{C}$ for 24 h prior to use. The quartz filters were cleaned in the same oven at $600\,^{\circ}\mathrm{C}$ for more than 20 h prior to use. Each filter was stored in a separate glass vial that was wrapped in aluminium foil and stored in a dark freezer (-30 $^{\circ}\mathrm{C}$) except when loading or during transport between Copenhagen and Utrecht. Two quartz filters (QBQ) were loaded at a time in a cleaned filter holder to account for possible sampling artefacts, such as adsorption and evaporation of organic vapours on / from the filters (Watson et al., 2009; Turpin et al., 2000). In this study the first filter, facing the sample stream, is called the *front* filter, while the second one (stacked below) is called the *back* filter. Storage time between loading and analysis was up to six months. During transport the filters stayed in the wrapped vials but



were not actively cooled. Prior to analysis the filters were cut into pieces of uniform size (0.5 and 1 cm diameter). Blank filters were treated identically to loaded filters but were not exposed to chamber air. Gloves were used whenever working directly with the filters and all tools were rinsed several times using first acetone, then ethanol.

## 2.5 Filter analysis

The filters with samples from the smog chamber experiments were analyzed for their chemical and isotopic composition at the Institute for Marine and Atmospheric research Utrecht (IMAU). Propagated uncertainties based on at least three measurements are given as 1-sigma errors.

### 2.5.1 Thermal-desorption chemical analysis of filters

The chemical analysis followed methods described earlier (Holzinger et al., 2010b; Timkovsky et al., 2015) and will only be outlined briefly here. The chemical analysis setup consisted of a two-stage oven. Filter pieces were heated up step-wise in the first oven stage to temperatures of 100, 150, 200, 250, 300 and 350 °C, while the temperature of the second oven stage was kept constant at 200 °C. SOA compounds desorbed from the filters according to their volatility and a gas flow of nitrogen (50 ml/min) carried them to the PTR-MS with time-of-flight detector (PTR-TOF8000, Ionicon Analytik GmbH, Austria) situated directly after the second oven. The PTR-MS inlet was heated to its maximum temperature of 180 °C, while the drift tube was operated at 120 °C. This declining temperature gradient from the first oven stage to the drift tube inside the PTR-MS reduced cold spots and minimised repeated sample condensation. The PTR-MS detected the desorbed compounds after protonation (addition of $^{1}$H) as ions with a mass to charge ratio $(m/z) + 1$.

The PTR-MS had a high mass resolution $(m/\Delta m \approx 4000)$ allowing detection of ions with differences in $m/z$ larger than 30 mDa. The algorithm for analyzing the PTR-MS data is based on a method reported earlier (Holzinger et al., 2010a; Holzinger, 2015). For each experiment (A or B), the ions detected on the front and back filters were combined in a unified mass list. These lists ensure that the same ions can be compared on all filters within one experiment. Ions with $m/z < 39$ Da were excluded (except 33.03 Da, methanol, and 31.01 Da, formaldehyde) as the PTR-MS mainly detects primary ions in this mass region, which do not originate from the filters. Water clusters with masses 37.026 and 55.038 Da can form in the PTR-MS and were not considered to be aerosol compounds. A total of 685 (753) ions were detected by PTR-MS in experiment A (B). In order to take the contribution from the blank filter into account, the blank filter loading was calculated from the PTR-MS signal and subtracted from the front and back filters for each ion on the unified mass list. All ion concentrations are then reported relative to smog chamber conditions, i.e. as mass of detected ion per unit volume of air in the smog chamber. Table 2 lists the blank-corrected total mass concentration, $M_{\mathrm{total}}^{\mathrm{PTR\text{-}MS}}$. Many ions had negligible concentrations. In order to streamline data analysis and to reduce noise only the 427 (451) ions from the unified mass list of experiment A (B) comprising 90 % of the total aerosol mass (counting from the ion with highest concentration downwards) were considered for further analysis.

The data analysis algorithm was also used to identify the molecular formulas of the detected ions (total number of most abundant C, H, O and N isotopes) based on their exact masses. In cases when the peak resolution did not allow unambiguous identification several candidates were suggested by the algorithm (Holzinger et al., 2010a). The suggested formulas for all





prominent peaks were checked manually on at least two filters and corrected if necessary, including when the suggested formula contained N or $^{13}$C. The former can be excluded due to the experimental low $NO_x$ conditions and the latter can be easily verified by the corresponding carbon-12 peak at $(m/z - 1) + 1$. Most of the ions were identified unambiguously and typically only ca. 5 % of the total ion mass from front filter desorption were attributed to ions with no clear molecular formula.

The oxygen-to-carbon ratios, O:C, of the ions were calculated similarly to the work of Holzinger et al. (2013) for each filter at each desorption temperature:

$$O\!:\!C = \frac{\sum w_i n_{O,i}}{\sum w_i n_{C,i}} \tag{1}$$

Here, the sum counts over all identified ions $i$, $w_i$ is the measured amount of ion $i$ in mol and $n_{C,i}$ and $n_{O,i}$ are the respective number of carbon and oxygen atoms for ion $i$ as given by its molecular formula. Equation (1) gives the ratio of oxygen to

10 carbon atoms in all identified ions.

### 2.5.2 Thermal-desorption isotope analysis of filters

The setup used to measure the carbon isotope composition of the filter samples was described in detail by Dusek et al. (2013). The isotope analysis setup also consisted of a two stage oven. The filter pieces were heated stepwise in the first oven stage to temperatures of 100, 150, 200, 250, 300, 340, 390 °C, desorbing different SOA compounds according to their volatility.

In the second oven stage, the gaseous compounds were fully combusted to $CO_2$ at 550 °C using a platinum wool catalyst. The $CO_2$ was dried and purified using two cold traps and a GC column before it was analyzed in a Delta plus XL isotope ratio mass spectrometer (Thermo, IR-MS) in continuous flow mode along with $CO_2$ laboratory standards. The measurement procedure followed a protocol where first the catalyst was charged using pure $O_2$ and then a new filter piece was placed in the oven which was subsequently flushed with helium. The heating of the filter and subsequent purification of $CO_2$ took place in

a helium carrier gas flow. In the order of analysis filter samples were bracketed by blank filter samples. IR-MS detects each isotopologue of $CO_2$ as a distinct peak with an associated peak area. The $\delta\!\left(^{13}C\right)$ value was calculated from these areas (see below). Reported data were corrected by taking the corresponding blank filter measurement into account.

Isotope data are commonly reported in delta notation, using Vienna Pee Dee Belemnite (VPDB), as an element-specific international standard for $^{13}$C:

$$\delta\!\left(^{13}C\right) = \frac{R_{sa}\!\left(^{13}C\right)}{R_{VPDB}\!\left(^{13}C\right)} - 1 \tag{2}$$

Here, $R_{sa}\!\left(^{13}C\right)$ and $R_{VPDB}\!\left(^{13}C\right)$ denote the isotope ratios ($^{13}C$ /$^{12}C$) in the sample and standard respectively.

In this study isotopic compositions of filter material are discussed relative to the isotopic composition of the initial $\alpha$-pinene, $\delta_{TC}^{\circ 1}\!\left(^{13}C\right)$, where TC denotes total carbon analysis (see below). Changes in isotopic composition are then reported as isotopic difference (Coplen, 2011):

$$\Delta\!\left(^{13}C\right) = \delta\!\left(^{13}C\right) - \delta_{TC}^{\circ 1}\!\left(^{13}C\right) \tag{3}$$

$\Delta\!\left(^{13}C\right) > 0$ indicates enrichment and $\Delta\!\left(^{13}C\right) < 0$ indicates depletion in $^{13}$C with respect to the initial $\alpha$-pinene.





### 2.6 Total carbon isotope analysis of $\alpha$-pinene and selected filters

The $\alpha$-pinene used in the smog chamber experiments and selected filters (cf. Table 2) were transferred into tin capsules, weighed and analysed for total carbon isotopic composition, $\delta_{\mathrm{TC}}\left(^{13}\mathrm{C}\right)$ in the ISOLAB of the Max-Planck-Institute for Bio-geochemistry in Jena, Germany. The analytical setup comprised an elemental analyser (EA-1100, Carlo Erba, Milano, Italy)

which was coupled to a Delta+ IR-MS (Finnigan MAT, Bremen, Germany) through a ConFlo III interface (Werner et al., 1999). The complete system was described by Brooks et al. (2003). All $\delta_{\mathrm{TC}}\left(^{13}\mathrm{C}\right)$ isotope ratios were referenced against the VPDB scale using an in-house working standard which itself is referenced against NBS 22, with a prescribed value of -30.03‰ (Coplen et al., 2006). Blank tests within each measurement sequence were used for blank correction. The analytical performance was maintained and monitored according to a measurement protocol that was described by Werner and Brand

(2001). The estimated uncertainty of the $\delta_{\mathrm{TC}}\left(^{13}\mathrm{C}\right)$ analysis was 0.11‰, based on long term performance records. The total carbon isotopic composition is reported here as $\delta_{\mathrm{TC}}^{\circ 1}\left(^{13}\mathrm{C}\right)$ for the initial $\alpha$-pinene used in the smog chamber experiments and as isotopic difference with subscript TC, $\Delta_{\mathrm{TC}}\left(^{13}\mathrm{C}\right) = \delta_{\mathrm{TC}}\left(^{13}\mathrm{C}\right) - \delta_{\mathrm{TC}}^{\circ 1}\left(^{13}\mathrm{C}\right)$, for all other samples.

### 2.7 Position-specific isotope analysis of $\alpha$-pinene

The sample preparation for NMR analysis consisted of the successive addition of $100\,\mu\mathrm{L}$ of a relaxing agent, Cr(Acac)$_3$

(Merck) at 0.1 M in the lock substance, Acetone-d$_6$ (EURISOTOP), to a 4 mL vial. Then $600\,\mu\mathrm{L}$ of pure $\alpha$-pinene was added and the mix was introduced into a 5 mm NMR tube. Quantitative $^{13}$C NMR spectra were recorded using a Bruker AVANCE III connected to a 5 mm i.d. BBFO probe tuned to the recording frequency of 100.62 MHz. The temperature of the probe was set to 303 K, without tube rotation. The acquisition conditions were those recommended in previous works (Bayle et al., 2014b; Silvestre et al., 2009) and are detailed in Sect. S3 in the supplementary information (SI). Isotope $^{13}$C/$^{12}$C ratios were calculated

from processed spectra (cf. Fig. S3 in SI) as described previously (Bayle et al., 2014b; Silvestre et al., 2009), cf. Sect. S3 in the SI. The measured position-specific isotopic compositions are given in delta-notation and denoted as $\delta_i\left(^{13}\mathrm{C}\right)$ where $i$ denotes the position of the C-atom, cf. Fig. 6. Typical accuracy of $\delta_i\left(^{13}\mathrm{C}\right)$ is 1‰.

Total carbon isotopic abundance, $\delta_{\mathrm{TC}}^{\circ j}\left(^{13}\mathrm{C}\right)$, was determined by IR-MS using an Integra2 spectrometer (Sercon Instruments, Crewe, UK) linked to a Sercon elemental analyser. Here, $j$ denotes different $\alpha$-pinene samples as listed in Table 1 (PSIA was

only performed for samples $j = 2..5$). A precision balance (Ohaus Discovery DV215CD) has been used to introduce 0.5 mg of pure $\alpha$-pinene into tin capsules (Thermo Fisher scientific), before loading them into the elemental analyser. The instrument was referenced against the VPDB scale using international reference materials NBS-22 ($\delta_{\mathrm{VPDB}}\left(^{13}\mathrm{C}\right) = -30.03‰$), SUCROSE-C6 ($\delta_{\mathrm{VPDB}}\left(^{13}\mathrm{C}\right) = -10.80‰$), and IAEA-CH-7 PEF-1 ($\delta_{\mathrm{VPDB}}\left(^{13}\mathrm{C}\right) = -32.15‰$) (IAEA, Vienna, Austria). Instrumental deviation was followed via a laboratory standard of glutamic acid. The corresponding position-specific isotopic difference is

reported as $\Delta_i\left(^{13}\mathrm{C}\right) = \delta_i\left(^{13}\mathrm{C}\right) - \delta^{\circ j}\left(^{13}\mathrm{C}\right)$.




## 3   Results and Discussion

### 3.1   Chamber and aerosol characteristics

The lifetime ($1/e$) of $\alpha$-pinene with respect to loss to ozone is $\tau_{O_3} = 40\,\text{min}$ based on an ozone concentration of 150 ppb and a second-order rate coefficient of $1.1 \times 10^{-16}\,\text{cm}^3\text{molecule}^{-1}\text{s}^{-1}$ (Witter et al., 2002). The nominal residence time of an air parcel in the chamber is $\tau_{\text{nominal}} \approx 3.4\,\text{h}$ which is exceeds the lifetime with respect to ozone loss by a factor of five. This means that more than 99 % of $\alpha$-pinene will have reacted during this time and for further evaluation and discussion it is assumed that $\alpha$-pinene ozonolysis was complete. As total mass is conserved for all isotopes, the isotopic composition of the initial ozonolysis product should be equal to the one of the initial reactant. As a consequence, the isotopic composition of generated SOA will not be influenced by the (unknown) kinetic isotope effect of the initial functionalization reaction in ozonolysis.

A detailed characterisation of the chamber aerosol can be found in Sect. S1 in the SI. In summary, the evolution of SMPS-derived size-distributions and total mass concentrations over time show that ca. one day after start of the experiment the aerosol population inside the bag is nearly constant for several days, cf. Fig. S1 and S2 in the SI. Sampling on filters was started at that point. The measured CCN activity shows good agreement between the SOA generated in this study and literature data for $\alpha$-pinene SOA generated in batch mode chambers (panel b in Fig. S2 in the SI). The integrated SMPS size distribution provides an estimate of the total SOA mass concentration in the chamber of 22 and 25 $\mu\text{g/m}^3$ for experiment A and B, respectively, although particles larger than 500 nm are not accounted for.

### 3.2   Chemical composition

#### 3.2.1   Blank filter

Very low surface loadings ($0.23\,\mu\text{g/cm}^2$) were found on the blank filter (#41), cf. Table 2. No peaks above 220 Da were detected. As the only exception, $C_3H_6OH^+$ (corresponding to protonated acetone) was found on the blank filter in concentrations above $5\,\text{ng/cm}^2$ at 100 °C while all other ions showed temperature-independent concentrations below that. High concentrations of acetone desorbed from all filters at 100 °C and are likely an artefact from cleaning. When handling the filters, some of the acetone used to clean the tweezers stayed on the filter and evaporated at the first temperature step. The total surface loadings of the front and back filters always exceeded those on the blank filter except for single ions with very low concentrations as sometimes found desorbing from back filters at temperatures other than 150 °C. Correction for the blank filter concentration then resulted in negative concentrations for these ions for the back filters. These cases were neglected in the further data analysis.

#### 3.2.2   Concentration thermograms

Figure 1 shows the sum of ion concentrations at each temperature step as measured by PTR-MS. All front filters show a similar profile with most of the mass desorbing at 150 °C. The back filters show small mass loadings compared to their front filters: The total desorbed mass detected by PTR-MS on back filters 3 and 5 only contain ca. 6 and 13 % of the masses of their respective





front filters. The large mass difference between front and back filters suggests efficient sampling of a dominant aerosol phase on front filters and some traces of sampling artefacts discussed in more detail below.

The total SOA mass concentration in the chamber derived from PTR-MS measurements was of the order of $10\,\mu g/m^3$, cf. Table 2. Overall the SMPS measured up to four times higher total mass concentrations than the PTR-MS. This difference
might be due to several reasons that are related to the individual steps of the chemical analysis, i.e. filter sampling, extraction from the filter and analysis by PTR-MS. Filter sampling losses of up to 10 % were attributed to negative sampling artefacts, i.e. evaporation from the filter, during sampling times of 24 h or longer in earlier work (Subramanian et al., 2004). Sampling losses might be even larger in this study because of the younger age of generated SOA compared to Subramanian et al. (2004), which indicates that the SOA is more volatile. During analysis incomplete desorption from the filter can lower recovery of
filter material because of too low top desorption temperatures. For comparison, the IR-MS detected a significant peak area at the final temperature step in the isotope analysis (390 °C compared to 350 °C in the chemical analysis), cf. panel c of Fig. 7. Other studies on $\beta$-pinene ozonolysis and photo-oxidation of terpenes also showed significant remaining volume fractions at desorption temperatures exceeding 400 °C (Emanuelsson et al., 2013, 2014). Oligomerization reactions on the filter might also change the aerosol composition. Finally, charring and fragmentation in the PTR-MS can additionally lower PTR-MS derived
total mass concentrations. Section S2 in the SI describes these processes in more detail, as well as other aspects relevant to PTR-MS data interpretation.

Figure 2 shows ion concentration thermograms of specific compounds desorbing from front filter 6. Table 4 complements information in Fig. 2 and Sect. S4 in the SI gives the full list of ions detected by PTR-MS from compounds desorbing from filter 6. In Fig. 2 the highest concentrations of most ions is seen at a desorption temperature of 150 °C, in agreement with Fig. 1.
Most ions are also detected at all other temperatures and often different ions show different temperature profiles. For instance, ions with masses 169.085 Da and 81.07 Da peak at nearly the same concentration at 150 °C but show different concentrations at 100 °C. Other ions, e.g. at masses 155.07 Da and 183.099 Da show an extra peak at 350 °C. In general the observation of a specific ion over a range of desorption temperatures does not indicate that a specific compound with the same mass as the ion is desorbing from the filter at these temperatures.

A single compound is expected to desorb from the filter at temperatures between its melting and boiling temperatures, if the vapour pressure above the solid is assumed to be small. This was confirmed when Dusek et al. (2013) used the same analytical setup extracting test compounds (dicarboxylic acids) from filters: No significant peak areas were detected in the IR-MS at temperatures higher than the boiling point of the respective compound. The duration of each temperature step was adjusted to ensure that all compounds had time to leave the filter before the oven ramps up the temperature again (Dusek et al., 2013).
Such a single compound should in principle be detected by the PTR-MS as an ion of similar mass and only in this temperature window.

Compared to this simplified case, the ion concentrations in Fig. 2 at temperatures above 150 °C are significantly larger than zero. Ions detected over a wide range of desorption temperatures are no indication for an aerosol compound that evaporates at temperatures beyond the compound's boiling point, as this was minimised by fine-tuning the heating durations as explained
above. Instead the detected ions are likely fragments of larger (heavier) compounds, e.g. oligomers, that break apart and the





fragments of which are detected at lower masses. Besides oligomers, complex organic mixtures could also generate such a pattern as the desorption temperature of complex mixtures is typically higher than the desorption temperature of their single constituents (Cappa et al., 2008). When an ensemble of compounds (e.g. SOA) is heated, a specific ion is detected when the desorption temperature of a compound of the same mass is reached and again when the desorption temperature of oligomers

is reached that the compound is part of. Besides fragmentation due to thermal decomposition, oligomers can also fragment in the PTR-MS during ionisation as detailed below and in Sect. S3 in the SI.

The chemical analysis of all front filters gave very similar results. For instance, the top ten ions (ranked by amount) desorbing from filter 6 at 150 °C (Panel a in Fig. 2 and Table 2) can be found in nearly the same order on filters 2,4 and 8 (not shown). However, filters 6 and 8 (experiment B) show higher concentrations compared to filters 2 and 4 (experiment A), see

also Table 2. Different aerosol yields were reported earlier when using different OH scavengers in carbohydrate ozonolysis experiments (Docherty and Ziemann, 2003; Keywood et al., 2004). The reaction of OH with different scavengers generates different products. Consequently, the $RO_2/HO_2$ ratios change depending on the scavenger used, influencing the volatility distribution of products of $\alpha$-pinene SOA. Overall, an increase of SOA yield is predicted when using cyclohexane as OH scavenger compared to 2-butanol (Jenkin, 2004). Assuming 1-butanol to behave like 2-butanol (Shilling et al., 2008), the larger desorbed

aerosol mass detected by PTR-MS in the cyclohexane experiments is consistent with these considerations.

The chemical composition of aerosol found in this study was compared to previously published chemical compositions of ambient aerosol and SOA derived from $\alpha$-pinene ozonolysis (in dry, dark, and low-$NO_x$ conditions). The references are listed in Table 3. Panel b in Fig. 2 shows concentration thermograms of ions with masses attributed to compounds reported previously, cf. Table 4. The high resolution time-of-flight PTR-MS allowed discrimination of the particulate reaction products reported

earlier (Holzinger et al., 2005, 2010a; Winterhalter et al., 2003; Jenkin, 2004; Jaoui and Kamens, 2003). In many cases the identification could be positively confirmed, but in some cases an original attribution could be falsified (details can be found in Sect. S2 in the SI). Bracketed references in Table 4 and in Sect. S4 in the SI indicate when the assigned formulae for ion masses differed from those in the literature. Compounds predicted by modelling studies are noted by their capitalised Master Chemical Mechanism name in the description field in Table 4 and Sect. S4 in the SI.

Figure 3 shows the mass spectrum of compounds that desorbed from filter 6 at 150 °C. Most ions have a mass below 250 Da, indicating no direct observation of oligomers. However, several fragmentation patterns were detected in the data as highlighted by the arrows in Fig. 3: the light green arrow connects peaks with a mass difference of 14.016 Da (corresponding to a $CH_2$ group) and the dark blue arrow connects peaks with a mass difference of 18.011 Da (water). Details on fragmentation patterns can be found in Sect. S2 in the SI. Fragmentation patterns in the mass spectra show indirectly that large compounds

like oligomers and/or complex organic mixtures were present on the filters as also reported in the literature (Docherty et al., 2005; Gao et al., 2004; Cappa et al., 2008) but that these compounds decompose during desorption or in the PTR-MS during ionization before they were detected as smaller ions. The case of acetaldehyde will serve as an example: Acetaldehyde with a molecular weight of 45.034 Da in its protonated form (counting all atoms as primary isotopes, $^{12}C$, $^{1}H$ and $^{16}O$), is one of the compounds found in this study and reported in the literature. At 100 °C the ion with assigned mass 45.033 Da (Panel b in





Fig. 2) is likely acetaldehyde, a VOC desorbing from the filter. At all higher temperatures the same ion is a typical fragment of larger complexes.

### 3.2.3 Back filters

Figure 4 compares the chemical composition of back and front filters 5 and 6. Panel a shows the ratio between the ion con-
centration on back and on front filters for each ion. Ratios close to one are observed for high desorption temperatures, such as shown for 350 °C. The lowest ratios are observed for 150 °C indicating that most ions are more abundant during front filter desorption than during back filter desorption. Ratios for specific compounds are also listed in Table 4 showing that ions desorbing from front filter 6 at 150 °C with highest concentration typically desorb with much lower concentrations from back filter 5.

Panel b in Fig. 4 compares the ranking of the detected ions (according to their concentration) on front and back filters. The rank of an ion from a filter measurement carries information about the relative abundance of the ion, independent of the total concentration on that filter. Plotting the back filter rank of an ion versus its rank on the front filter shows whether that ion contributes similarly to the overall mass on both filters. If most ions rank similarly on both filters a high correlation is expected, whereas if the ions contribute very differently to the total mass a low correlation is expected. The correlation of ranks of filters
5 and 6 give regression coefficients ($R^2$) of 0.47 (100 °C), 0.37 (150 °C) and 0.82 (350 °C), respectively.

The low correlation of the ion rankings and the low ion concentration ratios indicate that the material desorbed from the back and front filters at 150 °C differs in chemical composition. The difference in chemical composition complements the large difference in total desorbed mass (Fig. 1) and strengthens the conclusion that ions detected from front filter desorption represent aerosol compounds and that ions detected from back filter desorption represent gas-phase compounds. Gas phase compounds
adsorb on both front and back filters during sampling (positive gas-phase artefact). After a few hours of sampling, adsorption and re-evaporation are in equilibrium. The gas-phase artefact will remain constant once the equilibrium is reached, while aerosols are still accumulating over time on the front filter driving the division in concentration and chemical composition.

The correlation of the ion rankings and the ion concentration ratios highlight that the chemical composition of compounds desorbing from front and back filters is more similar at 100 °C than for compounds desorbing at 150 °C. Compounds desorbing
at 100 °C include the most volatile fraction of aerosols that is partly lost during sampling (negative aerosol artefact). Even though the negative sampling artefact decreases the difference between front and back filters in chemical composition and total mass detected at 100 °C, aerosol compounds constitute the largest fraction of the ion concentration detected at 100 °C on the front filter.

### 3.2.4 O:C ratio

Figure 5 shows the measured O:C ratio versus desorption temperature for selected filters. The O:C ratio of desorbed material increases from 0.18 to 0.25 when the desorption temperature increases from 150 to 250 °C. At higher desorption temperatures the O:C ratio levels off and stays at ca. 0.25. The O:C ratios of material desorbing from the back filters are similar to those of the front filters with averages of 0.21 (filter 3) and 0.22 (filter 5), cf. Table 2.





The observed increase in O:C ratio at increasing desorption temperatures (below 250 °C) is to be expected if functionalization yields oxygenated compounds with lowered volatility (Jimenez et al., 2009; Holzinger et al., 2010b). For material desorbing at temperatures above 250 °C, this correlation seems to break down and oligomerization is likely more important. Oligomerization reactions can be accompanied by exclusion of water (Tolocka et al., 2004) that goes undetected in the PTR-MS, lowering O:C

ratios. The desorption temperature of formed oligomers and complex mixtures is also higher than the desorption temperature of their single constituents (Cappa et al., 2008). Low O:C ratios of the constituents are therefore shifted towards higher desorption temperatures, resulting in the observed plateau of O:C ratios at high desorption temperatures.

Literature values for O:C ratios of SOA from $\alpha$-pinene ozonolysis are generally found to be somewhat higher than the values reported here: Shilling et al. (2008) investigated SOA from $\alpha$-pinene ozonolysis and report O:C ratios around 0.33

for aerosol loadings comparable to this study. Aiken et al. (2008) report O:C ratios around 0.3 for laboratory SOA from $\alpha$-pinene ozonolysis and observe in general more oxidized aerosol (and higher O:C ratios) in ambient samples. These two studies (Shilling et al., 2008; Aiken et al., 2008) used aerosol mass spectrometers to determine the O:C ratio, while Holzinger et al. (2010a) employed a PTR-MS as in the present study. They report a measured O:C ratio of 0.33-0.48 for remote ambient aerosol in the Austrian alps.

Besides the effect of the fraction of unidentified compounds on the O:C ratio, PTR-MS measurements may underestimate O:C ratios because of several factors including charring and fragmentation due to ionisation, cf. Sect. S2 in the SI. Holzinger et al. (2010a) assessed in detail how oxygen loss - common in PTR-MS measurements - lowers O:C ratios. However, this was not taken into account for data reported here.

### 3.3 Isotopic composition

#### 3.3.1 Position-specific isotope effect

Table 1 lists $\Delta_i\left(^{13}\mathrm{C}\right)$ for each C-atom of all analysed $\alpha$-pinene samples. Single sites show variations between $-6.9\%o$ and $10.5\%o$. The $\alpha$-pinene samples from manufacturers Sigma-Aldrich, Acros Organics and Merck have very similar position-specific isotope profiles. It is likely that those three manufacturers sell $\alpha$-pinene from the same origin (e.g. natural) and use similar preparation techniques. The last sample (from Alfa Aesar) has a slightly different profile which could tentatively be

explained by a different purification method. The data in Table 1 indicates that the position-specific isotope profiles of different $\alpha$-pinene samples is independent of the bulk (total carbon) enrichment, $\delta_{\mathrm{TC}}^{\circ j}\left(^{13}\mathrm{C}\right)$, of that sample: $\alpha$-pinene samples from Acros Organics ($j = 3$) and Merck ($j = 4$) have very similar $\Delta_i\left(^{13}\mathrm{C}\right)$ profiles but differ by $1.1\%o$ in their bulk value.

Unfortunately, PSIA could not be performed on the $\alpha$-pinene used in chamber experiments (it was used up and is not available from the manufacturer anymore). The $\alpha$-pinene used in chamber experiments has a bulk isotopic composition of

30 $\delta_{\mathrm{TC}}^{\circ 1}\left(^{13}\mathrm{C}\right) = (-29.96 \pm 0.08)\%o$ which differs by up to $3\%o$ from the other $\alpha$-pinene samples, cf. Table 1. Given that the bulk isotopic composition is no indicator for differing position-specific isotope profiles and assuming that the manufacturer did not change product origin or purification method, it is probable that the two $\alpha$-pinene samples from Sigma-Aldrich share similar position-specific isotope profiles. It is therefore assumed for the remainder of the discussion that the $\alpha$-pinene used for the SOA





experiments has the same position-specific isotope profile as the batch from Sigma-Aldrich on which PSIA was performed. The $\Delta_i\left({}^{13}\mathrm{C}\right)$ distribution of that sample is visualised in Fig. 6.

Some simple considerations regarding the expected isotopic composition of $\alpha$-pinene fragmentation products can be performed based on the $\Delta_i\left({}^{13}\mathrm{C}\right)$ profiles obtained from PSIA. Fragmentation of $\alpha$-pinene and subsequent expulsion of a fragment

with a certain number of C-atoms can alter the isotopic composition of both the expelled and the remaining fragment. Table 5 shows predicted enrichments/depletions if a single carbon atom or reasonable combinations of two carbon atoms are split off the parent compound, based on the assumption that such reactions run to completion. The choice of which carbon atom to expel was based on chemical reaction pathways presented by Camredon et al. (2010). In the case of expulsion of a single carbon atom ($C_2$, $C_7$ or $C_9$), the minor (potentially gaseous) expelled fragment is predicted to obtain an overall isotopic difference relative

to the initial $\alpha$-pinene, $\Delta_{\mathrm{gas}}\left({}^{13}\mathrm{C}\right)$, similar to the measured $\Delta_i\left({}^{13}\mathrm{C}\right)$ value for the carbon atom's former position as seen in Fig. 6. The remaining fragment, which would partition to the aerosol phase, is predicted to have an overall $\Delta_{\mathrm{aerosol}}\left({}^{13}\mathrm{C}\right)$ value equal to the average of the $\Delta_i\left({}^{13}\mathrm{C}\right)$ values of the remaining C atoms. Expelled C-atoms from positions with small $\Delta_i\left({}^{13}\mathrm{C}\right)$ values, e.g. $C_7$, will only have a small impact on the isotopic composition of the remaining fragment. For expulsion of $C_2$ ($C_9$), a depletion of -1.1‰ (enrichment of 0.8‰) is predicted for the aerosol fragment relative to the initial $\alpha$-pinene.

If two carbon atoms are expelled at once, the isotopic difference of the minor fragment relative to the initial $\alpha$-pinene is calculated as the average of the $\Delta_i\left({}^{13}\mathrm{C}\right)$ values of the respective two expelled positions. The gaseous fragments composed of two carbon atoms are predicted to show $\Delta_{\mathrm{gas}}\left({}^{13}\mathrm{C}\right)$ values of $-0.4$‰ and $0.3$‰ and the corresponding $\Delta_{\mathrm{aerosol}}\left({}^{13}\mathrm{C}\right)$ values for the larger fragment are $-0.4$‰ and $0.3$‰, cf. Table 5. These calculations are based on the measured position-specific enrichment for sample 2 in Table 1, but the results and conclusions drawn do not change significantly when performing similar

calculations for the other $\alpha$-pinene samples were PSIA data is available.

Figure 7, panel a, shows the isotopic composition of total carbon on the filters relative to the isotopic composition of the precursor $\alpha$-pinene. The aerosol on all front filters is enriched in carbon 13 relative to the initial $\alpha$-pinene, and the enrichment is larger for filter 4 (1.2‰) than for filters 6 (0.6‰) and 8 (0.7‰). Compounds desorbing from back filter 7, which potentially originate from the positive gas-phase artefact, are depleted by 0.8‰.

Panel b in Fig. 7 shows $\Delta\left({}^{13}\mathrm{C}\right)$ of thermally desorbed filter material as a function of desorption temperature. All fractions on the front filters show ${}^{13}\mathrm{C}$ enrichment of 0.2-2.8‰ relative to the initial compound and the most volatile fraction that desorbes at $100\,^{\circ}\mathrm{C}$ consistently shows the highest enrichment. SOA compounds desorbing at $100\,^{\circ}\mathrm{C}$ are enriched in ${}^{13}\mathrm{C}$ compared to $150\,^{\circ}\mathrm{C}$ by about 0.7 to 1.9‰, cf. Table 2. The $\Delta\left({}^{13}\mathrm{C}\right)$ values of the front filters do not change significantly with temperature at desorption temperatures above $150\,^{\circ}\mathrm{C}$. Filter 4 (experiment A) is enriched by an additional 0.2-1.3‰ compared to filters 6

and 8 (experiment B). The fraction desorbing at $150\,^{\circ}\mathrm{C}$ from back filter 3 is depleted by 0.7‰ compared to the initial $\alpha$-pinene and depleted by 1.9‰ with respect to front filter 4. The isotopic enrichment of back filter 3 at $100\,^{\circ}\mathrm{C}$ and $350\,^{\circ}\mathrm{C}$ is similar to that of the corresponding front filter 4. The very low concentrations detected on the back filters at these desorption temperatures preclude any in-depth discussion of the enrichment seen at those temperatures.

Panel c in Fig. 7 shows the volume-normalized peak areas, i.e. the peak areas detected by the IR-MS divided by the total air

volume sampled on the filter. This quantity allows comparison of the measured peak areas independent of sampling duration



and should therefore be similar for filters sampling the same experiment. Indeed, the volume-normalized peak areas of the front filters are similar. As was already shown from the PTR-MS measurements (Fig. 1 and 2), most compounds desorb from the front filters at 150 °C. The good correlation of the volume-normalized peak areas (measured by IR-MS) and the filter loadings (measured by PTR-MS) was also shown by Dusek et al. (2013). Also similar to the PTRMS results, the back filter shows

only very small peak areas over the whole temperature range. Back filters peak areas show a small peak at 150°C and slight further increase at higher temperatures. At 100°C the ratio of the peak area of the front filter to the peak area of the back filter exceeds 20. For comparison, the ratio of the peak area of the back filter over the peak area of the blank exceeds 10 at the same temperature. The smaller amount of carbon mass on the back filters result in larger uncertainty of $\Delta(^{13}\text{C})$ compared to the front filters.

$\Delta_{\text{TC}}(^{13}\text{C})$ values are very close to $\Delta(^{13}\text{C})$ values at 150 °C (Fig. 7 and Table 2). $\Delta_{\text{TC}}(^{13}\text{C})$ values represent a convolution of the volume-normalised peak area and $\Delta(^{13}\text{C})$, with a dominant contribution of $\Delta(^{13}\text{C})$ from 150 °C. Total carbon analysis however misses details like the enrichment of the most volatile mass fraction desorbing at 100 °C.

The observed enrichment of aerosol compounds desorbing from the front filters as well as the observed depletion of gas-phase compounds desorbing from the back filter are generally larger or of similar magnitude compared to the predicted position-

15 specific isotope effect analysis (Table 5). Given the strong assumptions behind the position-specific isotope effects predicted in table 5 (i.e. only one fragmentation reaction is considered out of many with potentially opposing effects on isotopes, and that reaction proceeds to completeness) the results suggest that the simple position-specific isotope effects considered here were not the leading cause for the observed isotope patterns in SOA. The hypothesis that only position-specific isotope effects based on simple fragmentation govern isotopic fractionation in formation of fresh SOA is rejected.

A more realistic set of assumptions should include consideration of incomplete reactions. However, this complicates the analysis significantly as other effects come into play, including most notably, position-specific isotopic fractionation. It has been shown previously in very simple systems (e.g. evaporation of solvents and sorption of vanillin) that each carbon position can have its own isotopic fractionation and that different positions can show normal and inverse isotope effects at the same time (Höhener et al., 2012; Julien et al., 2015). The substitution of a carbon-12 atom by $^{13}$C reduces the length of the bonds the atom

forms. Therefore, also positions that are not reaction sites show isotope effects, which have been termed non-covalent isotope effects (Wade, 1999). The bond length reduction results in an overall reduction in molecular volume and gives rise to position-specific isotope effects (Höhener et al., 2012). It is generally difficult to predict which position has which isotope effect, but it has been shown for sorption of vanillin that the volume reduction associated with isotopic substitutions at positions that carry functional groups leads to stronger position-specific isotope effects compared to positions that are part of the ring structure

and have no functional groups attached (Höhener et al., 2012). Similarly, the C-atoms in $\alpha$-pinene that are not part of the ring structure show large isotopic enrichment (Fig. 6) and potentially also have large position-specific isotope effects. However as Höhener et al. (2012) note for the case of vanillin, such effects leave the bulk isotopic composition largely unchanged, making the use of PSIA in SOA studies beyond what has been done here potentially challenging.



### 3.3.2 Isotope effect of fragmentation and functionalization

Incomplete fragmentation reactions leading to the loss of small, isotopically light compounds are more likely to cause the observed enrichment in the particle phase and depletion in the gas-phase than the position-specific isotope effects considered above. Isotope effects associated with heterogeneous fragmentation as outlined in the Introduction can explain the observations:

Expulsion of small, isotopically light fragments from the aerosol leaves the remaining aerosol enriched. However, it is unclear how gas-phase fragmentation of a compound into two fragments of equal size fractionates isotopes. In the case of one fragment being smaller than the other, however, normal kinetic fractionation is expected to deplete the expelled, small fragment in $^{13}$C due to the slightly faster reaction rate of the $^{12}$C isotopologue, similar to the heterogenous case. The main difference to the position-specific isotope effects discussed above is the extent of reaction: As long as fragmentation is not complete,

isotopic enrichment is governed by normal kinetic fractionation. If the fragmentation reaction went to completion, position-specific isotope effects would dominate. Another difference may play a role when several fragmentation reactions are taken into consideration: While position-specific isotope effects could cancel each other, the kinetic isotope effect consistently depletes smaller fragments in $^{13}$C.

Fragmentation in monoterpene ozonolysis is usually immediately followed by complete functionalization of the fragments.

In case of gas-phase fragmentation, this leads ultimately to partitioning of the larger fragment to the aerosol phase, while the smaller fragment stays in the gas phase (Chacon-Madrid and Donahue, 2011). The normal kinetic isotope effect associated with functionalization will not alter the isotopic composition if the reactions are assumed to go to completion, which is a reasonable assumption. The resulting isotopic signature is therefore governed by the fragmentation step which leads ultimately to a depleted gas phase and enriched aerosol phase, consistent with the observations in this study and with what has been

reported previously (Aggarwal and Kawamura, 2008; Kirillova et al., 2013, 2014). Fragmentation followed by functionalization is expected to be accompanied by elevated O:C ratios in both fragments as detailed in the introduction. This is in line with the observed increasing O:C ratios of desorbed material from front filters at increasing desorption temperatures below 250 °C (FIg. 5), and with O:C ratios that are generally elevated for back filter material desorbing at all temperatures (Table 2).

Large equilibrium fractionation due to gas-particle partitioning could also explain the observations given that individual

compounds might show isotope effects of $1 - 2 \permil$ due to partial volatilisation. In an ensemble of compounds, however, most individual compounds are likely to be found predominantly in one phase. Equilibrium isotope effects of single compounds are therefore diluted and can be ruled out as a leading cause behind the observed isotopic enrichment in SOA.

Fragments of oligomers make up a large fraction of compounds that desorb at temperatures above 150 °C. Stable $\Delta\left(^{13}C\right)$ values at these desorption temperatures suggest that oligomerization does not influence the isotopic composition significantly.

It further seems that oligomerization of already enriched oxidation products conserves the $\Delta\left(^{13}C\right)$ value.

The presented results allow to discuss isotope effects associated with sampling artefacts, which are generally not well known. While the reason behind the enrichment in material desorbing from front filters at 100 °C cannot be unambiguously identified, the negative sampling artefact provides a possible explanation. From the ensemble of sampled compounds, isotopically light isotopologues re-volatilise preferentially leading to an overall isotopic enrichment in compounds that are left on the filter. It





is unlikely that sampling artefacts caused the isotopic depletion of back filter material, as both positive and negative artefacts should enrich compounds on back filters. Negative sampling artefacts should give rise to isotopic enrichment similar to the case of the front filters outlined above. A positive sampling artefact should also lead to isotopic enrichment as the heavy isotopologues are more likely to partition into the condensed phase.

The present study allows linking ambient observations of isotopically enriched aerosol to SOA generated in the laboratory from $\alpha$-pinene. In some cases isotopic enrichment in ambient aerosol was attributed to photochemical ageing during long-range transport (Kirillova et al., 2013; Pavuluri et al., 2011). However, Fu et al. (2012) note that a normal kinetic isotope effect cannot explain their observation of aerosol that was even more enriched than aerosol from biomass burning. The authors noted that such aerosol occurred predominantly during daytimes of high abundance of biogenic SOA. The results in the present study suggest that fragmentation reactions have to be taken into account as they provide an additional way of enriching SOA in $^{13}$C.

## 4    Conclusions

The isotopic and chemical compositions of SOA generated from the dark ozonolysis of $\alpha$-pinene were determined using thermal-desorption techniques and related to each other.

High-resolution data retrieved by a PTR-MS allowed for detection of more than 400 ions. More than 90 % of the total desorbed mass detected (from front filters) was identified by chemical formulas, unambiguously, and discussed in the context of the current literature. Generated SOA mainly desorbed at 150 °C from the filters, but larger compounds likely formed in oligomerization reactions decomposed during extraction or ionisation. Besides the fragments from such oligomers, single constituents of complex organic mixtures formed on the filter were also detected as single ions of lower mass which show significant non-zero concentrations at desorption temperatures higher than 150 °C. The observed constant O:C ratio at desorption temperatures exceeding 250 °C also indicate fragments of larger molecules. At lower temperatures the O:C ratio increases from 0.18 to 0.25 indicating functionalization reactions during SOA formation.

Total carbon from SOA collected on front filters was enriched in $^{13}$C by $0.6 - 1.2‰$ with respect to the initial $\alpha$-pinene precursor. Total carbon adsorbed on the back filters, which supposedly originates from adsorption of gas-phase compounds to the filter was depleted by $-0.8‰$. Analysis of the isotopic composition as a function of desorption temperature showed that the isotopic composition of material desorbing at 150 °C was similar to the isotopic composition of total carbon. PSIA of $\alpha$-pinene from different manufacturers showed strong enrichments for specific C-atoms, between $-6.9‰$ and $+10.5‰$ relative to the bulk composition. When a simple fragmentation reaction is assumed to go to completion in which a moiety of two carbon atoms is split off, a potential isotopic signature on the formed SOA becomes much less pronounced. Based on the predicted position-specific isotope effect such simple fragmentation reactions are not likely to explain the observed enrichment of the aerosol compounds and depletion of the gas-phase compounds. Instead, incomplete fragmentation yielding small, isotopically light compounds is more likely to cause the observed enrichment in the particle phase and depletion in the gas-phase, independent of position-specific enrichment. Functionalization typically follows fragmentation in monoterpene ozonolysis and was shown to drive the O:C temperature profiles. However, the isotopic depletion in reaction products associated with func-





tionalization reactions does not seem to play a large role in the overall isotopic signature, most likely because the reactions were complete. While functionalization and fragmentation are known to occur simultaneously in monoterpene ozonolysis, it seems that fragmentation governs the isotopic fractionation of SOA relative to the precursor in the case of $\alpha$-pinene ozonolysis. Formation of oligomers does not seem to fractionate isotopes in $\alpha$-pinene ozonolysis. The enrichment due to fragmentation is

5 potentially important for interpreting isotopic compositions of ambient aerosol.

*Acknowledgements.* The authors thank IntraMIF and the University of Copenhagen for supporting this research. The research has received funding from the European Community's Seventh Framework Programme (FP7/2007-2013) under grant agreement number 237890. The thermogram $^{13}$C analysis was funded by The Netherlands Organisation for Scientific Research (NWO), grant number 820.01.001.





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





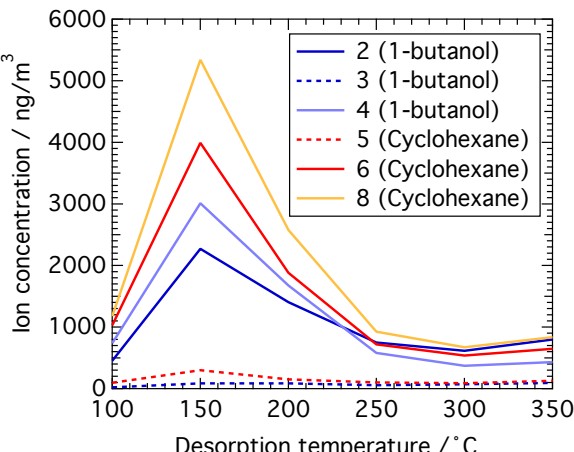

**Figure 1.** Ion concentrations of desorbed SOA filter material from $\alpha$-pinene ozonolysis as detected by PTR-MS. The concentrations of all detected ions were summed for each desorption temperature. Filter IDs are given with OH scavenger in brackets, cf. Table 2. Full lines represent front filters, dashed lines back filters.

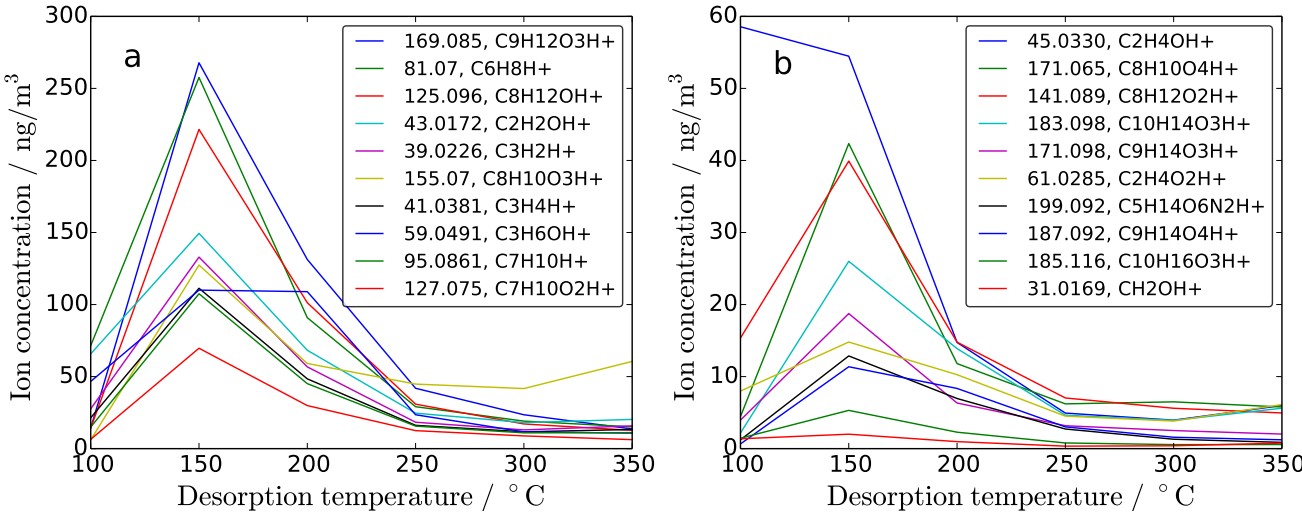

**Figure 2.** Ion concentration thermograms of compounds desorbing from filter 6 as detected by PTR-MS. Filter 6 contains SOA from $\alpha$-pinene ozonolysis in dark and dry conditions using cyclohexane as an OH scavenger. All ions are labeled by their masses in Da and their identified molecular formula. Ions with the largest concentrations are listed first. The ten ions with highest mass concentrations are plotted in Panel a. The most abundant ions found in this study with masses similar to compounds reported in the literature (Holzinger et al., 2005, 2010a; Winterhalter et al., 2003; Jenkin, 2004; Jaoui and Kamens, 2003) are plotted in Panel b. See Sect. S2 in the SI and Tables 3 and 4 for more details. Most ions show a peak at 150 °C, but their desorption pattern differs at lower and higher temperatures.

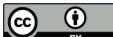


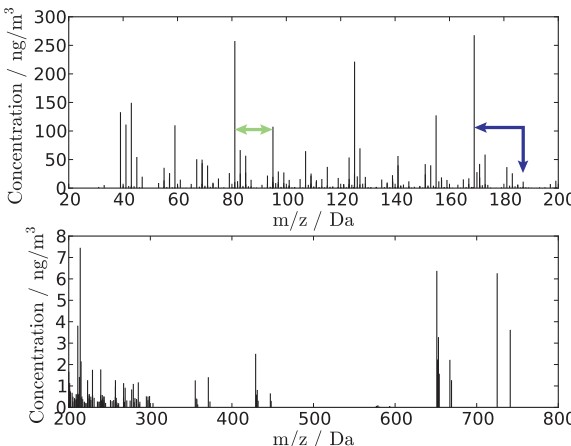

**Figure 3.** Mass spectrum of SOA from the dark ozonolysis of $\alpha$-pinene (using cyclohexane as OH scavenger) desorbed from filter 6 at 150 °C (note the difference in scale between the two panels). The arrows highlight two detected fragmentation patterns: that of a $CH_2$ group (light green arrow) and that of water (dark blue arrow), cf. Sect. S2 in the SI.

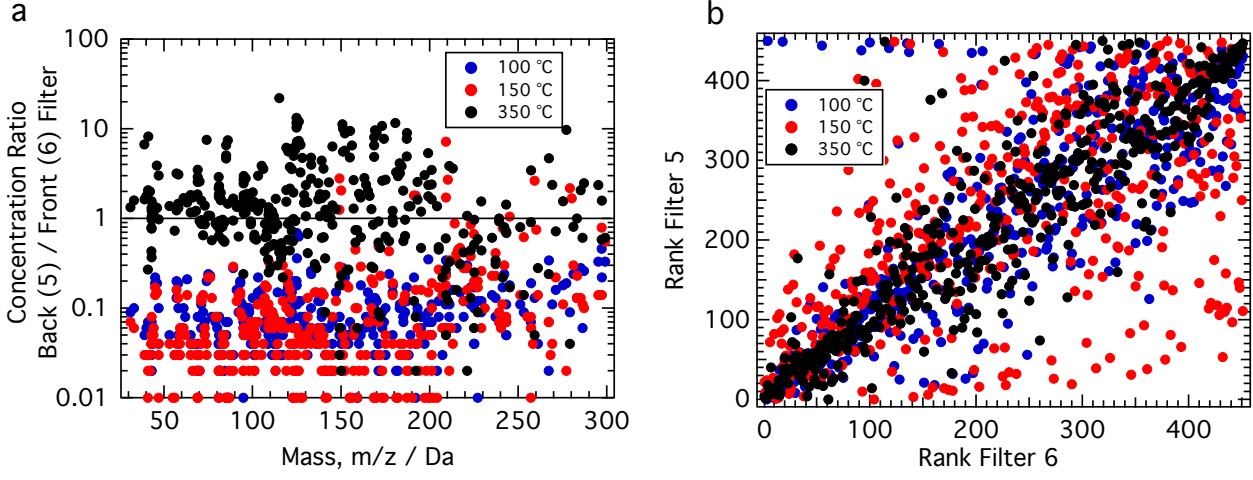

**Figure 4.** Comparison of back and front filters. Panel a shows the concentration ratio of back filter 5 to front filter 6 for three desorption temperatures. Only ions with mass below 300 Da are shown. Note the logarithmic axis. Panel b compares the relative abundance of ions from filter 5 to their relative abundance on filter 6 by correlating the concentration rankings of ions (largest concentration is ranked highest). The correlation is shown for three desorption temperatures (same color code as Panel a). Linear fits (not shown) to data in Panel b give regression coefficients ($R^2$) of 0.47 (100 °C), 0.37 (150 °C) and 0.82 (350 °C).



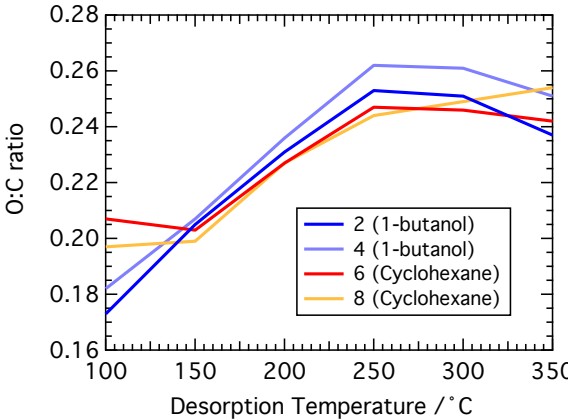

**Figure 5.** Measured oxygen to carbon ratios (O:C) of $\alpha$-pinene SOA desorbing from filters at different temperatures. Filter IDs are given with OH scavenger in brackets, cf. Table 2.

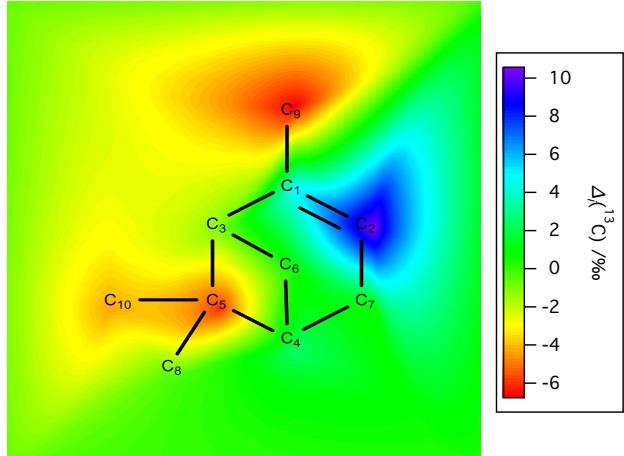

**Figure 6.** Heat map of $\Delta_i\left(^{13}\mathrm{C}\right)$ in $\alpha$-pinene (Sigma-Aldrich, lot MKBQ6213V) relative to it's bulk isotopic composition, cf. Table 1. The boundary of the image was set to $\Delta_i\left(^{13}\mathrm{C}\right) = 0$, i.e. the bulk isotopic composition. A large enrichment is visible at the double bond ($C_2$), and large depletions are visible at positions $C_5$ and $C_9$ where isotopic substitution is expected to reduce the molar volume significantly. C-atoms are numbered in order of decreasing $^{13}\mathrm{C}$ chemical shifts in the $^{13}\mathrm{C}$ NMR spectrum.



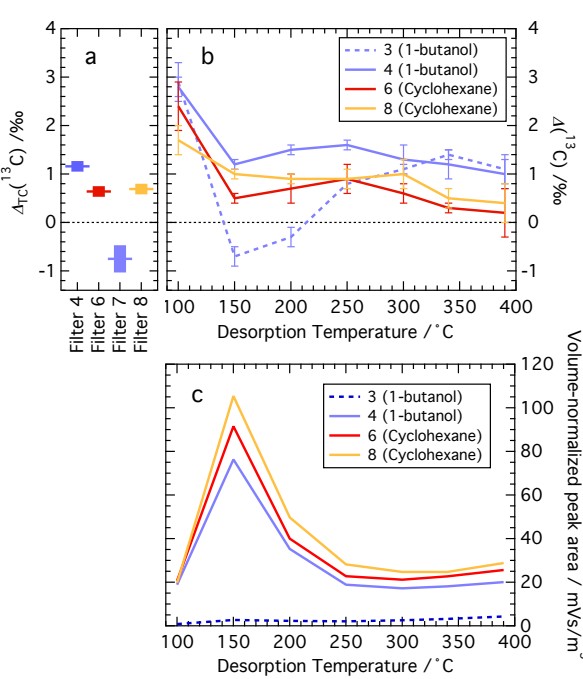

**Figure 7.** $\Delta\left(^{13}\mathrm{C}\right)$ values of alpha-pinene SOA filter samples and corresponding peak areas. Filter IDs are given with OH scavenger in brackets, cf. Table 2. Full lines represent front filters, and dashed lines back filters. Error bars denote 1-$\sigma$ standard deviations and are propagated; they vary with loading. Panel a: TC analysis of selected filters (see x-axis). Panel b: Thermal desorption analysis of filter material. Panel c: Volume-normalized (not blank corrected) peak area as function of desorption temperature for the same data as in Panel b.





**Table 1.** Manufacturer information and isotopic composition of $\alpha$-pinene samples. Position-specific isotopic fractionation is given as isotopic difference, $\Delta_i\left({}^{13}\text{C}\right) = \delta_i\left({}^{13}\text{C}\right) - \delta_{\text{TC}}^{\circ j}\left({}^{13}\text{C}\right)$ of individual C-atoms $(i)$ in $\alpha$-pinene sample $j$ with bulk isotopic composition $\delta_{\text{TC}}^{\circ j}\left({}^{13}\text{C}\right)$. Listed are the means of 5 measurements. See Fig. 6 for numbering of C-atoms. Sample 1 was used in the chamber experiments, but PSIA could not be performed as no more sample was available.

| Sample $j$ | Manufacturer | Purity /% | Code | Lot | $\delta_{\text{TC}}^{\circ j}\left({}^{13}\text{C}\right)$ / ‰ | $\Delta_i\left({}^{13}\text{C}\right)$ / ‰ | | | | | | | | |
|---|---|---|---|---|---|---|---|---|---|---|---|---|---|---|
| | | | | | | $C_1$ | $C_2$ | $C_3$ | $C_4$ | $C_5$ | $C_{6,7}$ | $C_8$ | $C_9$ | $C_{10}$ |
| 1 | Sigma-Aldrich | >99 | 268070 | 80796DJV | -30.0 | | | | | | | | | |
| 2 | Sigma-Aldrich | >99 | 268070 | MKBQ6213V | -27.7 | 4.8 | 10.5 | -1.0 | 2.4 | -6.0 | 0.6 | -0.3 | -6.7 | -4.3 |
| 3 | Acros Organics | 98 | 131261000 | A0310018 | -27.0 | 4.1 | 10.4 | -1.3 | 3.2 | -5.2 | 1.9 | -0.2 | -6.9 | -5.4 |
| 4 | Merck | >97 | 818632 | S21251 423 | -28.1 | 6.1 | 9.6 | -1.4 | 3.6 | -3.8 | -0.8 | -1.2 | -5.2 | -3.9 |
| 5 | Alfa Aesar | 98 | L04941 | 10175835 | -27.8 | 7.8 | 5.8 | -0.8 | 4.2 | 0.0 | -1.8 | -3.6 | -4.5 | -3.3 |





**Table 2.** Overview of $\alpha$-pinene ($\delta_{\text{TC}}^{\circ 1}\left(^{13}\text{C}\right) = (-29.96 \pm 0.08)\,‰$) ozonolysis experiments using 1-butanol (A) or cyclohexane (B) as OH-scavenger. Back and front filters are numbered for identification. The sampling time, $t$, and the sampled volume, $V$, are given. Isotopic data at $100\,°\text{C}$ and $150\,°\text{C}$ and for total carbon (TC) is given as $\Delta\left(^{13}\text{C}\right) = \delta\left(^{13}\text{C}\right) - \delta_{\text{TC}}^{\circ 1}\left(^{13}\text{C}\right)$. The total aerosol mass loading as detected by PTR-MS, $M_{\text{total}}^{\text{PTR-MS}}$, is listed next. The last column lists the measured O:C ratio averaged over all desorption temperatures.

| Experiment | Filter ID (type) | $t$ /h ($V$ /m$^3$) | $\Delta\left(^{13}\text{C}\right)$ /‰ | | | $M_{\text{total}}^{\text{PTR-MS}}$ / µg/m$^3$ | O:C |
| | | | $100\,°\text{C}$ | $150\,°\text{C}$ | TC | | |
|---|---|---|---|---|---|---|---|
| A[a] ($\alpha$-pinene, 1-butanol) | 1 (Back) | 16.8 (10.1) | – | – | – | – | – |
| | 2 (Front) | 16.8 (10.1) | – | – | – | 6.3 | 0.22 |
| | 3 (Back) | 47 (28.2) | 2.9±0.4 | -0.7±0.2 | – | 0.4 | 0.21 |
| | 4 (Front) | 47 (28.2) | 2.8±0.2 | 1.2±0.1 | 1.2±.1 | 6.9 | 0.23 |
| B ($\alpha$-pinene, cyclohexane) | 5 (Back) | 24.5 (14.7) | – | – | – | 0.9 | 0.22 |
| | 6 (Front) | 24.5 (14.7) | 2.4±0.5 | 0.5±0.1 | 0.6±0.1 | 8.9 | 0.23 |
| | 7 (Back) | 26.7 (16.5) | – | – | -0.8±0.3 | – | – |
| | 8 (Front) | 26.7 (16.5) | 1.7±0.3 | 1.0±0.1 | 0.7±0.1 | 11.7 | 0.23 |
| Handling blank | 41 | – | 10.0±0.8 | 8.9±0.1 | – | 0.23[b] | – |

[a] CCN data available, see Sect. S1 in the SI

[b] Surface loading in units of µg/cm$^2$

**Table 3.** Main references cited in this study for comparison of chemical composition. The identifier (last column) denotes the letter the reference corresponds to in Table 4 and Sect. S4 in the SI. Abbreviations: PTR-MS: Proton-Transfer-Reaction Mass Spectrometer, QP: Quadrupole, TD: Thermal-desorption, TOF: Time-of-Flight, LC-MS: Liquid-Chromatography Mass-Spectrometer, GC-MS: Gas-Chromatography Mass-Spectrometer, HP-LC: High-Performance Liquid-Chromatography.

| Study | Sample | Sample phase | Instrumentation / method | Identifier |
|---|---|---|---|---|
| Holzinger et al. (2005) | Ambient, forest | Gas | PTR-MS (QP) | - |
| Holzinger et al. (2010a) | Ambient, Austrian alps | Aerosol | TD-PTR-MS (TOF) | a |
| Winterhalter et al. (2003) | $\alpha$-pinene ozonolysis | Aerosol | LC-MS | b |
| Jaoui and Kamens (2003) | $\alpha$-pinene + $\beta$-pinene ozonolysis | Aerosol | GC-MS, HP-LC | c |
| Jenkin (2004) | $\alpha$-pinene ozonolysis | Gas + aerosol | Modelling | d |
| Camredon et al. (2010) | $\alpha$-pinene ozonolysis | Gas + aerosol | Modelling | - |





**Table 4.** The 20 ions with the highest concentrations detected from compounds desorbing from filter 6 (experiment A) at 150 °C, plus ten ions with nominal masses similar to protonated compounds commonly reported in the literature (below the line). Ions are sorted by their measured mass ($m/z$). Given are the ion's assigned chemical formula and its' concentration, $w$. Tthe ranking out of 451 considered compounds is given in brackets. The concentration ratio between back filter 5 and the corresponding front filter (6) is calculated for each listed compound at 150 °C. Literature references can be found in the footnotes (ambiguous associations are given in brackets, see Sect. S2 in the SI). One reference may contain several matches. Ions that are part of the same sequence of the $H_2O$-loss fragmentation pattern (cf. Sect. S2 in the SI for definition) are numbered relative to each other as $\pm n \times 18.011$ Da ($\pm n$ and the compound's ranking are given). Here a dash means that the peak was identified but not resolved. The list of all ions detected from compounds desorbing from filter 6 can be found in Sect. S4 in the SI.

| Mass / Da | Formula | $w$ / ng/m$^3$ (Rank) | Back / Front | Description | Literature | $H_2O$ number $n$ (Rank) |
|---|---|---|---|---|---|---|
| 39.0226 | C3H2H+ | 132.93 (5) | 0.03 | | | |
| 41.0381 | C3H4H+ | 111.32 (7) | 0.03 | | | |
| 43.0172 | C2H2OH+ | 149.37 (4) | 0.05 | | a | |
| 45.033 | C2H4OH+ | 54.47 (16) | 0.01 | Acetaldehyde | a | +1 (336) |
| 59.0491 | C3H6OH+ | 110.01 (8) | 0.06 | Acetone | a, c | |
| 67.0546 | C5H6H+ | 50.57 (18) | 0.04 | | | |
| 69.0337 | C4H4OH+ | 49.81 (19) | 0.01 | | | |
| 69.0699 | C5H8H+ | 43.60 (20) | 0.03 | | | |
| 81.07 | C6H8H+ | 257.57 (2) | 0.03 | Similar to #9 | | +1 (32) |
| 83.0496 | C5H6OH+ | 66.57 (11) | 0.05 | | a | +1 (62), -1 (96) |
| 85.065 | C5H8OH+ | 56.71 (14) | 0.01 | | | |
| 95.0861 | C7H10H+ | 107.50 (9) | 0.03 | Similar to #2 | | |
| 107.086 | C8H10H+ | 64.79 (12) | 0.19 | | | |
| 123.081 | C8H10OH+ | 53.69 (17) | 0.02 | | a | -1 (58) |
| 125.096 | C8H12OH+ | 221.48 (3) | 0.02 | | | |
| 127.075 | C7H10O2H+ | 69.69 (10) | 0.01 | | | |
| 141.054 | C7H8O3H+ | 56.22 (15) | 0.17 | | a | +1 (63) |
| 155.07 | C8H10O3H+ | 127.41 (6) | 0.01 | | a, (b, d) | +1 (13) |
| 169.085 | C9H12O3H+ | 267.79 (1) | 0.01 | | (c, d), a | +1 (72), -1 (22), -2 (202) |
| 173.079 | C8H12O4H+ | 58.75 (13) | 0.01 | Norpinic acid | b, d, (b) | -1 (6) |
| 31.017 | CH2OH+ | 2.01 (179) | 0.07 | Formaldehyde | a, c, d | |
| 61.0286 | C2H4O2H+ | 14.80 (59) | 0.06 | Acetic acid | a | |
| 141.089 | C8H12O2H+ | 39.91 (24) | 0.03 | 2,2-Dimethyl-cyclobutyl-1,3-diethanal | a, d, (d) | |
| 171.065 | C8H10O4H+ | 42.34 (21) | 0.03 | not norpinonic acid | a, (b, c, d) | -1 (-) |
| 171.098 | C9H14O3H+ | 18.75 (48) | 0.07 | norpinonic acid (?) | b, c, d | |
| 183.099 | C10H14O3H+ | 26.00 (35) | 0.03 | C109CO, 4-Oxopinonaldehyde | b, c, d | +1 (232), +2 (-) |
| 185.117 | C10H16O3H+ | 5.31 (114) | 0.12 | pinonic acid, OH-pinonaldehyde, PINONIC, C107OH, C109OH | (a), b, c, d | -1 (51), -2 (132) |
| 187.093 | C9H14O4H+ | 11.37 (71) | 0.01 | pinic acid, 10-OH norpinonic acid, PINIC | b, (b,) c, d | -1 (1), -2 (22), -3 (200) |
| 199.093 | C10H14O4H+ | 12.87 (66) | 0.01 | oxopinonic acid, keto-pinonic acid | b, d | -1 (27), -2 (105), +1 (380) |

[a] Holzinger et al. (2010a), [b] Winterhalter et al. (2003), [c] Jenkin (2004), [d] Jaoui and Kamens (2003)




**Table 5.** Calculated $\Delta\left(^{13}\mathrm{C}\right)$ values assuming that fragmentation yields certain C-atoms to be expelled to the gas phase and others to partition to the aerosol phase (details in the text). Calculations based on position-specific measurement results of the Sigma-Aldrich sample in Table 1. See Fig. 6 for numbering of C-atoms.

| Expelled C-atom(s) | $\Delta_{\mathrm{gas}}\left(^{13}\mathrm{C}\right)$ /‰ | $\Delta_{\mathrm{aerosol}}\left(^{13}\mathrm{C}\right)$ /‰ |
| :---: | :---: | :---: |
| $C_2$ | 10.5 | -1.1 |
| $C_7$ | 0.6 | 0.0 |
| $C_9$ | -6.7 | 0.8 |
| $C_1 + C_9$ | -1.0 | 0.3 |
| $C_2 + C_9$ | 1.9 | -0.4 |