# Peer review of "Chemical and isotopic composition of secondary organic aerosol generated by $\alpha$ -pinene ozonolysis"

_Atmospheric Chemistry and Physics, 2016_

## Short Comment (SC1) · 24 Feb 2016

I recently found this interesting publication in ACPD. In this article, the authors are citing my publications (thanks for using them). I downloaded it and looked how my publications were cited. I then realized that the authors may have misused one of my publications, "Irei et al. (2015)".

I have published a paper in 2015 in Atmospheric Environment, reporting isotopic composition of low-volatile fraction in airborne PM and fraction of m/z 44 signal in organic mass spectra (f44). The main focus was to report the limitation of f44 oxidation indicator measured by AMS. I have also published another in 2015 in Journal of Physical Chemistry A, which is the one the authors are citing. However, this paper reported a

different topic from ambient aerosol, a study of chemical reaction mechanism using the information of site-specific kinetic isotope effects, which were based on the results of laboratory studies for VOC oxidation.

After checking the discussion citing "Irei et al. (2015), I found that none of these does not seem to fit the referring sentence in the text (page 2, line 26-28), but likely the one published in 2014 in Environmental Science and Technology. Even if so, I had impression that the authors may have misinterpreted this publication; The authors seem to mean that the publication reported chemical reactions inside PM resulting in the 13C depletion in low-volatile fraction, however, I have not ever concluded so.

Rather, I meant in the paper that atmospheric oxidation of volatile organics more likely resulted in the light isotopic composition (less 13C) of low-volatile fraction. That is, the low-volatile fraction we observed was really SOA formation converted from volatile organics, but not oxidation products converted from chemicals that already stayed inside PM. These two are different processes. Unfortunately, I have not found a clue for the latter process in ambient measurements yet.

I would like to ask the authors to make corrections above in the future publication of this manuscript, if my points were correct. I apologize if I am misunderstanding the statement.

---

## Referee Comment (RC1) · Anonymous Referee #1 · 6 Apr 2016

Chemical and isotopic composition of secondary organic aerosol generated by alpha-pinene ozonolysis by Meusinger et al.

The paper reports a study for secondary organic aerosol (SOA) produced in a large continuous-flow chamber by ozonolysis of a-pinene. SOA was produced by the chemical reaction(s) of a-pinene with ozone in the gas-phase, and the produced SOA was sampled on fiber filters, and followed by analysis using a thermal desorption (TD) system coupled with a proton-transfer reaction ionization mass spectrometry (PTR-MS) for product identification/quantification of fragment ions from SOA components and with a stable isotope ratio mass spectrometry (IRMS) for their stable carbon isotope ratios (d13C). SOA substances on the filter samples were heated to six (or seven) different

temperatures in the TD apparatus to desorb the SOA components, and the evaporated substances/fragments of SOA components at each isothermal stage were analyzed by PTR-MS. Independently, d13C of the evaporated organic carbon at each stage was analyzed by the TD apparatus coupled with the IRMS. The study for isotope fractionation possibly provides valuable information to elucidate vailed production/sink processes of ambient SOA, which is one of hot topics in the subject of atmospheric science. Therefore, I think the topic is appropriate for an article of ACP.

General comments

After reading the manuscript, I had impression that points the authors argued were unclear: pieces of discussions seemed to be fragmented. I also had impression that some information and discussion (e.g., already studied C:O ratios and natural processes with isotope fractionations) were very detailed, and some (e.g., chemical reactions that lead to formation of low volatility products, calculation for predicted d13C in Table 5) were insufficient. In addition, the authors may want to make more focus on what they can conclude from the data they gained. For example, the position-specific isotope analysis (PSIA) sounded a key achievement in the abstract. However, if my understanding is correct, the PSIA was performed only on the unreacted a-pinene. I did not get what conclusion the authors were going to draw from the comparison of this information with d13C of step-wisely evaporated OCs, each of which is still possibly a complex mixture of multiple substances as indicated by more than 400 of fragment ions in the table in S4 of SI. If the authors could identify a specific product possessing enriched/depleted 13C atom at a specific position, which may have originated from a specific position of a-pinene, they should make discussion with the observed evidence (e.g., the difference in d13C between the different samples). Lack of reaction mechanism for production of SOA from ozonolysis of pinene was also a problem to follow the discussions. At last, I was impressed that the goal of this study the authors state at the end of introduction was not convincing to me. The authors may want to thoroughly revise the manuscript, including these points. The followings are specific comments.

[Figure]

Specific comments

P3l2-3: the statement "Typically..." does not sound right. As long as I know, the direction of condensation/evaporation isotope fractionation depends on substances.

P3l10: The authors may want to use a symbol of alpha instead of epsilon for the expression of fractionation factor.

P4l20-29: The objective the authors seems to be available without PSIA. The authors should state more clearly why they need PSIA. Referring to the PSIA study for isotope fractionations by NMR, Singleton et al. (1995, JACS, doi: 10.1021/ja00141a030) should be included. I recommend to reorganize this paragraph.

Section 2.1: a-pinene from different suppliers presented in Table 1 should be stated. Also, I recommend to present a figure for the chemical structure with carbon number for a-pinene here.

P5l10: the published year of the reference is missing.

P5l13: What did they feed through the flange?

Section 2.2.: I recommend to present an experimental scheme for flow chamber experiment.

P6l6: Provide the information of sampling flow rate. The information may give readers some idea of magnitude of artefact.

P6l20: The authors may want to rewrite the sentence like "Experiments were performed with two different OH scavenger: Experiment A with 1-butanol and Experiment B with cyclohexane."

Section 2.3.: The information presented here is the initial condition. It is preferable to present the information of extent of reaction of pinene because SOA's O:C and d13C possibly depends on it.

P7l13: delete "with time-of-flight detector" because PTR-MS is already defined as PTR-time of flight MS.

P7l20-12: Why did the authors combine the data from the front and back filter analysis together? Please state a reason using a sentence or so.

P8l30: Equation (3) seems to use inconsistent acronym used in Table 1. Please make those consistent.

P9l1: Present the size and the supplier's name for tin capsules used.

P9l16: replace "mix" with mixture.

P10l3: "(1/e)" does not make sense. Readers will appreciate if the authors define the lifetime more specifically.

P10l12: replace "is" with was. Also "Fig.S1 and S2 in the SI" probably means the two figures in Fig. S1. Please check.

P10l13: state more specifically what "that point" is.

P10l13-14: The sentence with "good agreement" should be discussed with numbers: do they agree within their own uncertainties or standard deviations?

Section 3.1.: The authors may want to present an example time series plot of ozone mixing ratios and pinene concentrations.

P10l19: What is "(#41)"?

P10l31: replace "contain" with contained.

P11l3-16: The discussion in this paragraph is critical. Possible problem(s) causing the discrepancy between the mass conc. by filter sample/PTR-MS and by SMPS may be narrowed down using the references the authors referred. The authors should present the recovery of filter/PTR-MS measurements here.

P11l17-24: the discussion is confusing. I do not get the point of argument in this paragraph. Additionally, the digits of masses presented are inconsistent. Please correct those.

P11l25-31: Similarly to the previous paragraph, I do not understand the point of this paragraph. The statements are kind of common sense within chemists.

P11l32: replace "significantly" with appreciably or considerably. If you use stats in your argument, significantly would be more appropriate.

P12l11-12: "The reaction of OH. . ." seems to suggest contribution of scavenger to SOA or SOA reactions. Any discussion with references?

Section 3.2.3: I am not sure how important the results of back filter analysis are. Figure 1 tells the amount in the back filters was very small relative to the front. How will the manuscript be without the section 3.2.1? If the results are necessary to justify a point of the authors' argument, they should write the point explicitly.

P16l10-19: The calculations for prediction of d13C are not described, although the heading of Table 5 state "details in the text". Please provide the way to calculate those as well as factors (i.e., epsilons) used in the calculations. In addition, according to Fig 7a, OC at 100 degree Celsius ($\sim$ 7% of TC) has the largest difference of -3 permil in d13C. Any discussion for this difference? Does this difference attribute to different isotopic composition of different chemical species? This is the part the authors should make deeper discussion, I think.

P16l2033: There are papers discussing site specific isotope fractionations. Why do the authors argue with reactive isotope fractionations using the reference of sorption isotope fractionation of vanillin?

P17l1-4: As long as I read, the authors never measured d13C of residual pinene and products in the gas-phase. The statements sound too speculative.

Section 3.3.2: The section seems to discuss possible phenomena of isotope fractionations that products the authors observed may have undergone. But many of them are

speculations. The authors may want to write the section more concisely.

Comments on Figures

The figure captions for Figure 1 and Figure 2: I do not know the difference between these figures by reading the figure captions. Please state what these are more specifically.

Figure 1: Why do the authors present the results for two back filters only? Likewise the front filters, the authors may want to present the results for all the back filters or to state a reason why not showing the rest of the results.

The figure caption for Figure 3: I don't understand the sentence "The allows highlight...". Please rewrite it.

Comments on Tables

I do not see a point presenting Table 3 because the authors do not argue with the information in this table.

Comments on Supplementary information

The authors may want to organize the SI more concisely. Please keep minimum to present extra figures and description even in SI.

---

## Referee Comment (RC2) · Anonymous Referee #2 · 15 Apr 2016

General

This is a manuscript on an interesting small flow chamber study on the oxidation of a-pinene by ozone. Its highlight is the isotope analytics.

The thermal desorption isotope analysis of filters is surely a nice and non-standrad method which should lead into new insights into VOC oxidation and SOA formation.

The paper is quite technical in a sense as it describes isotope techniques, the new chamber in detail.

What strike me is that there is a number of importants results all of which have to be put into the context of what is existing already. What is new here ? What has been

found already and is confirmed ?

I would suggest to go through the data of Table 4 and discuss each of, at least, the most important systems. The selection of compounds included into Table 4 is kind of rigorous as its elects 'the 20 ions with the highest concentrations'. There might be other important producst being identified when all the results are considered and a more detailed discussion of this finding is warranted to make the paper a more substantial contribution to what is already know on a-pinene SOA from ozonolysis. I am aware that introducing such discussion of results puts a demand on the authors but I think the paper would gain much value and much better contribute to the science related to the system studied here.

I am missing a condensed assessment of literature available which then leade to the selection of Holzinger et al., 2005, 2010a; Winterhalter et al., 2003; Jenkin, 2004; Jaoui and Kamens, 2003 for comparison as listed in Table 3.

Overall, the paper warrants publiction subject to a revision somewhere between minor and mayor.

Details

Abstract: I feel the abstract is very broad, hence should become more specific and highlight the most important quantitative findings. Especially, more substance-specific results should be given.

Experimental: The S/V of the new chamber should be given close to where the volume is mentioned.

Page 7, line 18: This is not a particularly high mass resolution. I would just state '...the PTR-MS had a mass resolution () allowing....

---

## Author Comment (AC1) · 2 Aug 2016

Reply to S. Irei (doi:10.5194/acp-2016-98)

Dear S. Irei,

Thank you very much for participating in the online discussion of our paper. Your comment is absolutely correct - we did cite the wrong paper of yours and we wish to apologize for the mistake. We meant to refer to your 2014 ES&T paper, but instead quoted the 2015 reference. Also, in order to make sure the content of your 2014 paper is cited correctly, we altered the statement to:
"Irei et al. (2014) found that SOA formation from oxidation of volatile organics lead to depletion in $^{13}$C in the low-volatile fraction of the aerosol. "
We did not mean PM-internal reactions and hope that the new formulation helps to avoid implying that.

These changes were included in the revised version that addresses also other issues raised by the reviewers. We will make sure that the final document cites your work appropriately.
Thank you for your help in making our paper better.

On behalf of the authors,
Carl Meusinger

---

## Author Comment (AC2) · 2 Aug 2016

Reply to reviewers, Chemical and isotopic composition of secondary organic aerosol generated by alpha-pinene ozonolysis by Meusinger et al.

Referee comments are written in **bold**, our answers in *italic* and the resulting changes in text are quoted in quotation marks "".

**Anonymous Referee #1**
The paper reports a study for secondary organic aerosol (SOA) produced in a large continuous-flow chamber by ozonolysis of a-pinene. SOA was produced by the chemical reaction(s) of a-pinene with ozone in the gas-phase, and the produced SOA was sampled on fiber filters, and followed by analysis using a thermal desorption (TD) system coupled with a proton-transfer reaction ionization mass spectrometry (PTR-MS) for product identification/quantification of fragment ions from SOA components and with a stable isotope ratio mass spectrometry (IRMS) for their stable carbon isotope ratios (d13C). SOA substances on the filter samples were heated to six (or seven) different temperatures in the TD apparatus to desorb the SOA components, and the evaporated substances/fragments of SOA components at each isothermal stage were analyzed by PTR-MS. Independently, d13C of the evaporated organic carbon at each stage was analyzed by the TD apparatus coupled with the IRMS. The study for isotope fractionation possibly provides valuable information to elucidate vailed production/sink processes of ambient SOA, which is one of hot topics in the subject of atmospheric science. Therefore, I think the topic is appropriate for an article of ACP.
*Thank you.*

General comments
**After reading the manuscript, I had impression that points the authors argued were unclear: pieces of discussions seemed to be fragmented.**
*The discussion was overhauled and there is now a separate section 'Discussion' to allow a clearer distinction between results and discussion. Furthermore, a reaction scheme (new FIgure 1) was added as suggested and used to guide the reader through the paper.*

**I also had impression that some information and discussion (e.g., already studied C:O ratios and natural processes with isotope fractionations) were very detailed, and some (e.g., chemical reactions that lead to formation of low volatility products, calculation for predicted d13C in Table 5) were insufficient.**
*We have shortened the former and expanded the latter. The calculations leading to the numbers shown in Table 5 are now illustrated with an example and Table 5 was overhauled. The new reaction scheme highlights the reaction pathways that lead to the calculations in Table 5.*

**In addition, the authors may want to make more focus on what they can conclude from the data they gained. For example, the position-specific isotope analysis (PSIA) sounded a key achievement in the abstract. However, if my understanding is correct, the PSIA was performed only on the unreacted a-pinene. I did not get what conclusion the authors were**

**going to draw from the comparison of this information with d13C of step-wisely evaporated OCs, each of which is still possibly a complex mixture of multiple substances as indicated by more than 400 of fragment ions in the table in S4 of SI. If the authors could identify a specific product possessing enriched/depleted 13C atom at a specific position, which may have originated from a specific position of a-pinene, they should make discussion with the observed evidence (e.g., the difference in d13C between the different samples).**

*Thank you for your comment. It is correct that PSIA was performed on the unreacted a-pinene only. PSIA on single components of the SOA mixture is not possible at this moment. The argument is that if one dominant reaction channel prefers a certain reaction site on the alpha-pinene, then the position specific isotope distribution might govern the overall isotope signature. However, we cannot fully conclude that. PSIA on a-pinene was performed for the first time and we believe that this is an important result for future research and therefore chose to leave it in the abstract. Other parts of the abstract and large parts of the introduction were rewritten. A reaction scheme was added to better guide the reader in the introduction specifically when it comes to the idea that PSIA could play a role in the 'bulk' isotopic composition. This part reads now as:*

*"The position-specific isotope composition could yield unexpected isotopic fractionation in atmospheric aerosol. For example, the C9-atom in α-pinene is found in many small, volatile ozonolysis products as it is expelled preferentially solely due to its position during fragmentation reactions (see blue squares in Fig. 1). If for example the C9-position was depleted in 13C this could lead to depletion of the gas phase not caused by kinetic fractionation."*

**Lack of reaction mechanism for production of SOA from ozonolysis of pinene was also a problem to follow the discussions.**

*Thank you for your comment. We made a new figure (Fig. 1) that identifies how the site-specific α-pinene d13C values propagate through the reaction mechanism highlighting some key arguments of our paper. The following text was added:*

*"The ozonolysis of α-pinene is often used as a test system for formation of SOA; it is fairly well studied. Figure 1 shows a reaction scheme for α-pinene ozonolysis, based on the Master Chemical Mechanism (MCMv3.1) as described by Camredon et al. (2010). In the first step ozone adds into the double bond of the molecule resulting in two branches depending on the usual Criegee mechanism. These two branches proceed by stabilisation, and subsequent fragmentation and isomerization, and subsequent reaction with RO2, HO2 and H2O to yield a wide range of oxidation products from CO, HCHO and acetone, to many larger oxidised low volatile molecules like pinic acid and pinonic acid and pinonaldehyde. The figure shows only formation of first generation products. Further reactions including dimer formation (Kristensen et al., 2016) and oligomerization reactions are not shown."*

**At last, I was impressed that the goal of this study the authors state at the end of introduction was not convincing to me.**

*The larger question is if stable carbon isotopes can be used to trace the origin of SOA and its formation mechanism. The goal was reformulated and now the last sentence of the introduction reads:*

*"The goal of this study was to provide detailed isotopic and chemical characterization of newly formed α-pinene SOA and to shed light on the mechanisms that govern isotopic fractionation in the formation of fresh SOA."*

**The authors may want to thoroughly revise the manuscript, including these points.**

*We thank the reviewer for the general comments. We adapted the manuscript according to the raised issues and thoroughly revised the manuscript. As part of the overhaul, the experiments and the filter ID's were renamed. Experiment B (former A) is the one using 1-butanol, and experiment C (former B) is the one using cyclohexane as OH scavenger. The new filter ID should make it easier for the reader to remember what as loaded initially on the filter. An explanation of the new Filter ID is given in Table 2 and in the text. It reads:*

*"Filter ID's are composed of a capital letter denoting the scavenger used, a number counting experiments using that scavenger and a small letter indicating the filter position: 'b' for back filter and 'f' for front filter."*

In case the reviewers have their own notes on the previous version of the manuscript, we give a matrix on how the filter names were re-assigned:

[Figure]

**Specific comments**

**P3l2-3: the statement "Typically. . ." does not sound right. As long as I know, the direction of condensation/evaporation isotope fractionation depends on substances.**

*The sentence was omitted.*

**P3l10: The authors may want to use a symbol of alpha instead of epsilon for the expression of fractionation factor.**

*We rewrote the section in question and now describe the Kinetic Isotope Effect (KIE) using ε values, as is common practice. We believe this makes the explanation easier to follow and more clear. The replacement of ε by α does not change the manuscript as they are simply related by ε=α−1 which is common knowledge within the field of isotope research (see for example Coplen, 2011). The new text now reads as:*

*"Kinetic fractionation results from isotope-dependent differences in reaction rates. Let 12k denote a reaction rate constant for a reaction involving a compound containing only 12C and let 13k denote the reaction rate constant for the reaction involving a single-substituted 13C isotopologue; the kinetic isotope effect (KIE) can be written as ε = 13k/12k−1. It is common to distinguish "normal" and "inverse" KIEs."*

**P4l20-29: The objective the authors seems to be available without PSIA. The authors should state more clearly why they need PSIA. Referring to the PSIA study for isotope fractionations by NMR, Singleton et al. (1995, JACS, doi: 10.1021/ja00141a030) should be included. I recommend to reorganize this paragraph.**

*We did not add this reference because it is an old article where the relative position-specific isotopic composition was determined based on elaborate calculations, but without determining delta values. Since about 10 years now, we are able to directly access the delta values using NMR. For the sake of completeness, we follow the reviewers comment and added the citation to the SI, near to the methodology references of the irm-13C NMR protocol (Tenailleau et al. 2007 and Caytan 2007).*

*The paragraph in question was rewritten (quoted above). Since the reaction scheme was added as Figure 1, the line of argument was sharpened. It should now be easier for the reader to follow our line of thought on why PSIA could play a role by following single C-atoms. We also added calculated examples in the results section, see below.*

**Section 2.1: a-pinene from different suppliers presented in Table 1 should be stated. Also, I recommend to present a figure for the chemical structure with carbon number for a-pinene here.**

*The manufacturers names were included in the text.*
*A figure showing the numbering of the C atoms was included in Table 1.*

**P5l10: the published year of the reference is missing.**

*"Viessmann" is not a reference but the name of the company that manufactured the temperature controlled room. In order to make the distinction more clear, we now write "Viessmann A/S" (full danish company name).*

**P5l13: What did they feed through the flange?**

*The flanges connect the inside of the bag with the surrounding. We decided to simplify the description and leave out the flanges. We hope that Figure S1 helps illustrating the setup. The text now reads:*

"Reactant and sampling gasses are fed through the insulating walls on opposite sides and provide numerous ports for injection and sampling."

**Section 2.2.: I recommend to present an experimental scheme for flow chamber experiment.**

*Thank you for the comment. A flow scheme was Inserted in the SI and is the new Figure S1. The Figure was referenced wherever suitable.*

**P6l6: Provide the information of sampling flow rate. The information may give readers some idea of magnitude of artefact.**

*The sampling rate is given in the sentence at the beginning of the paragraph. In order to improve the reader's flow, we rearranged the paragraph and placed the last sentence after the first. Thus, sampling time and rate are mentioned next to each other:*

"On the outlet side, the generated aerosol was sampled after an ozone scrubber on doubly stacked quartz-fibre filters (4.7 cm diameter, QMA 1851, Whatman) for offline chemical and isotope analysis at 10 L/min. Collection times were around 1-2 days in order to provide sufficient amounts of carbon on the filters for isotope analysis, see Table 2. "

**P6l20: The authors may want to rewrite the sentence like "Experiments were performed with two different OH scavenger: Experiment A with 1-butanol and Experiment B with cyclohexane."**

*Thank you for the comment. We replaced the sentence with the one you suggested, except that the experiments were renamed.*

**Section 2.3.: The information presented here is the initial condition. It is preferable to present the information of extent of reaction of pinene because SOA's O:C and d13C possibly depends on it.**

*Thank you very much. We followed your suggestion and moved the first paragraph of section 3.1 here.*

**P7l13: delete "with time-of-flight detector" because PTR-MS is already defined as PTR- time of flight MS.**

*Thank you for spotting the redundancy. We followed your recommendation and removed it.*

**P7l20-12: Why did the authors combine the data from the front and back filter analysis together? Please state a reason using a sentence or so.**

*There is a technical explanation for this as explained in the referenced paper (Holzinger et al., 2010a):* "the derived mass for a detected ion may vary a few mDa from file to file due to statistical uncertainty and the limited mass resolution of the mass spectrometer. The homogenization routine creates a "unified-peak-list"; the overall mass accuracy of the detected ions is improved."

*"Unified mass list" was replaced througout the text by "unified-peak-list". The text in question now reads:* "For each experiment (B or C), the ions detected on the front and back filters were combined in a unified-peak-list to minimise statistical uncertainty and improve overall mass accuracy (Holzinger et al., 2010a). "

**P8l30: Equation (3) seems to use inconsistent acronym used in Table 1. Please make those consistent.**

*We can't see an inconsistency here. Table 1 lists the a-pinene samples, their bulk isotopic values $d^{oj}{}_{TC}(^{13}C)$, and their position specific isotopic difference, $\Delta_i(^{13}C)$. "j" is just a counter to distinguish the different manufacturer samples. The last sentence in 2.7 gives the formula for the position specific isotopic difference, which is restated in the table's header. This formula is similar but not identical to equation 3, which states how the "bulk" isotopic difference was calculated. We choose not to use the word "bulk" as we discuss different fractions evaporating at different evaporation temperatures.*

**P9l1: Present the size and the supplier's name for tin capsules used.**

*For the bulk measurements alongside the PSIA we used capsules of size 2x5 mm, from Thermo Fisher scientific. For the bulk analysis of the other samples, we used 4x6mm capsules from Lüdi AG (Flawil, Switzerland). The corresponding text passages were added to the manuscript:*

"*The α-pinene used in the smog chamber experiments and selected filters (cf. Table 2) were transferred into tin capsules (4x6 mm capsules from Lüdi AG, Flawil, Switzerland), weighed and analysed for total carbon isotopic composition, $\delta_{TC}^{13}$ (C) in the ISOLAB of the Max-Planck-Institute for Biogeochemistry in Jena, Germany.*"

"*A precision balance (Ohaus Discovery DV215CD) has been used to introduce 0.5 mg of pure α-pinene into tin capsules (2 x 5 mm, Thermo Fisher scientific), before loading them into the elemental analyser.*"

**P9l16: replace "mix" with mixture.**

*Thank you, we followed your suggestion.*

**P10l3: "(1/e)" does not make sense. Readers will appreciate if the authors define the lifetime more specifically.**

*Thank you. We meant the e-folding time assuming a batch experiment with a single VOC injection and changed the sentence to:*

"The e-folding time of α-pinene with respect to loss to ozone is $\tau_{O_3} = (k[O3])^{-1} = 40$ min based on an ozone concentration of 150 ppb and a second-order rate coefficient of $k = 1.1 \times 10^{-16}$ cm$^3$ molecule$^{-1}$s$^{-1}$ (Witter et al., 2002)."

**P10l12: replace "is" with was.**

*Thank you, we followed your recommendation.*

**Also "Fig.S1 and S2 in the SI" probably means the two figures in Fig. S1. Please check.**

*We meant to include only figure S1 (now S2) and panel a of Figure S2 (now S3). the text was changed accordingly.*

**P10l13: state more specifically what "that point" is.**

*"That point" refers directly to the sentence before where the only time is given in this context:* "ca. one day after start of the experiment". *We modified the sentence in question to: "Sampling on filters was started ca. 24 h after start of injection of VOCs into the chamber."*

**P10l13-14: The sentence with "good agreement" should be discussed with numbers: do they agree within their own uncertainties or standard deviations?**

*We want to avoid a detailed discussion about different techniques to determine CCN activity and their associated errors as it exceeds the scope of this paper. The important point is that the SOA generated in our experiments is similar in its physicochemical properties to SOA generated by other authors.*

*Our statement was changed to:* "The measured CCN activity of SOA generated in this study resembles literature data for α-pinene SOA generated in batch mode chambers (panel b in Fig. S3 in the SI)."

**Section 3.1.: The authors may want to present an example time series plot of ozone mixing ratios and pinene concentrations.**

*Among authors we discussed providing a plot of the ozone mixing ratios but decided against it. The main reason against it being an interference of organic vapours on the ozone detection system yielding spikes of the ozone mixing ratios. This was tested and confirmed in extra experiments, but with no conclusion / suggestion on how to resolve the issue or how to correct our data for it. An example plot is given here (Experiment B):*

[Figure]

*In order to avoid an extended discussion on operation and detection mechanisms of ozone monitors, we opted to leave the plot out.*
*The pinene concentration was not measured.*

**P10l19: What is "(#41)"?**
*"#41" denotes the filter number. As all other filters, also the blank filter has an ID, and was simplified now to 'HB' for handling blank. The text in question was changed to: "Very low surface loadings (0.23µg/cm2) were found on the blank filter (HB), cf. Table 2."*

**P10l31: replace "contain" with contained.**
*Thank you, we corrected the mistake.*

**P11l3-16: The discussion in this paragraph is critical. Possible problem(s) causing the discrepancy between the mass conc. by filter sample/PTR-MS and by SMPS may be narrowed down using the references the authors referred. The authors should present the recovery of filter/PTR-MS measurements here.**
*We were contemplating earlier to include the ratio of SMPS/PTRMS total masses, but didn't follow up on it since both measurements come with severe restrictions: the cut-off diameter of the DMA and the general issues of the PTRMS detection (filter extraction, fragmentation, charring, too low maximum extraction temperature, …). We included an earlier study to comment on the typical recovery*

*of such an extraction / detection system. The section was also trimmed a bit to make our points more clear (based on the references). It now reads:*

*"An earlier study using an impactor-based thermal-desorption PTR-MS concluded that the total aerosol mass measured was typically 20 % lower than the total aerosol mass measured with an SMPS (Holzinger et al., 2010b). The authors estimated conservatively that their PTR-MS setup detected 55-80 % of the total aerosol mass. Filter sampling losses of up to 10 % were attributed to negative sampling artefacts, i.e. evaporation from the filter, during sampling times of 24 h or longer in earlier work (Subramanian et al., 2004). The maximum desorption temperature during chemical analysis was only 350 C and previous studies on β-pinene ozonolysis and photo-oxidation of terpenes also showed significant remaining volume fractions at desorption temperatures exceeding 400 C (Emanuelsson et al., 2013, 2014). Finally, charring and fragmentation in the PTR-MS can additionally lower PTR-MS derived total mass concentrations. Section S3 in the SI describes these processes in more detail, as well as other aspects relevant to PTR-MS data interpretation."*

**P11l17-24: the discussion is confusing. I do not get the point of argument in this paragraph. Additionally, the digits of masses presented are inconsistent. Please correct those.**

*Thank you. We overhauled the paragraph. The last sentence was moved and the detials on single masses omitted. The following paragraph (see next item) was removed, and the paragraph after that was completely rewritten in order to simplify the discussion. The main point was to explain the difference between detected mass and inferred compound, but we agree that this is not central to our main arguments and maybe too common.*

*The section now reads:*

*"Figure 3 shows ion concentration thermograms of specific compounds desorbing from front filter C1f. Table 4 complements information in Fig. 3 and Sect. S6 in the SI gives the full list of ions detected by PTR-MS from filter C1f. In Fig. 3 most ions 5 show the highest concentrations at a desorption temperature of 150 C, in agreement with Fig. 2, but also show significant concentrations at other temperature steps.*

*A pure compound is expected to desorb from the filter at temperatures between its melting and boiling temperatures. Dusek et al. (2013) observed this on the same analytical setup for dicarboxylic acids. Such a pure compound should in principle be detected by the PTR-MS as an ion of similar mass and only in this temperature window. There are several possible reasons, why the same ion is observed over a range of temperatures. Since fragmentation of chemical compounds can occur during thermal desorption in the oven and ionization in the PTR-MS (see Sect. S4 in the SI) a fraction of the detected ions are likely fragments of larger (heavier) compounds. This fragmentation can occur at all desorption temperatures and consequently fragments are detected over a range of temperatures. Moreover, an SOA particle usually does not consist of a single compound, but a complex mixture of compounds (Cappa et al., 2008). A specific compound in this mixture will only desorb significantly, when the melting point of the mixture is reached, which might differ from the melting point of the single compound. Specifically, the detection of small ions that are not likely to be present in the particle phase by themselves over a wide range of desorption temperatures indicates oligomers. Recent studies show that high molecular weight dimer esters contribute significantly to SOA from the ozonolysis of α-pinene (Kristensen et al., 2016). These low volatile compounds are believed to form from gas-phase reactions of the α-pinene derived Criegee Intermediate with abundant α-pinene oxidation products such as pinic acid. Decomposition of the dimer esters such as those reported in Kristensen et al. (2016) and subsequent volatilization of the carboxylic acid moieties could at least partly explain the detection of specific ions over a range of desorption temperatures."*

**P11l25-31: Similarly to the previous paragraph, I do not understand the point of this paragraph. The statements are kind of common sense within chemists.**
*The paragraph was omitted, see previous item.*

**P11l32: replace "significantly" with appreciably or considerably. If you use stats in your argument, significantly would be more appropriate.**
*This sentence was omitted, see previous items.*

**P12l11-12: "The reaction of OH. . ." seems to suggest contribution of scavenger to SOA or SOA reactions. Any discussion with references?**
*In the same paragraph that contains this sentence, we discuss our result briefly in light of other publications, e.g. Docherty and Ziemann, 2003, Keywood et al., 2004, Jenkin, 2004, Shilling et al., 2008. As we don't feel this discussion will lead to any major insights concerning the isotopic fractionation, we did choose not to deepen it further. However, in order to strengthen this point we added the following sentence:*
*"However, the most abundant reaction products were not affected."*

**Section 3.2.3: I am not sure how important the results of back filter analysis are. Figure 1 tells the amount in the back filters was very small relative to the front. How will the manuscript be without the section 3.2.1? If the results are necessary to justify a point of the authors' argument, they should write the point explicitly.**
*Thank you for your comment. We discussed this issue several times between authors. Due to your comment we decided to place the main part of this discussion in the SI. The backup filters support one of our main conclusions, namely if a-pinene is almost completely reacted, and particle phase is enriched, the gas phase must be depleted. This we see on the backup filters (which collect adsorbed gas-phase molecules). We consolidated the main results of the back filter analysis into one paragraph and now write in the main manuscript (first paragraph in Sect. 3.2.2):*
*"Figure 2 shows the sum of ion concentrations at each temperature step as measured by PTR-MS. All front filters (B1f, B2f, C1f, and C2f) show a similar profile with most of the mass desorbing at 150 C. The back filters (B1b and C1b) are used to characterize the positive sampling artifact, namely gas phase compounds that adsorb to the quartz fiber filters. Material collected on QBQ back filters can be assumed to mainly consist of adsorbed gas-phase compounds corresponding to a positive gas-phase artefact (Cheng and He, 2015). This is confirmed by the chemical analysis of back filters in this study, which differs considerably from that of the corresponding front filters, as detailed in Sect. S3 in the SI. The back filters show small mass loadings - roughly 6 and 13 % of the masses of their respective front filters. The large mass difference between front and back filters suggests efficient sampling of a dominant aerosol phase on front filters and a small positive sampling artifact. The front filters were not corrected for the sampling artifact."*

**P16l10-19: The calculations for prediction of d13C are not described, although the heading of Table 5 state "details in the text". Please provide**

**the way to calculate those as well as factors (i.e., epsilons) used in the calculations.**

*We now speak of maximum expected enrichment and added the assumption that branching ratios would not affect the isotopic distribution. We give examples of our calculation, but feel that no more details are necessary: please note that under the assumption that such an expulsion reaction ran to completion, the isotopic distribution is independent of any epsilon value, be it position-specific or not. We also found an error in our manuscript, as there are no reaction products composed of only 2 carbon atoms, but only 1 (e.g. CO) and 3 (e.g. acetone). Table 5 was updated and the text now reads:*

*"Some simple considerations regarding the maximum expected isotopic enrichment of α-pinene fragmentation products can be performed based on the $\Delta i\ 13C$ profiles obtained from PSIA. Table 5 shows predicted maximal enrichments/depletions if a single carbon atom or reasonable combinations of three carbon atoms are split off the parent compound, based on the simple assumptions that such reactions run to completion and that other competing reactions (branching indicated by arrows in Fig. 1) have no effect on the isotopic enrichment. Based on the chemical reaction pathways presented in Fig. 1, volatile reaction products such as acetone, CO, and formaldehyde can in most cases be assigned as originating from specific sites of the parent α-pinene. The minor (potentially gaseous) expelled fragment is predicted to have an overall isotopic difference relative to the initial α-pinene, $\Delta gas\ 13C$, similar to the measured $\Delta i\ 13C$ value for the carbon atom's former position as seen in Fig. 7. The larger fragment, which would partition to the aerosol phase, is predicted to have an overall $\Delta aerosol\ 13C$ value equal to the average of the $\Delta i\ 13C$ values of the remaining C atoms. For example, the pathway leading to formaldehyde in the sixth box in Fig. 1 is predicted to deplete formaldehyde by $\Delta 9\ 13C$ = 6.7‰ relative to the initial compound and leave the corresponding major fragment (denoted as 'R' in Fig. 1) enriched by $\Delta i\ 13C$ = 0.8‰. Here, [...]av(i=1–8,10) av(i=1–8,10) denotes the mean of $\Delta i\ 13C$ values for C atoms 1-8 and 10. Expelled C-atoms from positions with small $\Delta i\ 13C$ values, e.g. C7, will only have a small impact on the isotopic composition of the remaining fragment. For expulsion of C2, a depletion of -1.1 ‰ is predicted for the aerosol fragment relative to the initial α-pinene.*

*If three carbon atoms are expelled as in the case of acetone, the isotopic difference of the minor fragment relative to the initial α-pinene is calculated as the average of the $\Delta i\ 13C$ values of the respective expelled positions C8, C1, C9 or C10, C1, C9, see Fig. 1. The formation of acetone involves methyl migration of either the C8 or C10 atom. The gaseous fragments composed of three carbon atoms are predicted to show $\Delta gas\ 13C$ values of –0.7‰ and –2.0‰ and the corresponding $\Delta aerosol\ 13C$ values for the larger fragment are 0.4 ‰ and 1.0 ‰, cf. Table 5. These calculations are based on the measured position-specific enrichment for sample 2 in Table 1, but the results and conclusions drawn do not change significantly when performing similar calculations for the other α-pinene samples were PSIA data is available."*

**In addition, according to Fig 7a, OC at 100 degree Celsius (    7% of TC) has the largest difference of -3 permil in d13C. Any discussion for this difference? Does this difference attribute to different isotopic composition of different chemical species? This is the part the authors should make deeper discussion, I think.**

*We suspect the reviewer was commenting on panel b of that Figure and will give our answer accordingly. The second last paragraph of section 3 discusses the strong enrichment observed at 100 C. However, the discussion has to stay vague as no clear conclusions can be drawn from the data. Unfortunately, no species or group of species could be assigned to this enrichment. We omitted the last part of the paragraph which dealt with the back filters and added another consideration based on oligomerisation. The text was changed to:*

"The reason behind the enrichment in material desorbing from front filters at 100 C cannot be unambiguously identified. The chemical analysis did not allow to identify single compounds or groups of compounds that contributed significantly more to the total aerosol concentration at 100 C than at 150 C and therefore could lead to the observed enrichment. Isotope effects associated with sampling artefacts, which are generally not well known, provide room for speculation on how to interpret the enrichment at 100 C. During the negative sampling artefact, isotopically light isotopologues re-volatilise from the ensemble of sampled compounds preferentially leading to an overall isotopic enrichment in compounds that are left on the filter. Re-volatilisation should have it's largest effect at 100 C. Another explanation can be based on oligomerisation. Hall and Johnston (2012) observed significant evaporation of oligomers in a thermodenuder already at desorption temperatures below 100 C. The effect of oligomerisation on isotopes is not known, but if it leads to enrichment, the fragments of decomposed oligomers could be enriched and explain our observations."

**P16l2033: There are papers discussing site specific isotope fractionations. Why do the authors argue with reactive isotope fractionations using the reference of sorption isotope fractionation of vanillin?**

*There is not much literature available on the topic as the analysis technique is rather new. However, we now included the Botosoa et al., 2009b reference in the text and omitted speculations about bond length variations. The full paragraph reads now as:*

"A more realistic set of possible explanations for the observed fractionation of SOA relative to α-pinene should include isotope-dependent changes in branching ratios in the reaction mechanism (Fig. 1) and incomplete reactions. These effects complicate the analysis significantly as new factors come into play, including most notably, kinetically-derived position-dependent isotopic fractionation. It has been shown previously in simple systems (e.g. evaporation of solvents and sorption of vanillin) that each carbon position can have its own isotopic fractionation and that different positions can show normal and inverse isotope effects at the same time (Höhener et al., 2012; Julien et al., 2015). In chemical reactions, the substitution of a 12C atom by 13C will affect isomerisation and stabilisation dynamics by changing vibrational frequencies with an associated change in zero point energies. Therefore, positions that are not reaction sites can also show isotope effects, which have been termed non-covalent isotope effects (Wade, 1999), as has been observed during the chain-shortening reaction for the bioconversion of ferulic acid to vanillin (Botosoa et al., 2009b). It is generally difficult to predict which position has which isotope effect, but it has been shown that isotopic substitution in ring structures at positions that carry functional groups leads to stronger position- specific isotope effects compared to positions that have no functional groups attached (Höhener et al., 2012; Botosoa et al., 2009b). Similarly, the C-atoms in α-pinene that are not part of the ring structure

might have large position-specific isotope effects. However as Höhener et al. (2012) note for the case of vanillin, and as we also show in Table 5 in a simplified scenario, such effects leave the bulk isotopic composition largely unchanged, making the use of PSIA in SOA studies beyond what has been done here potentially challenging."

**P17l1-4: As long as I read, the authors never measured d13C of residual pinene and products in the gas-phase. The statements sound too speculative.**

*That is correct, we have not directly measured isotope ratios of gas-phase compounds. There should be no residual α-pinene left in the gas phase. The first sentence was omitted. See also next item.*

**Section 3.3.2: The section seems to discuss possible phenomena of isotope fractionations that products the authors observed may have undergone. But many of them are speculations. The authors may want to write the section more concisely.**

*We agree on the speculative nature of the section. First of all, we renamed it ("Discussion") and added an explanation in the beginning to highlight that there is no unambiguous conclusion possible. We therefore discuss a couple of options (fragmentation, partitioning, etc.) and how they would affect the isotopic composition. The section was completely rewritten. Here we only quote the new beginning of the section:*

*"SOA formation in α-pinene ozonolysis includes several chemical processes that influence the isotopic compositions of product species. The presented data does not allow unambiguous identification of the underlying effect that lead to enrichment of the aerosol phase with respect to the gas phase and this section discusses different possibilities to explain the observations."*

**Comments on Figures**
**The figure captions for Figure 1 and Figure 2: I do not know the difference between these figures by reading the figure captions. Please state what these are more specifically.**

*Thank you for raising this point. The caption and y-axis label of the first Figure now reads "Sum of ion concentrations" in order to ease understanding of the reader.*

**Figure 1: Why do the authors present the results for two back filters only? Likewise the front filters, the authors may want to present the results for all the back filters or to state a reason why not showing the rest of the results.**

*Data for filters is only presented when enough sample was available for multiple measurements. Some of the samples were 'lost' to adjusting the setup, etc. the following line was added:*

*"Data is only shown for filters when multiple measurements were performed."*

**The figure caption for Figure 3: I don't understand the sentence "The allows highlight. . .". Please rewrite it.**

*We mean that there are lobes of peaks present in the spectra and that they result from fragmentation patterns. The arrows highlight two elements in such a cascade of lobes. The text in question was changed to:*

*"The presence of lobes with specific periodicity is apparent. The arrows highlight*

*two detected fragmentation patterns that cause the periodicity: that of a CH2 group (light green arrow) and that of water (dark blue arrow), cf. Sect. S3 in the SI."*

**Comments on Tables**

**I do not see a point presenting Table 3 because the authors do not argue with the information in this table.**

*Thank you for the comment. As the second reviewer pointed out that there is information missing on how the literature was selected, we feel the two reviewers differ in their opinion on this matter. We chose to include more information on the matter of choosing these publications and keeping Table 3 in the manuscript. See mored etails in the response to the second reviewer below.*

**Comments on Supplementary information**

**The authors may want to organize the SI more concisely. Please keep minimum to present extra figures and description even in SI.**

*Thank you for your comment, the SI was completely restructured. However, in order to slim down the main document, we had to move some parts to the SI that we feel should not be left out totally (e.g. chamber design, blank filter analysis). We hope that the SI still meets the high standards of ACP and the reviewer.*

**Anonymous Referee #2**

**General**
**This is a manuscript on an interesting small flow chamber study on the oxidation of a-pinene by ozone. Its highlight is the isotope analytics.**
**The thermal desorption isotope analysis of filters is surely a nice and non-standrad method which should lead into new insights into VOC oxidation and SOA formation.**
*Thank you very much.*

**The paper is quite technical in a sense as it describes isotope techniques, the new chamber in detail.**
*Thank you. We feel that it is necessary to detail some of the techniques in the main document as they are non-standard in the field of aerosol science at the moment. On the other hand, we put some content (specifically on the chamber design and operation) in the SI in order to trim the main document.*

**What strike me is that there is a number of importants results all of which have to be put into the context of what is existing already. What is new here ?**
*Thank you for your comment, we think it is an important point that this is a source characterization study that can be used for the interpretation of ambient measurements. Some simple observation that we highlighted more clearly:*
- *SOA formation from a-pinene is a mechanism producing material with typical desorption temperatures that are relatively low (100 – 150 C)*
- *The observed PTR-MS O/C ratio is in the range of 0.18-0.25*
- *d13C of formed SOA shows enrichment in the particle phase and stays relatively constant over different desorption temperatures*

*Those points were more pronounced in the abstract and main text of the document. Some papers using the same analytical setup focussing on typical ambient spectra are unfortunately just in revision: Oyama et al., on tunnel aerosol, and Masalaite et al., on ambient aerosol collected in Lithuania. We hope that our paper provides the basis in future work to be used as a reference for SOA compounds.*

**What has been found already and is confirmed ?**
*We restructured the manuscript which now includes a general discussion section. Here, we present how different chemical processes might lead to the observed enrichment in the aerosol phase, and discuss these effect in light of other studies. The result section (former results and discussion section) stays closer to the data and compares our findings to other studies. We will not quote these extensive changes here, as they are too long to include but invite the reviewer to read the reformulated Sections 3 and 4.*

**I would suggest to go through the data of Table 4 and discuss each of, at least, the most important systems. The selection of compounds included into Table 4 is kind of rigorous as its elects 'the 20 ions with the highest concentrations'. There might be other important producst being identified when all the results are considered and a more detailed discussion of this**

**finding is warranted to make the paper a more substantial contribution to what is already know on a-pinene SOA from ozonolysis. I am aware that introducing such discussion of results puts a demand on the authors but I think the paper would gain much value and much better contribute to the science related to the system studied here.**

*Thank you for the comment. We followed your advice and included a Discussion section on its own, as described before. However, we must point out that the suggested task is very cumbersome and exceeds the scope of this manuscript (which is pretty long already). We hope that the presented work could be the basis of another paper comparing detected ions/compounds and their sources.*

*In order to better communicate our findings, we added a reaction scheme (Figure 1) and cross-referenced certain products that were found in our study, throughout the manuscript. Compounds predicted by the MCM can for example now be found in Figures 1 and 3, Table 4, and Section S6. The accompanying text in Sect. 3.2.2 reads now:*

*"Compounds predicted by modelling studies are noted by their capitalised Master Chemical Mechanism name in the description field in Table 4 and Sect. S6 in the SI, confirming the presence of several predicted species. These compounds include pinic acid (compound #72 in Sect. S6) and pinonic acid (#116 in Sect. S6) which are also shown in Fig. 1."*

*We added the following line to the caption of Figure 3:*

*"These are the same compounds as below the line in Table 4 and include those predicted by the MCM, e.g. pinic acid (187.093 Da) and pinonic acid (185.117 Da)."*

**I am missing a condensed assessment of literature available which then leade to the selection of Holzinger et al., 2005, 2010a; Winterhalter et al., 2003; Jenkin, 2004; Jaoui and Kamens, 2003 for comparison as listed in Table 3.**

*Thank you for your comment. In the second-last paragraph in section 3.2.2, we detailed some aspects of selecting literature to compare our results to. The main point was that the references presented detailed lists of detected constituents of particulate matter that allowed for direct comparison to our findings. Preference was given to such studies that also used PTR-MS and that were investigating α-pinene ozonolysis. The following sentences were added:*

*"The chosen references (cf. Table 3) preferably listed detected constituents of particulate matter to allow direct comparison with compounds found in the present study. Preference was furthermore given to such studies that also used PTR-MS and that were investigating α-pinene ozonolysis."*

**Overall, the paper warrants publication subject to a revision somewhere between minor and mayor.**

*Thank you very much. We included both minor and major changes in the manuscript and hope the reviewer agrees with us that the revised version addresses most raised points. Please also see the reply to reviewer 1 for changes in filter ID's and experiment numbering - all of which are intended to ease the experience for the reader.*

**Details**

**Abstract: I feel the abstract is very broad, hence should become more specific and highlight the most important quantitative findings. Especially, more substance-specific results should be given.**

*The abstract was rewritten. However, as most of the results from the isotope analysis are not substance-specific, it was not possible to be more specific in that sense.*

**Experimental: The S/V of the new chamber should be given close to where the volume is mentioned.**

*Of course, thank you for pointing that out. We added the following text:*
"(the volume/surface ratio is 0.275 m)"

**Page 7, line 18: This is not a particularly high mass resolution. I would just state '...the PTR-MS had a mass resolution () allowing....**

*Thank you for your comment. We changed the sentence in question to:*
*"The PTR-MS had a mass resolution of $m/\Delta m \approx 4000$ allowing detection of ions with differences in m/z larger than 30mDa."*

---

## Referee Report (RR1)

I checked the revised manuscript. The authors responded to our comments in professional manner, and attempted to reorganize the manuscript along the comments. It is intriguing to see what products the ozonolysis of alpha-pinene produces and how carbon isotopes are balanced between the gas-phase and condensed phase products. Furthermore, application of PSIA to reaction mechanism study seems to have potential to fingerprint its products in general. After reading the manuscript carefully, however, it was found that the corrections and modifications were not satisfactory and the manuscript has writing issues that need to be solved. Overall, the manuscript still needs substantial revision. The followings are some examples of writing issues I encountered.

There were many sentences that it was hard to understand. For example, in the sentence in P4L29, "If...", I didn't know what "depletion of the gas-phase" means, and I didn't get the logic why depletion of 13C at C9 position leads to it. For the sentence in P7L26, I could not understand what kind of correction the authors have made. The authors should make a message in every sentence clear as writing. There were also unnecessary sentences and paragraphs (I was impressed that those were unnecessary, but those may turn to be valuable, depending on for what the authors are going to explain using those sentences and paragraphs). For example, a paragraph in P3L9-18 defines a KIE and an epsilon value. However, any KIE and epsilon value for the pinene reaction never came out in the text. Those definitions may not be necessary, if the authors are not going to use the KIE in the interpretation. It is also not clear how the authors are going to discuss the observed isotope fractionation with functionalization, fragmentation, and oligomerization introduced in P3L27-P4L15. This paragraph impressed me at the beginning that the authors would discuss isotope fractionation at these reactions later to interpret the own data, but actually those were not discussed. If those are not available from own data or references, statements for combinational use of "isotope" and "functionalization (or oligomerization, fragmentation)" using one or two paragraphs may want to be avoided. Many of these things were piled up, and, at the end, I was not sure what I gained by reading the manuscript. The way to refer Supplementary Information, Tables, and Figures, also needs to be checked with the author's guide. Many figure captions and table headings also included unnecessary description, which can be inserted in the text. For these reasons, I regret to say, but the manuscript

does not seem to be ready to be published. Thorough revision is still needed. It may be the best to revise the manuscript with professional editors for expression in English, but this does not secure to turn the manuscript well organized. It is still the authors' responsibility that the order of explanations (sentence by sentence) and discussion (paragraph by paragraph and subsection by subsection) need to be well organized prior to the English check in order to make the flow of context smooth in the final form.

It is a minor thing, but the column containing "t/h" in Table 2 has also the information of "V/m^3". Those are different information and should be shown in different column. I also recommend the authors to show units in brackets instead of slash in this table for consistency (i.e., the slash of "micro g / m^3" is meant a division, while the slash of "V/m^3" is meant for unit.

Besides the writing issues, the followings are my responses (underlined) to the authors' comments (Italic) to my previous comments (bold). Please note that I am responding to only the authors' comments that I have opinions on. For those that I am not responding to, it is either of that I am satisfied with the changes or issues are overlapping with those I am already responding to.

General comments

**After reading the manuscript, I had impression that points the authors argued were unclear: pieces of discussions seemed to be fragmented.**

*The discussion was overhauled and there is now a separate section 'Discussion' to*
*allow a clearer distinction between results and discussion. Furthermore, a reaction scheme (new FIgure 1) was added as suggested and used to guide the reader through the paper.*

Thanks for attempting to revise along my advice, but this change is not satisfactory. I wanted the authors to focus on the important messages that the authors want to tell readers, and logically justify the arguments stepwise using own experimental results and references. The current form seems that the discussion section in the previous manuscript was just divided. The contents and arguments were still confusing. In addition, the "Results"

section in the current manuscript substantially includes authors' interpretation and discussion. It is okay to separate the "Results" and "Discussion" sections, but please give only the observed results in the "Results", and analyze and evaluate own data with references in "Discussion".

I also had impression that some information and discussion (e.g., already studied C:O ratios and natural processes with isotope fractionations) were very detailed, and some (e.g., chemical reactions that lead to formation of low volatility products, calculation for predicted d13C in Table 5) were insufficient.

*We have shortened the former and expanded the latter. The calculations leading to the numbers shown in Table 5 are now illustrated with an example and Table 5 was overhauled. The new reaction scheme highlights the reaction pathways that lead to the calculations in Table 5.*

I was a bit confused to find "example" illustration for the calculation of position specific d13C of Dgas and Daerosol in Table 5, but the authors probably meant S5 in the Supplementary Information. I understood that the calculations were based on NMR data, which are for unreacted alpha-pinene. From the revised manuscript, however, I could not retrieve important information to explain the d13C of product Dgas and Daerosol in Table 5 using the position specific d13C of unreacted alpha-pinene.

In addition, the authors may want to make more focus on what they can conclude from the data they gained. For example, the position-specific isotope analysis (PSIA) sounded a key achievement in the abstract. However, if my understanding is correct, the PSIA was performed only on the unreacted a-pinene. I did not get what conclusion the authors were going to draw from the comparison of this information with d13C of stepwisely evaporated OCs, each of which is still possibly a complex mixture of multiple substances as indicated by more than 400 of fragment ions in the table in S4 of SI. If the authors could identify a specific product possessing enriched/depleted 13C atom at a specific position, which may have

originated from a specific position of a-pinene, they should make discussion with the observed evidence (e.g., the difference in d13C between the different samples).

*Thank you for your comment. It is correct that PSIA was performed on the unreacted a-pinene only. PSIA on single components of the SOA mixture is not possible at this moment. The argument is that if one dominant reaction channel prefers a certain reaction site on the alpha-pinene, then the position specific isotope distribution might govern the overall isotope signature. However, we cannot fully conclude that. PSIA on a-pinene was performed for the first time and we believe that this is an important result for future research and therefore chose to leave it in the abstract. Other parts of the abstract and large parts of the introduction were rewritten. A reaction scheme was added to better guide the reader in the introduction specifically when it comes to the idea that PSIA could play a role in the 'bulk' isotopic composition. This part reads now as:*

*"The position-specific isotope composition could yield unexpected isotopic fractionation in atmospheric aerosol. For example, the C9-atom in a-pinene is found in many small, volatile ozonolysis products as it is expelled preferentially solely due to its position during fragmentation reactions (see blue squares in Fig.1). If for example the C9-position was depleted in 13C this could lead to depletion of the gas phase not caused by kinetic fractionation."*

Biased 13C distribution in intra molecular carbons of pinene is interesting, and I am not going to be offensive against this finding. However, again, my point is that it is not clear how this PSIA is related to the product analysis presented in this paper. It is reasonable that the authors will combine this result with results of future compound-specific product study to explore isotope balance between the products, but those are not available at this moment. The presentation of the result here may be justified if Dgas and Daerosol in Table 5 were calculated based on the measurement results of PSIA with reasonable assumptions. However, the assumptions (i.e., "maximum expected enrichment" and "branching ratios do not influence the isotopic composition of products") do not sound. According to Hoefs (1997), the initial isotope ratio, the degree of processing, and the fractionation factor are needed to calculate the isotope ratio of a processing substance or product at any arbitrary time. Presumably, the authors assumed ~ 100% of degree of

alpha-pinene oxidation ("completion" in the text probably suggests this), however, this does not secure that the complex subsequent reactions also completed, depending on relative reaction speeds (relative to the speed of ozonolysis of alpha-pinene) of subsequent reactions leading to the production of formaldehyde, acetone, CO, and other products. That is, branching reactions significantly influence product isotopic composition, unless otherwise there is no isotope fractionation (i.e., KIE = 0 permil) at all of these reactions. If even only one of these branching reactions had a significant KIE, which seems more reasonable assumption according to Fisseha et al. (2009), this KIE will significantly influence the isotopic composition of remaining alpha-pinene or intermediate(s) as reactions proceeded. This varying isotopic composition of precursors will then influence the isotopic composition of products from other branching reaction channels because all or a part of products originate from the same precursor.

In the text, it is not clear what substances are defined as "aesosol" and "gas-phase", what number of KIEs and branching ratios were used for the calculation. I am not sure what "the maximum expected enrichment" is, but if the authors assume zero of KIEs at all branching reactions and no branching reactions, I disagree with the assumptions. From the text I could not retrieve those information.

**Lack of reaction mechanism for production of SOA from ozonolysis of pinene was also a problem to follow the discussions.**

*Thank you for your comment. We made a new figure (Fig. 1) that identifies how the site-specific α-pinene d13C values propagate through the reaction mechanism highlighting some key arguments of our paper. The following text was added:*

*"The ozonolysis of α-pinene is often used as a test system for formation of SOA; it is fairly well studied. Figure 1 shows a reaction scheme for α-pinene ozonolysis, based on the Master Chemical Mechanism (MCMv3.1) as described by Camredon et al. (2010). In the first step ozone adds into the double bond of the molecule resulting in two branches depending on the usual Criegee mechanism. These two branches proceed by stabilisation, and subsequent fragmentation and isomerization, and subsequent reaction with RO2, HO2 and H2O to yield a wide range of oxidation products from CO,*

*HCHO and acetone, to many larger oxidised low volatile molecules like pinic acid and pinonic acid and pinonaldehyde. The figure shows only formation of first generation products. Further reactions including dimer formation (Kristensen et al., 2016) and oligomerization reactions are not shown."*

Thanks for providing the reaction scheme. This will definitely make readers easier to follow the discussion. However, the scheme is incomplete and not reader-friendly to follow the reaction steps producing high and low volatility products. For example, taking a look at the production mechanism of pinonic acid and pinaldehyde in Fig. 1, I don't know what was changed after the second processing step, "stabilization". I see that the red star mark on the carbon disappeared after the step, but what does this mean? I also don't get how "H2O" was involved in the subsequent reaction after the "stabilization". The same problem for the reaction with RO2 and HO2. The authors may want to check how other papers are presenting those reaction schemes. Second point is that the figure also does not show important elemental steps for the production of small products shown in the boxes in Fig. 1. The hidden reaction channels seem to involve reactions with C atoms, thus, must be key steps to explain the results of d13C calculations. Third point is that it is helpful for readers to see branching ratios in the figure.

Specific comments

**P3l10: The authors may want to use a symbol of alpha instead of epsilon for the expression of fractionation factor.**

*We rewrote the section in question and now describe the Kinetic Isotope Effect (KIE) using $\varepsilon$ values, as is common practice. We believe this makes the explanation easier to follow and more clear. The replacement of $\varepsilon$ by $\alpha$ does not change the manuscript as they are simply related by $\varepsilon = \alpha - 1$ which is common knowledge within the field of isotope research (see for example Coplen, 2011). The new text now reads as:*

*"Kinetic fractionation results from isotope-dependent differences in reaction rates. Let 12k denote a reaction rate constant for a reaction involving a compound containing only 12C and let 13k denote the reaction rate constant for the reaction involving a single-substituted 13C isotopologue; the kinetic isotope effect (KIE) can be written as $\varepsilon = 13k/12k - 1$. It is common to*

*distinguish "normal" and "inverse" KIEs."*

Regarding the definition of kinetic isotope effects, Colpen (2011) defines a kinetic isotope effect as 12k/13k, while Hoefs (1997) defines a fractionation factor as 13k/12k. The authors may want to stick to either of these.

**Section 3.1.: The authors may want to present an example time series plot of ozone mixing ratios and pinene concentrations.**

*Among authors we discussed providing a plot of the ozone mixing ratios but decided against it. The main reason against it being an interference of organic vapours on the ozone detection system yielding spikes of the ozone mixing ratios. This was tested and confirmed in extra experiments, but with no conclusion / suggestion on how to resolve the issue or how to correct our data for it. An example plot is given here (Experiment B):*

*In order to avoid an extended discussion on operation and detection mechanisms of ozone monitors, we opted to leave the plot out. The pinene concentration was not measured.*

It is very important to have information of pinene concentration change over time at least. Is it possible to present such info based on MCM calculation with given experimental condition?

**In addition, according to Fig 7a, OC at 100 degree Celsius (□ 7% of TC) has the largest difference of -3 permil in d13C. Any discussion for this difference? Does this difference attribute to different isotopic composition of different chemical species? This is the part the authors should make deeper discussion, I think.**

*We suspect the reviewer was commenting on panel b of that Figure and will give our answer accordingly. The second last paragraph of section 3 discusses the strong enrichment observed at 100 C. However, the discussion has to stay vague as no clear conclusions can be drawn from the data. Unfortunately, no species or group of species could be assigned to this enrichment. We omitted the last part of the paragraph which dealt with the back filters and added another consideration based on oligomerisation. The text was changed to:*

*"The reason behind the enrichment in material desorbing from front filters at 100· C cannot be unambiguously identified. The chemical analysis did not allow to identify single compounds or groups of compounds that contributed significantly more to the total aerosol concentration at 100 · C than at 150 · C and therefore could lead to the observed enrichment. Isotope effects associated with sampling artefacts, which are generally not well known, provide room for speculation on how to interpret the enrichment at 100 · C. During the negative sampling artefact, isotopically light isotopologues re-volatilise from the ensemble of sampled compounds preferentially leading to an overall isotopic enrichment in compounds that are left on the filter. Re-volatilisation should have it's largest effect at 100 · C. Another explanation can be based on oligomerisation. Hall and Johnston (2012) observed significant evaporation of oligomers in a thermodenuder already at desorption temperatures below 100· C. The effect of oligomerisation on isotopes is not known, but if it leads to enrichment, the fragments of decomposed oligomers could be enriched and explain our observations."*

Sorry for confusion. It is correct that I was pointing on the panel b. I think it is not necessary to deeply argue with speculation, if there is no reasonable scientific explanation for the difference. However, it is important to point out the difference explicitly and briefly state possible reasons for the difference so that others can hint for future studies to progress the science.

P16l20-33: There are papers discussing site specific isotope fractionations. Why do the authors argue with reactive isotope fractionations using the reference of sorption isotope fractionation of vanillin?

*There is not much literature available on the topic as the analysis technique is rather new. However, we now included the Botosoa et al., 2009b reference in the text and omitted speculations about bond length variations. The full paragraph reads now as:*

*"A more realistic set of possible explanations for the observed fractionation of SOA relative to α-pinene should include isotope-dependent changes in branching ratios in the reaction mechanism (Fig. 1) and incomplete reactions. These effects complicate the analysis significantly as new factors come into play, including most notably, kinetically-derived position-dependent isotopic fractionation. It has been shown previously in simple systems (e.g.*

*evaporation of solvents and sorption of vanillin) that each carbon position can have its own isotopic fractionation and that different positions can show normal and inverse isotope effects at the same time (Höhener et al., 2012; Julien et al., 2015). In chemical reactions, the substitution of a 12C atom by 13C will affect isomerisation and stabilisation dynamics by changing vibrational frequencies with an associated change in zero point energies. Therefore, positions that are not reaction sites can also show isotope effects, which have been termed non-covalent isotope effects (Wade, 1999), as has been observed during the chain-shortening reaction for the bioconversion of ferulic acid to vanillin (Botosoa et al., 2009b). It is generally difficult to predict which position has which isotope effect, but it has been shown that isotopic substitution in ring structures at positions that carry functional groups leads to stronger position- specific isotope effects compared to positions that have no functional groups attached (Höhener et al., 2012; Botosoa et al., 2009b). Similarly, the C-atoms in α-pinene that are not part of the ring structure might have large position-specific isotope effects. However as Höhener et al. (2012) note for the case of vanillin, and as we also show in Table 5 in a simplified scenario, such effects leave the bulk isotopic composition largely unchanged, making the use of PSIA in SOA studies beyond what has been done here potentially challenging."*

I am not sure if we are misunderstanding our comments each other, but I can find references for position specific carbon isotope fractionation at different types of chemical reactions (not for physical phenomena), such as Singleton and Szymanski (1999, JACS) and etc. Because the authors are discussing "carbon isotope fractionation" there, it is not necessary to compare only with the results obtained by NMR measurements. Carbon KIEs for elemental reactions obtained by different techniques have been studied for long, some of those may be comparable. Of course, the current reference is also okay, I think.

References
Colpen, T.B. Rapid Commun. Mass Spectrom. 25, 2538-2560, 2011.

Fisseha et al. JGR Atmos. 114, D02304, doi:10.1029/2008JD011326, 2009.

Hoefs, J., "Stable Isotope Geochemistry", Springer, 1997.

---

## Referee Report (RR2)

This is a report of the 3rd round. In this report, my comments at the 1st, 2nd, 3rd round is shown as "R1", "R2", and "R3". Similarly, the authors' replies to my comments in the 1st and 2nd communications are shown as "A1" and "A2".

I checked the manuscript, but I still think of that the revision is not satisfactory. The main reasons are that the authors have not clearly explain how the predicted isotopic compositions of "gas phase products" and "aerosol products" were calculated using the results of PSIA and the simple assumptions the authors have made do not sound. In this paper, the calculation for the isotopic compositions of gas and condensed phase products using the results of PSIA for the unreacted alpha-pinene seems the novelty. The approach itself is interesting, if the calculation makes sense. Unfortunately, the current information given in the manuscript was insufficient for readers to evaluate and follow the method, thus, the reviewer still has a question how the results of PSIA are connected to the results of thermal-desorption/PTR-MS analysis. If the connection between those logically makes sense, the contents should be published in a paper, but the current version of manuscript and the replies from the authors did not convince me.

Before going into the third report in detail, I want to explain my thought on how KIEs at reactions influence the isotopic compositions of reactants and products.

It is generally known that a KIE is reaction-specific. A KIE for a unidirectional reaction of substance A (e.g., functionalization reactions by oxidants) can be determined experimentally, regardless of the intra-molecular 13C distribution in the reactant A.

$$A \xrightarrow[\text{Rate}]{\text{KIE}} B \qquad \textbf{Rxn1}$$

The carbon isotopic composition of the reactant A and the single product B can be predicted using the Rayleigh function with the extent of reaction, the initial isotopic composition of A, and the carbon KIE for this reaction. This does not require the understanding the intra-13C distribution in the reactant A. It is, however, worthwhile to note that the magnitude of KIE will depend on the intra-molecular 13C distribution. This is because the KIE depends on

the probability of primary 13C involvement in the reaction. For this reason, the understanding the reaction mechanism and intra-molecular position of 13C will be important upon the evaluation of KIE, but not the calculation of isotopic composition using the Rayleigh function.

When the single unidirectional reaction produces multiple products (i.e., fragmentation likewise Rxn2 shown below), the story is different.

$$A \xrightarrow[\text{Rate}]{\text{KIE}} B + C$$  **Rxn2**

The Rayleigh function allows to calculate the isotopic compositions of A and B+C, but not of individual product. To predict the isotopic composition of individual product, the intra-molecular 13C distribution of A and the fragmentation mechanism are also needed in addition to the parameters and valuable referred earlier because the chance that bond-dissociation involves a 13C atom will depend on the intra-molecular distribution of 13C atoms, and this may cause the biased distribution of 13C between the product B and C, which cannot be determined by the Rayleigh function only.

The prediction would be complex in case of branching reactions. In the simple branching reaction shown in Rxn3, each pathway is assumed to produce a single product. Even for such a simple branching reaction the variations of isotopic compositions for A, B, and C can be complex because the KIEs and rates for the two pathways influence the isotopic composition of the parent reactant A, consequently the isotopic compositions of product B and C as well.

[Figure]

**Rxn3**

One can easily imagine that the prediction of isotopic composition will be more complex when each of the branching reactions produces multiple products. Furthermore, the prediction of isotopic composition for products is more difficult when oligomerization of these fragmented products takes place subsequently.

With consideration of these effects, I re-read the manuscript. The story does not seem to be so simple as the authors assume, but the attempt using the

combination of PTR-MS and isotope technique is interesting. The reviewer's third responses (R3) to the authors' second responses (A2) are written below.

R2: There were many sentences that it was hard to understand. For example, in the sentence in P4L29, "If…", I didn't know what "depletion of the gas phase" means, and I didn't get the logic why depletion of 13C at C9 position leads to it.

*A2: We have rewritten the passage to be more clear: 'It is widely understood that the enrichment/depletion of a product depends on the enrichment of the starting material, the isotopic fractionation occurring in the mechanism of its formation, and the extent of reaction. Using PSIA we can take this analysis one step further: the enrichment of a product will depend on the positiondependent enrichments of the atoms from which it is formed. For example the ozonolysis mechanism transfers the C9-atom in a-pinene into several small, volatile products (see blue squares in Fig. 1). If the C9-position was depleted in 13C the gas phase products containing this atom would be depleted, and the SOA correspondingly enriched, if the position dependent effect was stronger than kinetic isotope effects.'*

R3: This seems to explain the case of Rxn2 referred above, but what is not clear in the revised sentences is the statement, "if the position dependent effect was stronger than kinetic isotope effects." It is because the case the authors described (i.e., 13C depletion in one of the products and 13C enrichment in the other product) would be true regardless of the magnitude of position dependent (or kinetic isotope) effect. The authors need to rewrite this part.

R2: For the sentence in P7L26, I could not understand what kind of correction the authors have made. The authors should make a message in every sentence clear as writing.

*A2: This passage describes the process of assigning chemical formulas and controlling / checking the assigned chemical formulas. No 'correction' was described or made in this passage. We are sorry to say that we are not sure how to rewrite the passage to improve clarity.*

R3: I re-read the sentences there, and I likely misunderstood the explanation. I agree with this authors' response.

R2: There were also unnecessary sentences and paragraphs (I was impressed that those were unnecessary, but those may turn to be valuable, depending on for what the authors are going to explain using those sentences and paragraphs). For example, a paragraph in P3L9-18 defines a KIE and an epsilon value. However, any KIE and epsilon value for the pinene reaction never came out in the text. Those definitions may not be necessary, if the authors are not going to use the KIE in the interpretation.

*A2: The passage in question was completely rewritten:*

*'Isotopic substitution can cause reactions to be faster or slower than for the unsubstituted case, kinetically fractionating the isotopes and leading to isotopic enrichment or depletion in the products. This is known as the kinetic isotope effect (KIE). If a reaction leads to a single product, the product may initially have a different abundance, but due to the law of mass balance will achieve the same abundance as the reagent as the reaction goes to completion. If a reaction has multiple product channels, enrichment or depletion will occur if there are isotope dependent changes in the product branching ratios.'*

R3: The described cases are Rxn1-2, but the actual case shown in Fig. 1 is likely the combination of Rxn 2 and 3. The authors should describe such possibility and their influence too. The revised sentences already described are also better spelled out. Feel free to use the reaction schemes likewise Rxn1-3 in their clearer explanation, if necessary.

R2: It is also not clear how the authors are going to discuss the observed isotope fractionation with functionalization, fragmentation, and oligomerization introduced in P3L27-P4L15. This paragraph impressed me at the beginning that the authors would discuss isotope fractionation at these reactions later to interpret the own data, but actually those were not discussed. If those are not available from own data or references, statements for combinational use of "isotope" and "functionalization (or oligomerization, fragmentation)" using one or two paragraphs may want to be avoided. Many of these things were piled up, and, at the end, I was not sure what I gained by reading the manuscript.

*A2: We disagree. These terms are used extensively throughout the manuscript in discussing the origin of the observed results. In fact we refer back to these terms many times, be it when discussing O:C ratios or the isotope effects. We have revised the conclusions to be sure that we resolve questions raised earlier in the manuscript.*

R3: If so, the authors should fairly evaluate the possible influence (how and how much) from the functionalization and fragmentations on the estimation of d13C shown in Table 5. By giving the range of d13C variations by these effects, readers will have an idea of the feasibility of this method. The current evaluation on "position dependent isotope effect" seems biased. Both are needed.

R2: The way to refer Supplementary Information, Tables, and Figures, also needs to be checked with the author's guide.

*A2: Thank you, we have checked that our usage is consistent with the preparation guidelines. Specifically, the term 'in the SI' was omitted throughout the manuscript.*

R3: Corrections were confirmed.

R2: Many figure captions and table headings also included unnecessary description, which can be inserted in the text.

*A2: We have revised the figure captions and table headings in line with the preparation guidelines. However, please note that the author's guide asks for concise but descriptive captions. We tried our best to delete unnecessary content from the captions. In case abbreviations were included in the captions, doubles were deleted if they were introduced in the text before.*

R3: The way the authors changed is not satisfactory, but the editor may accept this form. If not, the authors may consider the following suggestion.

Please check figure captions and table headings in other papers. Figure captions and table headings are sort of titles in general, but those in this manuscript are made of several sentences, which are unusual to me. The

following is just an example table heading for a Table 2. "*Table 2. Stable carbon isotope ratio of OC and TC ($\Delta(^{13}C)$), total mass loaded ($M^{PTR\text{-}MS}$), and O:C ratio for filter samples collected from smog chamber experiments.*"
Then provide some detail information in footnotes like
"[a] $\Delta(^{13}C) = \delta(^{13}C) - \delta_{TC}(^{13}C)$", "[b] Measured by Thermal-desorption isotope analysis.", "Measured by PTR-MS"

R2: For these reasons, I regret to say, but the manuscript does not seem to be ready to be published. Thorough revision is still needed. It may be the best to revise the manuscript with professional editors for expression in English, but this does not secure to turn the manuscript well organized. It is still the authors' responsibility that the order of explanations (sentence by sentence) and discussion (paragraph by paragraph and subsection by subsection) need to be well organized prior to the English check in order to make the flow of context smooth in the final form.

*A2: Thank you for the opportunity to revise the manuscript to improve clarity, organization and readability.*

R3: No problem.

R2: It is a minor thing, but the column containing "t/h" in Table 2 has also the information of "V/m^3". Those are different information and should be shown in different column.

*A2: Since the sample flow was constant, these two values actually represent the same information. However, we follow your recommendation and separated them into two columns.*

R3: Confirmed.

R2: I also recommend the authors to show units in brackets instead of slash in this table for consistency (i.e., the slash of "micro g / m^3" is meant a division, while the slash of "V/m^3" is meant for unit.

*A2: In both cases, the slash is division. This is the method recommended by SI, NIST, IUPAC and others, see for example the IUPAC Green Book.*

R3: The authors likely misunderstood my message. According to this explanation, "$V$/m³" in Table 2 would mean volume per cubic meter. I suggested to change the "V/m³" to V (m³), the "$M^{PTR\text{-}MS}$ / µg/m³" to $M^{PTR\text{-}MS}$ (µg/m³), and all other column headings and units accordingly.

R2: Besides the writing issues, the followings are my responses (underlined) to the authors' comments (Italic) to my previous comments (bold). Please note that I am responding to only the authors' comments that I have opinions on. For those that I am not responding to, it is either of that I am satisfied with the changes or issues are overlapping with those I am already responding to.

*A2: Thank you. We will add our response after your underlined comments. We omitted the quoted texts from the last revision for the sake of readability.*

R3: Okay.

General comments

R1: After reading the manuscript, I had impression that points the authors argued were unclear: pieces of discussions seemed to be fragmented.

*A1: The discussion was overhauled and there is now a separate section 'Discussion' to allow a clearer distinction between results and discussion. Furthermore, a reaction scheme (new FIgure 1) was added as suggested and used to guide the reader through the paper.*

R2: Thanks for attempting to revise along my advice, but this change is not satisfactory. I wanted the authors to focus on the important messages that the authors want to tell readers, and logically justify the arguments stepwise using own experimental results and references. The current form seems that the discussion section in the previous manuscript was just divided. The contents and arguments were still confusing. In addition, the "Results" section in the current manuscript substantially includes authors' interpretation and discussion. It is okay to separate the "Results" and "Discussion" sections, but please give only the observed results in the *"Results", and analyze and evaluate own data with references in "Discussion".*

*A2: We rearranged some of the content, but there is still some discussion left*

*in the 'Results' section, as this is very hard to be avoided in a paper like this. We have tried many different structures prior to submitting among the authors and in fact didn't want to have them divided until we go the reviewer's comments. Concerning the numbers in Table 5, they are still part of the 'Results' section as they are derived from basic algebra.*

R3: The data given in Table 5 are the results of estimation, which include the authors' interpretation that others may not agree with. In general, a result section provides measurement results (like raw data in Table 1, 2, and 4, and Figure 2 - 7), which are independent of way to interpret. Then, data interpretation, likewise the information provided in Table 5, is shown in a discussion section. Similarly, the text in the result section should describe what we can see in the raw data, such as time series variation, magnitudes, etc. Keep it as simple, and analysis of data and its interpretation should be described in the discussion section. If you are going to discuss the interpretation right after presenting the raw data, then combine these sections as "results and discussion".

R1: I also had impression that some information and discussion (e.g., already studied C:O ratios and natural processes with isotope fractionations) were very detailed, and some (e.g., chemical reactions that lead to formation of low volatility products, calculation for predicted d13C in Table 5) were insufficient.

*A1: We have shortened the former and expanded the latter. The calculations leading to the numbers shown in Table 5 are now illustrated with an example and Table 5 was overhauled. The new reaction scheme highlights the reaction pathways that lead to the calculations in Table 5.*

R2: I was a bit confused to find "example" illustration for the calculation of position specific d13C of Dgas and Daerosol in Table 5, but the authors probably meant S5 in the Supplementary Information. I understood that the calculations were based on NMR data, which are for unreacted alpha-pinene. From the revised manuscript, however, I could not retrieve important information to explain the d13C of product Dgas and Daerosol in Table 5 using the position specific d13C of unreacted alpha-pinene.

*A2: We have changed the text to make it clear that the values calculated in*

*table five are based on the MCM mechanism presented in Figure 1. Note that the precision of our argument is limited as the branching ratios are not known with certainty as is now pointed out even more clearly, and in any case, will change depending on temperature, pressure, humidity, OH, NOx and so on. We do not refer to S5 (or NMR data) but Table 5 and it is hard for us to explain the very basic algebra behind the numbers in Table 5 in more simple terms than we have done already. We have however, pointed out the assumptions very clearly at the beginning of the paragraph. The numbers in T5 are derived from the position-specific enrichment listed in Table 2 as described in the text. The beginning of the passage now reads:*

*'Here a limiting case is presented for the isotopic difference of a number of plausible oxidation products. The underlying assumptions exclude isotope dependent changes in product branching ratios, as well as effects of temperature, relative humidity, pressure, ozone concentration, etc. This simplistic approach allows to estimate the maximum isotopic enrichment in a-pinene fragments using the Δi(13C) profiles obtained from PSIA.'*

R3: It is not necessary to care parameters at different conditions, but please use parameters under own experimental condition in order to interpret own data. There will be readers who wonder how the calculation was done for the numbers in Table 5, so please clearly explain it. I tried to calculate the isotope ratios using the information given in Table 1 (not Table 2, I guess) by assuming that all KIEs were ignored and the pinene reaction with ozone completed, but I could not reach to the point that the isotopic compositions of gas and aerosol phase products were determined. It seemed that quantitative information of each product (specifically carbon fractions of each product relative to the total product carbon), which wasn't given, was needed to determine d13C of gas-phase and condensed-phase products, if the calculation was based on the isotopic composition of each product in either gas or aerosol phase, which was estimated based on the information in Table1. Did the author get this quantitative information from MCM? If so, I expect specific values of rates are supposed to be given for all the reaction pathways, which will give branching ratios at branching reactions and yields of each product. By the way, please state explicitly if only CO, formaldehyde, and acetone are treated as "the gas-phase products" and the rest of products in Fig. 1 are in aerosol phase products in the calculation.

There is another point I would like to make. Although the authors replied

that they have pointed out the assumptions explicitly, those are still not clear in the text: that is, "The underlying assumptions exclude…" and "if a single carbon atom or reasonable combination…" are not clear enough to get what assumptions were actually used. Please describe them explicitly.

Given the reasonable assumptions and calculation for the isotopic compositions of products in gas and aerosol phases, it will be interesting to compare the determined values with the results from the simpler carbon mass balance calculation for the isotopic composition of gas and aerosol phase products (Irei et al., 2006).

R1: In addition, the authors may want to make more focus on what they can conclude from the data they gained. For example, the position-specific isotope analysis (PSIA) sounded a key achievement in the abstract. However, if my understanding is correct, the PSIA was performed only on the unreacted a-pinene. I did not get what conclusion the authors were going to draw from the comparison of this information with d13C of stepwisely evaporated OCs, each of which is still possibly a complex mixture of multiple substances as indicated by more than 400 of fragment ions in the table in S4 of SI. If the authors could identify a specific product possessing enriched/depleted 13C atom at a specific position, which may have originated from a specific position of a-pinene, they should make discussion with the observed evidence (e.g., the difference in d13C between the different samples).

*A1: Thank you for your comment. It is correct that PSIA was performed on the unreacted a-pinene only. PSIA on single components of the SOA mixture is not possible at this moment. The argument is that if one dominant reaction channel prefers a certain reaction site on the alpha-pinene, then the position specific isotope distribution might govern the overall isotope signature. However, we cannot fully conclude that. PSIA on a-pinene was performed for the first time and we believe that this is an important result for future research and therefore chose to leave it in the abstract. Other parts of the abstract and large parts of the introduction were rewritten. A reaction scheme was added to better guide the reader in the introduction specifically when it comes to the idea that PSIA could play a role in the 'bulk' isotopic composition.*

R2: Biased 13C distribution in intra molecular carbons of pinene is

interesting, and I am not going to be offensive against this finding. However, again, my point is that it is not clear how this PSIA is related to the product analysis presented in this paper. It is reasonable that the authors will combine this result with results of future compoundspecific product study to explore isotope balance between the products, but those are not available at this moment. The presentation of the result here may be justified if Dgas and Daerosol in Table 5 were calculated based on the measurement results of PSIA with reasonable assumptions. However, the assumptions (i.e., "maximum expected enrichment" and "branching ratios do not influence the isotopic composition of products") do not sound. According to Hoefs (1997), the initial isotope ratio, the degree of processing, and the fractionation factor are needed to calculate the isotope ratio of a processing substance or product at any arbitrary time. Presumably, the authors assumed ~ 100% of degree of alpha-pinene oxidation ("completion" in the text probably suggests this), however, this does not secure that the complex subsequent reactions also completed, depending on relative reaction speeds (relative to the speed of ozonolysis of alpha-pinene) of subsequent reactions leading to the production of formaldehyde, acetone, CO, and other products. That is, branching reactions significantly influence product isotopic composition, unless otherwise there is no isotope fractionation (i.e., KIE = 0 permil) at all of these reactions. If even only one of these branching reactions had a significant KIE, which seems more reasonable assumption according to Fisseha et al. (2009), this KIE will significantly influence the isotopic composition of remaining alpha-pinene or intermediate(s) as reactions proceeded. This varying isotopic composition of precursors will then influence the isotopic composition of products from other branching reaction channels because all or a part of products originate from the same precursor. In the text, it is not clear what substances are defined as "aesosol" and "gasphase", what number of KIEs and branching ratios were used for the calculation. I am not sure what "the maximum expected enrichment" is, but if the authors assume zero of

KIEs at all branching reactions and no branching reactions, I disagree with the assumptions. From the text I could not retrieve those information.

*A2: It would be great data to analyse if one had the opportunity to perform PSIA on several of the product species, however, that was not possible in this experimental setup and has to be left for further studies. In the meantime we only have the starting picture (PSIA of α-pinene) and the end point (bulk δ*

*values of SOA). While maybe not 100 % satisfying, the best we can do with these fragments is to figure out the potential effect, or so to say how much of the starting picture is left at the end. The presented results demonstrate conclusively that both PSIA and isotope dependent product branching ratios must be considered in studies of isotopic fractionation in SOA formation reactions, especially for natural products. The assumptions made behind the numbers in Table 5 are of course not perfect, but our goal here was to get an estimate of the potential size of the effect. The branching ratios in the alpha pinene ozonolysis mechanism are unknown and very importantly, the branching ratios will depend on temperature, relative humidity, and the concentrations of ozone, OH, NOx and other species. We applied a simplification to a complex mechanism, and we did that for the first time with strong results. Concerning the product branching ratios anything is possible. We argue in the text one reaction having a very large KIE is not likely based on mechanistic considerations (standard addition reactions to a large molecule) and on e.g. the experimental results of e.g. Anderson at al 2004 and similar papers. The anomalously large KIE's seen in e.g. the ozone formation reaction are anomalies. In line with this, Fisseha et al.'s KIE is for the initial ozonation reaction, not for subsequent reactions, up to our understanding. The assumption to neglect the isotope effects of the branching ratios is not as far-fetched as it may seem at first.*

R3: I agree with that Fisseha et al. (2008) provide the carbon KIE at the first reaction step, and the reported KIE in their paper can be significantly different from the KIE at the initial step of alpha-pinene ozonolysis here due to possibly different intra-molecular 13C distribution. However, this does not necessarily mean that the KIEs in subsequent reactions shown in Fig. 1 are negligible. Rather, it is more likely that significant carbon KIEs exist in the mechanism in Fig. 1 because many primary carbon reactions are involved. In addition, the authors consider the production of CO, formaldehyde, acetone, pinoaldehyde, pinonic and pinic acids in the mechanism, and this indicates that the authors accept branching reactions. The assumptions sound contradictory to the mechanism. In my opinion, the calculation based on the combination case of Rxn 2 and 3 is feasible, but the authors should give scientific reasons (e.g., checking a trend for yields of products vs extent of the initial reaction) why their assumptions can be justified. With this dataset I could have seen the biased interpretation on isotopic compositions of specific

products. For this reason, I referred in the previous reply that publishing the PSIA results here with results from future product-specific isotope analysis makes more sense.

R1: Lack of reaction mechanism for production of SOA from ozonolysis of pinene was also a problem to follow the discussions.

*A1: Thank you for your comment. We made a new figure (Fig. 1) that identifies how the site-specific α-pinene d13C values propagate through the reaction mechanism highlighting some key arguments of our paper.*

R2: Thanks for providing the reaction scheme. This will definitely make readers easier to follow the discussion. However, the scheme is incomplete and not reader-friendly to follow the reaction steps producing high and low volatility products. For example, taking a look at the production mechanism of pinonic acid and pinoaldehyde in Fig. 1, I don't know what was changed after the second processing step, "stabilization". I see that the red star mark on the carbon disappeared after the step, but what does this mean? I also don't get how "H2O" was involved in the subsequent reaction after the "stabilization". The same problem for the reaction with RO2 and HO2. The authors may want to check how other papers are presenting those reaction schemes. Second point is that the figure also does not show important elemental steps for the production of small products shown in the boxes in Fig. 1. The hidden reaction channels seem to involve reactions with C atoms, thus, must be key steps to explain the results of d13C calculations. Third point is that it is helpful for readers to see branching ratios in the figure.

*A2: Considering point 1, the mechanism presented in Figure 1 is consistent with the best available understanding, as found in the Master chemical Mechanism and in the paper we reference (Camredon et al., 2010). By quoting these references, we also invite the reader (and reviewer) to read the literature on the known reactions in case theya re not presented in enough detail. The high complexity of the involved reactions disallows much more simplified representations. Points 2 and 3: Many people would like to know these branching ratios and the elementary steps of the reactions, but they are not known and might never be. Many experiments have only been able to detect products, and even with this information, it is ambiguous what the*

*exact contribution from each branch of the mechanism is. We would happily include this information if it was available to a level of detail that was helpful to interpreting our results, however, this is not the case, up to our knowledge.*

R3: It is true that most of rates at each elemental step are not studied, but regardless of correctness MCM requires specific numbers for the calculation, as long as I understand. Please provide the used numbers in the kinetic calculation or state explicitly what the authors has done using MCM, if such a calculation was not made.

By the way, Fig. 1 seems to include the continuous errors from the paper by Camredon et al.: according to Fig. 1, Criegee intermediates are followed by fragmentation reactions, but the scheme labels some of those as "isomerization". The authors may want to correct those (check if there are such errors through the text, figures, and tables thoroughly), and to be consistent I suggest to label all reaction steps in Fig. 1 with "stabilization", "fragmentation", and "functionalizatoin". The current form shows some are labelled, some are not.

Specific comments

R1: P3l10: The authors may want to use a symbol of alpha instead of epsilon for the expression of fractionation factor.

*A1: We rewrote the section in question and now describe the Kinetic Isotope Effect (KIE) using $\varepsilon$ values, as is common practice. We believe this makes the explanation easier to follow and more clear. The replacement of $\varepsilon$ by $\alpha$ does not change the manuscript as they are simply related by $\varepsilon = \alpha - 1$ which is common knowledge within the field of isotope research (see for example Coplen, 2011).*

R2: Regarding the definition of kinetic isotope effects, Colpen (2011) defines a kinetic isotope effect as $12k/13k$, while Hoefs (1997) defines a fractionation factor as $13k/12k$. The authors may want to stick to either of these.

*A2: The paragraph was completely rewritten, see above.*

R3: Confirmed.

R1: Section 3.1.: The authors may want to present an example time series plot of ozone mixing ratios and pinene concentrations.

*A1: Among authors we discussed providing a plot of the ozone mixing ratios but decided against it. The main reason against it being an interference of organic vapours on the ozone detection system yielding spikes of the ozone mixing ratios. This was tested and confirmed in extra experiments, but with no conclusion / suggestion on how to resolve the issue or how to correct our data for it. In order to avoid an extended discussion on operation and detection mechanisms of ozone monitors, we opted to leave the plot out. The pinene concentration was not measured.*

R2: It is very important to have information of pinene concentration change over time at least. Is it possible to present such info based on MCM calculation with given experimental condition?

*A2: Based on the ozone concentration, the reaction rate of α-pinene with ozone and the length of the experiment, we are confident that all of the alpha pinene has reacted, as we state in Section 2.3.*

R3: Specific information will avoid unnecessary argument. This can be easily calculated with the initial pinene conc., minimal ozone conc., and the rate constant. It's up to the authors.

R1: In addition, according to Fig 7a, OC at 100 degree Celsius (7% of TC) has the largest difference of -3 permil in d13C. Any discussion for this difference? Does this difference attribute to different isotopic composition of different chemical species? This is the part the authors should make deeper discussion, I think.

*A1: We suspect the reviewer was commenting on panel b of that Figure and will give our answer accordingly. The second last paragraph of section 3 discusses the strong enrichment observed at 100 C. However, the discussion has to stay vague as no clear conclusions can be drawn from the data. Unfortunately, no species or group of species could be assigned to this enrichment. We omitted the last part of the paragraph which dealt with the back filters and added another consideration based on oligomerisation. The*

*text was changed to:*

*"The reason behind the enrichment in material desorbing from front filters at 100 °C cannot be unambiguously identified. The chemical analysis did not allow to identify single compounds or groups of compounds that contributed significantly more to the total aerosol concentration at 100 °C than at 150 °C and therefore could lead to the observed enrichment. Isotope effects associated with sampling artefacts, which are generally not well known, provide room for speculation on how to interpret the enrichment at 100 °C. During the negative sampling artefact, isotopically light isotopologues revolatilise from the ensemble of sampled compounds preferentially leading to an overall isotopic enrichment in compounds that are left on the filter. Revolatilisation should have it's largest effect at 100 °C. Another explanation can be based on oligomerisation. Hall and Johnston (2012) observed significant evaporation of oligomers in a thermodenuder already at desorption temperatures below 100 °C. The effect of oligomerisation on isotopes is not known, but if it leads to enrichment, the fragments of decomposed oligomers could be enriched and explain our observations."*

R2: Sorry for confusion. It is correct that I was pointing on the panel b. I think it is not necessary to deeply argue with speculation, if there is no reasonable scientific explanation for the difference. However, it is important to point out the difference explicitly and briefly state possible reasons for the difference so that others can hint for future studies to progress the science.

*A2: The section was removed.*

R3; Okay.

R1: P16l20-33: There are papers discussing site specific isotope fractionations. Why do the authors argue with reactive isotope fractionations using the reference of sorption isotope fractionation of vanillin?

*A1: There is not much literature available on the topic as the analysis technique is rather new. However, we now included the Botosoa et al., 2009b reference in the text and omitted speculations about bond length variations.*

R2: I am not sure if we are misunderstanding our comments each other, but I can find references for position specific carbon isotope fractionation at different types of chemical reactions (not for physical phenomena), such as

Singleton and Szymanski (1999, JACS) and etc. Because the authors are discussing "carbon isotope fractionation" there, it is not necessary to compare only with the results obtained by NMR measurements. Carbon KIEs for elemental reactions obtained by different techniques have been studied for long, some of those may be comparable. Of course, the current reference is also okay, I think.

*A2: Thank you, we agree that the reference we chose is appropriate.*
R3: Okay.

---

## Author Response (AR2)

**Author response to reviewer comments**

**acp-2016-98**
'**Chemical and isotopic composition of secondary organic aerosol generated by α-pinene ozonolysis'**
Carl Meusinger, et al.

I checked the revised manuscript. The authors responded to our comments in professional manner, and attempted to reorganize the manuscript along the comments. *Thank you.*
It is intriguing to see what products the ozonolysis of alpha-pinene produces and how carbon isotopes are balanced between the gas-phase and condensed phase products. *We agree.*
Furthermore, application of PSIA to reaction mechanism study seems to have potential to fingerprint its products in general. After reading the manuscript carefully, however, it was found that the corrections and modifications were not satisfactory and the manuscript has writing issues that need to be solved. Overall, the manuscript still needs substantial revision. The followings are some examples of writing issues I encountered.
There were many sentences that it was hard to understand. For example, in the sentence in P4L29, "If...", I didn't know what "depletion of the gasphase" means, and I didn't get the logic why depletion of 13C at C9 position leads to it.
*We have rewritten the passage to be more clear:*
*'It is widely understood that the enrichment/depletion of a product depends on the enrichment of the starting material, the isotopic fractionation occurring in the mechanism of its formation, and the extent of reaction. Using PSIA we can take this analysis one step further: the enrichment of a product will depend on the position-dependent enrichments of the atoms from which it is formed. For example the ozonolysis mechanism transfers the C9-atom in a-pinene into several small, volatile products (see blue squares in Fig. 1). If the C9-position was depleted in 13C the gas phase products containing this atom would be depleted, and the SOA correspondingly enriched, if the position dependent effect was stronger than kinetic isotope effects.'*

For the sentence in P7L26, I could not understand what kind of correction the authors have made. The authors should make a message in every sentence clear as writing.
*This passage describes the process of assigning chemical formulas and controlling / checking the assigned chemical formulas. No 'correction' was described or made in this passage. We are sorry to say that we are not sure how to rewrite the passage to improve clarity.*

There were also unnecessary sentences and paragraphs (I was impressed that those were unnecessary, but those may turn to be valuable, depending on for what the authors are going to explain using those sentences and paragraphs). For example, a paragraph in P3L9-18 defines a KIE and an epsilon value. However, any KIE and epsilon value for the pinene reaction never came out in the text. Those definitions may not be necessary, if the authors are not going to use the KIE in the interpretation.
*The passage in question was completely rewritten:*
*'Isotopic substitution can cause reactions to be faster or slower than for the*

*unsubstituted case, kinetically fractionating the isotopes and leading to isotopic enrichment or depletion in the products. This is known as the kinetic isotope effect (KIE). If a reaction leads to a single product, the product may initially have a different abundance, but due to the law of mass balance will achieve the same abundance as the reagent as the reaction goes to completion. If a reaction has multiple product channels, enrichment or depletion will occur if there are isotope dependent changes in the product branching ratios.'*

It is also not clear how the authors are going to discuss the observed isotope fractionation with functionalization, fragmentation, and oligomerization introduced in P3L27-P4L15. This paragraph impressed me at the beginning that the authors would discuss isotope fractionation at these reactions later to interpret the own data, but actually those were not discussed. If those are not available from own data or references, statements for combinational use of "isotope" and "functionalization (or oligomerization, fragmentation)" using one or two paragraphs may want to be avoided. Many of these things were piled up, and, at the end, I was not sure what I gained by reading the manuscript.

*We disagree. These terms are used extensively throughout the manuscript in discussing the origin of the observed results. In fact we refer back to these terms many times, be it when discussing O:C ratios or the isotope effects. We have revised the conclusions to be sure that we resolve questions raised earlier in the manuscript.*

The way to refer Supplementary Information, Tables, and Figures, also needs to be checked with the author's guide.

*Thank you, we have checked that our usage is consistent with the preparation guidelines. Specifically, the term 'in the SI' was omitted throughout the manuscript.*

Many figure captions and table headings also included unnecessary description, which can be inserted in the text.

*We have revised the figure captions and table headings in line with the preparation guidelines. However, please note that the author's guide asks for concise but descriptive captions. We tried our best to delete unnecessary content from the captions. In case abbreviations were included in the captions, doubles were deleted if they were introduced in the text before.*

For these reasons, I regret to say, but the manuscript does not seem to be ready to be published. Thorough revision is still needed. It may be the best to revise the manuscript with professional editors for expression in English, but this does not secure to turn the manuscript well organized. It is still the authors' responsibility that the order of explanations (sentence by sentence) and discussion (paragraph by paragraph and subsection by subsection) need to be well organized prior to the English check in order to make the flow of context smooth in the final form.

*Thank you for the opportunity to revise the manuscript to improve clarity, organization and readability.*

It is a minor thing, but the column containing "t/h" in Table 2 has also the information of "V/m^3". Those are different information and should be shown in different column.

*Since the sample flow was constant, these two values actually represent the same information. However, we follow your recommendation and separated them into two*

*columns.*

I also recommend the authors to show units in brackets instead of slash in this table for consistency (i.e., the slash of "micro g / m^3" is meant a division, while the slash of "V/m^3" is meant for unit.
*In both cases, the slash is division. This is the method recommended by SI, NIST, IUPAC and others, see for example the IUPAC Green Book.*

Besides the writing issues, the followings are my responses (underlined) to the authors' comments (Italic) to my previous comments (bold). Please note that I am responding to only the authors' comments that I have opinions on. For those that I am not responding to, it is either of that I am satisfied with the changes or issues are overlapping with those I am already responding to.
*Thank you. We will add our response after your underlined comments. We omitted the quoted texts from the last revision for the sake of readability.*

**General comments**
**After reading the manuscript, I had impression that points the authors argued were unclear: pieces of discussions seemed to be fragmented.**
*The discussion was overhauled and there is now a separate section 'Discussion' to allow a clearer distinction between results and discussion. Furthermore, a reaction scheme (new FIgure 1) was added as suggested and used to guide the reader through the paper.*
Thanks for attempting to revise along my advice, but this change is not satisfactory. I wanted the authors to focus on the important messages that the authors want to tell readers, and logically justify the arguments stepwise using own experimental results and references. The current form seems that the discussion section in the previous manuscript was just divided. The contents and arguments were still confusing. In addition, the "Results" section in the current manuscript substantially includes authors' interpretation and discussion. It is okay to separate the "Results" and "Discussion" sections, but please give only the observed results in the "Results", and analyze and evaluate own data with references in "Discussion".
We rearranged some of the content, but there is still some discussion left in the 'Results' section, as this is very hard to be avoided in a paper like this. We have tried many different structures prior to submitting among the authors and in fact didn't want to have them divided until we go the reviewer's comments. Concerning the numbers in Table 5, they are still part of the 'Results' section as they are derived from basic algebra.

**I also had impression that some information and discussion (e.g., already studied C:O ratios and natural processes with isotope fractionations) were very detailed, and some (e.g., chemical reactions that lead to formation of low volatility products, calculation for predicted d13C in Table 5) were insufficient.**
*We have shortened the former and expanded the latter. The calculations leading to the numbers shown in Table 5 are now illustrated with an example and Table 5 was overhauled. The new reaction scheme highlights the reaction pathways that lead to the calculations in Table 5.*
I was a bit confused to find "example" illustration for the calculation of position specific d13C of Dgas and Daerosol in Table 5, but the authors

probably meant S5 in the Supplementary Information. I understood that the calculations were based on NMR data, which are for unreacted alpha-pinene. From the revised manuscript, however, I could not retrieve important information to explain the d13C of product Dgas and Daerosol in Table 5 using the position specific d13C of unreacted alpha-pinene.

We have changed the text to make it clear that the values calculated in table five are based on the MCM mechanism presented in Figure 1. Note that the precision of our argument is limited as the branching ratios are not known with certainty as is now pointed out even more clearly, and in any case, will change depending on temperature, pressure, humidity, OH, NOx and so on. We do not refer to S5 (or NMR data) but Table 5 and it is hard for us to explain the very basic algebra behind the numbers in Table 5 in more simple terms than we have done already. We have however, pointed out the assumptions very clearly at the beginning of the paragraph. The numbers in T5 are derived from the position-specific enrichment listed in Table 2 as described in the text.

The beginning of the passage now reads:

'Here a limiting case is presented for the isotopic difference of a number of plausible oxidation products. The underlying assumptions exclude isotope dependent changes in product branching ratios, as well as effects of temperature, relative humidity, pressure, ozone concentration, etc. This simplistic approach allows to estimate the maximum isotopic enrichment in a-pinene fragments using the $\Delta i(13C)$ profiles obtained from PSIA.'

**In addition, the authors may want to make more focus on what they can conclude from the data they gained. For example, the position-specific isotope analysis (PSIA) sounded a key achievement in the abstract. However, if my understanding is correct, the PSIA was performed only on the unreacted a-pinene. I did not get what conclusion the authors were going to draw from the comparison of this information with d13C of stepwisely evaporated OCs, each of which is still possibly a complex mixture of multiple substances as indicated by more than 400 of fragment ions in the table in S4 of SI. If the authors could identify a specific product possessing enriched/depleted 13C atom at a specific position, which may have originated from a specific position of a-pinene, they should make discussion with the observed evidence (e.g., the difference in d13C between the different samples).**

*Thank you for your comment. It is correct that PSIA was performed on the unreacted a-pinene only. PSIA on single components of the SOA mixture is not possible at this moment. The argument is that if one dominant reaction channel prefers a certain reaction site on the alpha-pinene, then the position specific isotope distribution might govern the overall isotope signature. However, we cannot fully conclude that. PSIA on a-pinene was performed for the first time and we believe that this is an important result for future research and therefore chose to leave it in the abstract. Other parts of the abstract and large parts of the introduction were rewritten. A reaction scheme was added to better guide the reader in the introduction specifically when it comes to the idea that PSIA could play a role in the 'bulk' isotopic composition.*

Biased 13C distribution in intra molecular carbons of pinene is interesting, and I am not going to be offensive against this finding. However, again, my point is that it is not clear how this PSIA is related to the product analysis presented in this paper. It is reasonable that the authors will combine this result with results of future compoundspecific product study to explore isotope balance between the products, but those are not available at this moment. The presentation of the result here may be justified if Dgas and Daerosol in Table 5 were calculated based on the measurement results of PSIA with reasonable assumptions. However, the assumptions (i.e., "maximum expected enrichment" and "branching ratios do not influence the isotopic composition of products") do not sound. According to Hoefs (1997), the initial isotope ratio, the degree of processing, and the fractionation factor are needed to calculate the isotope ratio of a processing substance or product at any arbitrary time. Presumably, the authors assumed ~ 100% of degree of alpha-pinene oxidation ("completion" in the text probably suggests this), however, this does not secure that the complex subsequent reactions also completed, depending on relative reaction speeds (relative to the speed of ozonolysis of alpha-pinene) of subsequent reactions leading to the production of formaldehyde, acetone, CO, and other products. That is, branching reactions significantly influence product isotopic composition, unless otherwise there is no isotope fractionation (i.e., KIE = 0 permil) at all of these reactions. If even only one of these branching reactions had a significant KIE, which seems more reasonable assumption according to Fisseha et al. (2009), this KIE will significantly influence the isotopic composition of remaining alpha-pinene or intermediate(s) as reactions proceeded. This varying isotopic composition of precursors will then influence the isotopic composition of products from other branching reaction channels because all or a part of products originate from the same precursor.

In the text, it is not clear what substances are defined as "aesosol" and "gasphase", what number of KIEs and branching ratios were used for the calculation. I am not sure what "the maximum expected enrichment" is, but if the authors assume zero of KIEs at all branching reactions and no branching reactions, I disagree with the assumptions. From the text I could not retrieve those information.

It would be great data to analyse if one had the opportunity to perform PSIA on several of the product species, however, that was not possible in this experimental setup and has to be left for further studies. In the meantime we only have the starting picture (PSIA of a-pinene) and the end point (bulk δ values of SOA). While maybe not 100 % satisfying, the best we can do with these fragments is to figure out the potential effect, or so to say how much of the starting picture is left at the end. The presented results demonstrate conclusively that both PSIA and isotope dependent product branching ratios must be considered in studies of isotopic fractionation in SOA formation reactions, especially for natural products.

The assumptions made behind the numbers in Table 5 are of course not perfect, but our goal here was to get an estimate of the potential size of the effect. The branching ratios in the alpha pinene ozonolysis mechanism are unknown and very importantly, the branching ratios will depend on temperature, relative humidity, and the concentrations of ozone, OH, NOx and other species. We applied a simplification to a complex mechanism, and we did that for the first time with strong results.

Concerning the product branching ratios anything is possible. We argue in the text one reaction having a very large KIE is not likely based on mechanistic considerations (standard addition reactions to a large molecule) and on e.g. the experimental results of e.g. Anderson at al 2004 and similar papers. The anomalously large KIE's seen in e.g. the ozone formation reaction are anomalies. In line with this, Fisseha et al.'s KIE is for the initial ozonation reaction, not for subsequent reactions, up to our understanding. The assumption to neglect the isotope effects of the branching ratios is not as far-fetched as it may seem at first.

**Lack of reaction mechanism for production of SOA from ozonolysis of pinene was also a problem to follow the discussions.**

*Thank you for your comment. We made a new figure (Fig. 1) that identifies how the site-specific α-pinene d13C values propagate through the reaction mechanism highlighting some key arguments of our paper.*

Thanks for providing the reaction scheme. This will definitely make readers easier to follow the discussion. However, the scheme is incomplete and not reader-friendly to follow the reaction steps producing high and low volatility products. For example, taking a look at the production mechanism of pinonic acid and pinoaldehyde in Fig. 1, I don't know what was changed after the second processing step, "stabilization". I see that the red star mark on the carbon disappeared after the step, but what does this mean? I also don't get how "H2O" was involved in the subsequent reaction after the "stabilization". The same problem for the reaction with RO2 and HO2. The authors may want to check how other papers are presenting those reaction schemes. Second point is that the figure also does not show important elemental steps for the production of small products shown in the boxes in Fig. 1. The hidden reaction channels seem to involve reactions with C atoms, thus, must be key steps to explain the results of d13C calculations. Third point is that it is helpful for readers to see branching ratios in the figure.

Considering point 1, the mechanism presented in Figure 1 is consistent with the best available understanding, as found in the Master chemical Mechanism and in the paper we reference (Camredon et al., 2010). By quoting these references, we also invite the reader (and reviewer) to read the literature on the known reactions in case theya re not presented in enough detail. The high complexity of the involved reactions disallows much more simplified representations.

Points 2 and 3: Many people would like to know these branching ratios and the elementary steps of the reactions, but they are not known and might never be. Many experiments have only been able to detect products, and even with this information, it is ambiguous what the exact contribution from each branch of the mechanism is. We would happily include this information if it was available to a level of detail that was helpful to interpreting our results, however, this is not the case, up to our knowledge.

**Specific comments**
**P3l10: The authors may want to use a symbol of alpha instead of epsilon for the expression of fractionation factor.**

*We rewrote the section in question and now describe the Kinetic Isotope Effect (KIE) using ε values, as is common practice. We believe this makes the explanation easier to follow and more clear. The replacement of ε by α does not change the manuscript as they are simply related by ε=α−1 which is common knowledge within the field of isotope research (see for example Coplen, 2011).*

Regarding the definition of kinetic isotope effects, Colpen (2011) defines a kinetic isotope effect as 12k/13k, while Hoefs (1997) defines a fractionation factor as 13k/12k. The authors may want to stick to either of these.

The paragraph was completely rewritten, see above.

**Section 3.1.: The authors may want to present an example time series plot of ozone mixing ratios and pinene concentrations.**

*Among authors we discussed providing a plot of the ozone mixing ratios but*

*decided against it. The main reason against it being an interference of organic vapours on the ozone detection system yielding spikes of the ozone mixing ratios. This was tested and confirmed in extra experiments, but with no conclusion / suggestion on how to resolve the issue or how to correct our data for it.*

*In order to avoid an extended discussion on operation and detection mechanisms of ozone monitors, we opted to leave the plot out. The pinene concentration was not measured.*

It is very important to have information of pinene concentration change over time at least. Is it possible to present such info based on MCM calculation with given experimental condition?

Based on the ozone concentration, the reaction rate of a-pinene with ozone and the length of the experiment, we are confident that all of the alpha pinene has reacted, as we state in Section 2.3.

**In addition, according to Fig 7a, OC at 100 degree Celsius (7% of TC) has the largest difference of -3 permil in d13C. Any discussion for this difference? Does this difference attribute to different isotopic composition of different chemical species? This is the part the authors should make deeper discussion, I think.**

*We suspect the reviewer was commenting on panel b of that Figure and will give our answer accordingly. The second last paragraph of section 3 discusses the strong enrichment observed at 100 C. However, the discussion has to stay vague as no clear conclusions can be drawn from the data. Unfortunately, no species or group of species could be assigned to this enrichment. We omitted the last part of the paragraph which dealt with the back filters and added another consideration based on oligomerisation.*

Sorry for confusion. It is correct that I was pointing on the panel b. I think it is not necessary to deeply argue with speculation, if there is no reasonable scientific explanation for the difference. However, it is important to point out the difference explicitly and briefly state possible reasons for the difference so that others can hint for future studies to progress the science.

The section was removed.

**P16l20-33: There are papers discussing site specific isotope fractionations. Why do the authors argue with reactive isotope fractionations using the reference of sorption isotope fractionation of vanillin?**

*There is not much literature available on the topic as the analysis technique is rather new. However, we now included the Botosoa et al., 2009b reference in the text and omitted speculations about bond length variations.*

I am not sure if we are misunderstanding our comments each other, but I can find references for position specific carbon isotope fractionation at different types of chemical reactions (not for physical phenomena), such as Singleton and Szymanski (1999, JACS) and etc. Because the authors are discussing "carbon isotope fractionation" there, it is not necessary to compare only with the results obtained by NMR measurements. Carbon KIEs for elemental reactions obtained by different techniques have been studied for long, some of those may be comparable. Of course, the current reference is also okay, I think.

Thank you, we agree that the reference we chose is appropriate.

[revised manuscript text omitted]

---

## Author Response (AR3)

Letter to the Editor

We would like to thank the reviewer for constructive feedback which has allowed us, we believe, to reconstruct the manuscript in order to clearly communicate the work and remove and or clarify potential points of misunderstanding.

The reviewer writes, *The main reasons are that the authors have not clearly explain how the predicted isotopic compositions of "gas phase products" and "aerosol products" were calculated using the results of PSIA and the simple assumptions the authors have made do not sound. In this paper, the calculation for the isotopic compositions of gas and condensed phase products using the results of PSIA for the unreacted alpha-pinene seems the novelty. The approach itself is interesting, if the calculation makes sense. Unfortunately, the current information given in the manuscript was insufficient for readers to evaluate and follow the method, thus, the reviewer still has a question how the results of PSIA are connected to the results of thermal-desorption/PTR-MS analysis.*

We feel we did not adequately distinguish experimental observations and discussion based on the observations, and have addressed that by restructuring the manuscript, moving Table 5 into the discussion and rewriting the text. Our logic is as follows. We performed the described experiment and collected data, and we present the data. The reviewer has not raised concerns about the experiment or the observations, and we believe that these are the primary result of the study. Next comes the discussion. Due to mass balance, the observed enrichment in SOA must be balanced by depletion in gas phase products, since as we argue, alpha pinene ozonolysis has gone to completion. However, since the reaction mechanism is not known in detail (complex, not described in the literature, dependent on humidity, temperature, rate of oxidation, $O_3$ to OH and so on) and cannot be determined from our experiment (incomplete information about gas phase), we are left to interpret an incomplete case. One instructive limiting case, is to ask which atoms in alpha pinene are transformed into the known gas phase products? The possibilities are presented in Table 5, which has now been moved to the discussion. We show that while there are some gas phase products that would be enriched, it is quite reasonable that, overall, the gas phase products will be depleted. In particular, acetone always seems to be depleted. Due to limited data on the gas phase (average isotopic composition only), we cannot use data to make an argument as to whether the origin of the SOA depletion/gas phase enrichment is due to site specific effects or the effect of isotopic substitution on branching ratios in the ozonolysis. Nonetheless we believe we have made a solid case for the position that it is very important to consider position sensitive isotopic enrichments and depletions, and not just bulk KIEs, especially for natural products which are known as a group to have significant site specific enrichments and depletions relative to fossil/synthetic compounds. The last paragraph in the introduction has been completely rewritten and now addresses many of the reviewer's raised concerns upfront. We hope that, this approach 'sets the stage' for the paper in a fair and clear manner. It has to be pointed out that the manuscript has been heavily rewritten

several times since first submission. The last paragraph in the introduction now reads as :

"In this study we have designed an experiment to use a combination of position specific, sample-average and molecule-average isotopic abundance data to investigate the formation of secondary organic aerosol from the ozonolysis of a-pinene. Due to mass balance, the enrichment (or depletion) in SOA must be balanced by depletion (or enrichment) in gas phase products since in our experiment a-pinene ozonolysis has gone to completion. However, since the reaction mechanism is not known in detail and cannot be determined from our experiment, we are left to interpret an incomplete case. One instructive limiting case is to ask which atoms in alpha pinene are transformed into known gas phase products. We argue that the observed isotope distribution in SOA is likely balanced by the opposite pattern in the gas phase products, and that this pattern could be produced by site specific enrichments in the starting material and current incomplete knowledge of the reaction mechanism alone, aside from whatever effects isotopic substitution may have on product branching ratios. Thus, we make the case that position sensitive isotopic enrichments and depletions, and not just bulk KIEs, are an important element of explaining field observations, especially for natural products which are known as a group to have significant site specific enrichments and depletions relative to fossil/synthetic compounds. The goal of this study is to provide detailed isotopic and chemical characterization of newly formed a-pinene SOA and to shed light on the mechanisms that govern isotopic fractionation in the formation of fresh SOA."

We assert that the novelty is the measurements themselves, which suggest the novel conclusion that bulk KIEs may not be the most important factor in explaining the observed fractionation from precursor to SOA. We have modified the manuscript in order to address this potential confusion, by clarifying what is observation and what is discussion/interpretation.

The reviewer writes, *It is generally known that a KIE is reaction-specific. A KIE for a unidirectional reaction of substance A (e.g., functionalization reactions by oxidants) can be determined experimentally, regardless of the intra-molecular 13C distribution in the reactant A.* And further, *To predict the isotopic composition of individual product, the intra-molecular 13C distribution of A and the fragmentation mechanism are also needed in addition to the parameters and valuable referred earlier because the chance that bond-dissociation involves a 13C atom will depend on the intra-molecular distribution of 13C atoms, and this may cause the biased distribution of 13C between the product B and C, which cannot be determined by the Rayleigh function only.*

We reply that mathematically, it is straightforward to distinguish between the isotope abundance in an element in a given molecule (by definition an average over all the atoms at all positions), and the isotope abundance at specific sites. Similarly, KIEs could be either an average for a molecule as a whole, or for specific sites. Similarly, Rayleigh plots could be constructed for both cases, molecule average and site-specific.

*One can easily imagine that the prediction of isotopic composition will be more complex when each of the branching reactions produces multiple products. Furthermore, the prediction of isotopic composition for products is more difficult when oligomerization of these fragmented products takes place subsequently*

We agree! But still, a very interesting and relevant problem. One simplifying factor in the current experiment is that the alpha pinene ozonolysis reaction has gone to completion.

*…but what is not clear in the revised sentences is the statement, "if the position dependent effect was stronger than kinetic isotope effects." It is because the case the authors described (i.e., 13C depletion in one of the products and 13C enrichment in the other product) would be true regardless of the magnitude of position dependent (or kinetic isotope) effect. The authors need to rewrite this part.*

First of all, the kinetic isotope effect can be excluded since the alpha pinene ozonolysis has gone to completion. The choices are between position sensitive/atom specific isotopic propagation, and isotope dependent effects on product branching ratios. We may have an opinion or a guess about this, but we simply cannot know, based on this experiment and without further experiments, which is more important. We do not claim to know in the manuscript. With this paper we would simply like to raise awareness of the issue. We have rewritten the text to be clear on this point.

*R3: The described cases are Rxn1-2, but the actual case shown in Fig. 1 is likely the combination of Rxn 2 and 3. The authors should describe such possibility and their influence too. The revised sentences already described are also better spelled out.*

Mass balance must always apply. Rxn 2 is central to the oxidation mechanism, but too, if the reaction goes to completion, then we would not be able to observe Rxn 2 type KIEs. In contrast isotope dependent changes in Rxn 3 would be observed.

*..the authors should fairly evaluate the possible influence (how and how much) from the functionalization and fragmentations on the estimation of d13C shown in Table 5. By giving the range of d13C variations by these effects, readers will have an idea of the feasibility of this method. The current evaluation on "position dependent isotope effect" seems biased. Both are needed.*

This is exactly the point of Table 5, to show what fractionations can be expected in reaction fragments. We have included the spreadsheet calculation used to calculate the values presented in Table 5 so the editor and reviewer can see how they are obtained; we believe it is straightforward. We will ask the editorial office to make the excel sheet 'Alpha-pinene site specific data analysis.xlsx' available to the reviewer and editor to be completely transparent and show the calculations used for determining the values in Table 5 in the manuscript. The reviewer is invited to check the calculations (we do not intend to publish this document). Hopefully the reviewer can agree that it is very simple algebra and in fact exactly as explained in the manuscript. Specifically, the excel sheet shows

IRMS bulk data and the PSIA data (as in Table 1 in the manuscript) in the boxed cells. The main calculations are highlighted in blue and the exported data (i.e. what is shown in Table 5 in the manuscript) is highlighted in yellow.
The calculations are based on the chemical mechanism that relates specific atoms in alpha pinene to specific products. These are highlighted by the 'x' marks below the PSIA data. The blue columns are then calculated based on this indication. We hope we can convince the reviewer that the calculations are straightforward and do not include additional, hidden or otherwise obscure assumptions, other than stated in the manuscript. We have moved Table 5 into the discussion section to further emphasize the point that it is discussion and not an observed experimental result.

If the editorial office or typesetters feel we have gone against the accepted style of the journal we will be happy to rewrite the figure titles; we do not feel it is a problem.

*It may be the best to revise the manuscript with professional editors for expression in English, but this does not secure to turn the manuscript well organized. It is still the authors' responsibility that the order of explanations (sentence by sentence) and discussion (paragraph by paragraph and subsection by subsection) need to be well organized prior to the English check in order to make the flow of context smooth in the final form.*

We have reorganized the manuscript to more clearly indicate what is a result and what is discussion. In addition, native speakers of English have reviewed the text multiple times.

*The authors likely misunderstood my message. According to this explanation, "V/m₃" in Table 2 would mean volume per cubic meter. I suggested to change the "V/m₃" to V (m₃), the "M$_{PTR-MS}$ / mg/m₃" to M$_{PTR-MS}$ (mg/m₃), and all other column headings and units accordingly.*

We insist that our usage is in accord with the recommendations of international organizations including the Système international d'unités, the International Union of Pure and Applied Chemistry, the CRC Handbook, the American Physical Society and so on, where for example 'V/m$^3$' means volume in units of cubic meter, and the result of dividing V by m$^3$ is a pure number, which is what is written in the table entries. One example of many in ACP is the article by Atkinson, R., Baulch, D. L., Cox, R. A., Crowley, J. N., Hampson, R. F., Hynes, R. G., ... & Wallington, T. J. (2008). Evaluated kinetic and photochemical data for atmospheric chemistry: Volume IV-gas phase reactions of organic halogen species. Atmospheric Chemistry and Physics, 8, 4141-4496. The sixth column heading of Table 1 of this article is 'Temp. range/K' and the fifth column heading is 'Temp. dependence of $k$/cm$^3$ molecule$^{-1}$ s$^{-1}$'

On behalf of the coauthors,
C. Meusinger and M. S. Johnson

[revised manuscript text omitted]
\left(^{13}\mathrm{C}\right) = \frac{R_{\mathrm{sa}}\left(^{13}\mathrm{C}\right)}{R_{\mathrm{VPDB}}\left(^{13}\mathrm{C}\right)} - 1 \tag{2}$$

Here, $R_{\mathrm{sa}}\left(^{13}\mathrm{C}\right)$ and $R_{\mathrm{VPDB}}\left(^{13}\mathrm{C}\right)$ denote the isotope ratios ($^{13}\mathrm{C}/^{12}\mathrm{C}$) in the sample and standard respectively.

In this study isotopic compositions of filter material are discussed relative to the isotopic composition of the initial $\alpha$-pinene, $\delta_{\mathrm{TC}}^{\mathrm{o1}}\left(^{13}\mathrm{C}\right)$, where TC denotes total carbon analysis (see below). Changes in isotopic composition are then reported as an isotopic difference (Coplen, 2011):

$$\Delta\left(^{13}\mathrm{C}\right) = \delta\left(^{13}\mathrm{C}\right) - \delta_{\mathrm{TC}}^{\mathrm{o1}}\left(^{13}\mathrm{C}\right) \tag{3}$$

$\Delta\left(^{13}\mathrm{C}\right) > 0$ indicates enrichment and $\Delta\left(^{13}\mathrm{C}\right) < 0$ 
[revised manuscript text omitted]

---

## Author Response (AR4)

**Reply to the reviewer.**

First of all we want to thank the reviewer for the useful comments. In the following we will answer the specific comments (marked in yellow), with text added to the manuscript written in italic.

This paper describes measurements of secondary organic aerosol (SOA) using two techniques – thermal desorption proton transfer reaction mass spectrometry (PTR-MS) and isotope-ratio mass spectrometry (IR-MS) – to better understand key properties of the aerosol, as well as the general processes underlying SOA formation. The authors use a novel approach for examining the source of the isotopic fragmentation, position-specific isotopic analysis (PSIA) of the parent VOC (a-pinene), and the primary conclusion is that carbon-to-carbon isotopic differences in the parent can describe much (if not all) of the observed isotopic fractionation in SOA formation. This is an interesting and novel result, and to my knowledge the first time this effect has been discussed in the context of atmospheric chemistry or SOA; it may be very helpful in the interpretation of data on 13C-ratios of ambient organic aerosol. It is thus certainly an appropriate topic for ACP. Below are comments that need to be addressed prior to publication.

Thank you very much.

Most importantly, the PTR-MS and IR-MS results seem largely unconnected. In particular, the PTR-MS results are barely referred to in the entire discussion section, which focuses entirely on the isotopic measurements. (The only exception is a brief mention on p. 16, lines 12-14.) Thus, in the paper's current form, the reason for including the PTR-MS data in this paper is quite unclear. Either the discussion of how the PTR-MS data relate to the IR-MS results needs to be expanded dramatically (i.e., with detailed discussions about how individual ions, or inferred reaction pathways), or else the PTR-MS part of the paper should be left out completely.

Thank you, this is a valid criticism. The PTRMS results are central to the work as they document the increase in O:C ratio as a-pinene is converted into SOA. One might have expected that a kinetic isotope effect would show that more highly oxidized species would have higher fractionations, this was not the case, we saw no association. This supports our conclusion that position sensitive effects are important. We have added this discussion to the conclusions We have added the following to the Discussion:
*As demonstrated by cross correlating the IRMS analysis with PTRMS, the D13C does not show a correlation with the aerosol oxidation state and the O:C ratio.*
In addition we thought we might be able to identify single compounds as the carrier of the isotopic signal but this was not the case. We have also added the following to

the Discussion:

*The PTRMS data showed that the isotopic enrichment in the SOA is carried by a broad range of oxidation products rather than one or a few dominant products.*
In addition, we use the PTRMS to check for any differences in SOA yield when using 1-butanol vs cyclohexane as the OH scavenger. We comment on this below in our reply to P. 13, line 21, where we have added the text:
*The higher enrichment observed for SOA formed in the presence of 1-butanol scavenger relative to cyclohexane scavenger may be because the latter has a higher SOA yield. This is corroborated by the PTRMS data see Table 2.*

Other comments

Throughout: PTR data are described in terms of "ion concentrations"; this is the incorrect term to use, since the mass concentrations (ng/m3) refer to the parent molecule, not the ion.
Thank you for the comment, we have gone through the manuscript and changed the nomenclature.

P. 9, line 12: An introductory sentence on the basic approach for PSIA (13C NMR) would be helpful here. (It's well described in the SI.)
Thanks for pointing that out. We added the following sentence at the start of section 2.7:
*Quantitative NMR spectrometry tuned for isotopic measurements was used to quantify the relative abundance of each carbon-isotopomer of alpha pinene.*

P. 11, line 18: I assume fragmentation "in the oven" refers to thermal decomposition of the molecules upon heating; this is the same effect as described later (lines 23-28). Probably this sentence should focus on decomposition within the PTR-MS only.
Yes, it seemed that the structure of the paragraph was confusing. We have moved the sentence:
*The detection of small ions that are not likely to be present in the particle phase by themselves over a wide range of desorption temperatures indicates oligomers.*
So that it does not break up the discussion of thermal decomposition.

P. 11, line 23-28: This effect has been shown to be important in a couple recent papers (Hilfiker-Lopez et al, 2016 ES&T 50:2200, Isaacman-VanWertz et al. ES&T 50:9952); those should be cited here.
Thank you for this interesting point. We have added the following to the paragraph in question: *Recent work by Hilfiker-Lopez et al. (2016) and Isaacman-VanWertz et al. (2016) have presented important examples of simultaneous chemical characterization of organic molecules in both the particle and gas phases, in order to better understand partitioning.*

We welcome these efforts.

We understand the reviewer's request, but want to point out that this correlation was only made qualitatively and does not add to the further discussion in the manuscript. However, we do believe it's worthwhile noting, but without taking up too much space. We have added the following text at the end of Section 3.2.2:
*While the arrows in Figure 4 indicate the possibility that the peaks, separated by a water or CH2 mass, are from the same molecule, it is only circumstantial evidence and not proof.*

Thank you for the comment, we removed panel b. And yes, we believe the high masses represent siloxanes, too.

Thank you for pointing out the missing precision of the method. We have added the following sentence to the presentation of the method (Section 2.5.2):
*For typical ambient samples the reproducibility lies below 0.3‰ for oven temperatures below 200 C and below 0.5‰ for oven temperatures above 200 C (Dusek et al., 2013).*
All results are presented with error bars, see for example panel a of Figure 6. The discussion of results in section 3.3 is in accordance with the precision values given above. Therefore, the text in section 3.3 was not changed. For example, when we wrote: 'The Δ13C values of SOA do not change significantly with temperature at desorption temperatures above 150 ∘C.', we mean significant in a statistical manner, i.e. referring to the precision.

Yes, thank you for the comment. Unfortunately, there was not enough sample available to conduct these measurements (there were only very small quantities

collected on the back filters). We would have loved to provide all corresponding back and front data, but simply can't do it as there was no more material left for analysis.

Figure 6c: This adds relatively little information in the paper, except for showing that the two techniques (IR-MS and PTR-MS) are qualitatively consistent. This could go in the SI; alternatively, the integrated, calibrated signal could give a "total carbon" measurement, which could help understand why the PTR-MS and SMPS loadings were so different.

We followed the reviewer's recommendation and removed panel c in Figure 6. The corresponding text describing panel c in section 3.3.1 was removed as well.

P. 13, line 21: The butanol-cyclohexane differences are quite large. The possible reasons for these should be discussed somewhere.

Thank you, an interesting point. We have added the following text at this point:
*The higher enrichment observed for SOA formed in the presence of 1-butanol scavenger relative to cyclohexane scavenger may be because the latter has a higher SOA yield. This is corroborated by the PTRMS data see Table 2.*

P. 14 line 13 (and P. 5 line 9): It is probably worth mentioning explicitly this is a different "lot" of the same chemical, from the same supplier.

We thank the reviewer for helping us to minimize confusion. Hence we added '(different lot)' on P.5, and '(different lot from SOA experiments)' on P.14. The lot numbers are listed explicitly in Table 1, which is mentioned in the text already.

Figure 7: The use of a heat map is unusual here, since there are just ten individual values; the interpolations/extrapolations inherent in a heat map are meaningless in this case.

We have thought about this comment. We have had many positive reactions to the figure, when presenting it to colleagues and to students. While it is formally correct that the enrichments occur at nuclei and therefore should not be presented as a topographical map, the figure shows clearly that there are isotopically hot and cold regions in alpha-pinene. Therefore, we have decided to keep the figure.

P. 15, line 18: Is this true even for the Alfa Aesar sample, which has a considerably different isotopic distribution than the others?

This is a good point, and at the end of Section 4.1 we have added to the text:
*(with the exception of the anomalous Alfa Aesar sample).*

P. 16, line 14: Some discussion of oligomerization would be useful here. Oligomerization can bring volatile carbon from the gas phase to the particle phase; and therefore it could offset some of the fractionation arising from fragmentation to small volatile species. Might that be occurring in this case?

An interesting point, yes, this is definitely possible. We have added the following to the conclusions:

[revised manuscript text omitted]

---

## Author Response (AR5)

**Reply to reviewer**
The author reply will be *italic.*

The authors have addressed my concerns; I feel this paper is suitable for publication in ACP. I have only a few minor followup comments:
*Thank you.*

- The cross correlation between the IRMS and PTR data (referenced in p. 14, line 18) is never shown explicitly; an additional figure might be useful. Alternatively, the relevant figures (5 and 6) could be mentioned in this sentence.

*Author Reply. Thank you for the comment. We did not include the figure as it does simply not show a correlation and due to space constraints. We have changed the passage to read:*
*In this experiment we have run $\alpha$-pinene ozonolysis to completion and analysed the SOA and some of the gas phase material. Comparing the IRMS (Figure 6) analysis with the PTRMS data (Figure 5), $\Delta^{13}C$ does not show a correlation with the O:C ratio.*

- P. 14, line 19 also references "aerosol oxidation state", but this is never discussed – instead only O/C is. Carbon oxidation state is actually a useful metric in this study since oligomerization reactions can involve changes in O/C (e.g., by dehydration, as mentioned in the text) but not is oxidation state.
*Author Reply. The reviewer is correct, the O:C ratio is an inaccurate proxy for the oxidation state of carbon since for example dehydration changes the ratio but not the oxidation state. We have removed this phrase from the passage, see previous reply. We also ensured that the 'aerosol oxidation state' is not mentioned elsewhere – this is not the case.*

- Figure 7: I think one can visually show "isotopically hot and cold regions" without relying on a heat map (which, as the authors admit, is essentially unphysical) – just show a large circle at each carbon atom, colored by isotopic abundance (using the same color scale as in the current figure).

*Author Reply. We followed the reviewer's line of thought and overhauled the figure as suggested.*

- Similar to my comments in the original manuscript about the Discussion, the Conclusions section treats the IRMS and PTR results completely separately; some synthesis of the two would be useful here.

*Author Reply. We modified the conclusions section and included the following statement:*

[revised manuscript text omitted]